# Twist-assisted all-antiferromagnetic tunnel junction in the atomic limit

Yuliang Chen[1], Kartik Samanta[2,3], Naafis A. Shahed[2,3], Haojie Zhang[1], Chi Fang[1], Arthur Ernst[1,4], Evgeny Y. Tsymbal[2,3] & Stuart S. P. Parkin[1✉]

Antiferromagnetic spintronics[1,2] shows great potential for high-density and ultrafast information devices. Magnetic tunnel junctions (MTJs), a key spintronic memory component that are typically formed from ferromagnetic materials, have seen rapid developments very recently using antiferromagnetic materials[3,4]. Here we demonstrate a twisting strategy for constructing all-antiferromagnetic tunnel junctions down to the atomic limit. By twisting two bilayers of CrSBr, a 2D antiferromagnet (AFM), a more than 700% nonvolatile tunnelling magnetoresistance (TMR) ratio is shown at zero field (ZF) with the entire twisted stack acting as the tunnel barrier. This is determined by twisting two CrSBr monolayers for which the TMR is shown to be derived from accumulative coherent tunnelling across the individual CrSBr monolayers. The dependence of the TMR on the twist angle is calculated from the electron-parallel momentum-dependent decay across the twisted monolayers. This is in excellent agreement with our experiments that consider twist angles that vary from 0° to 90°. Moreover, we also find that the temperature dependence of the TMR is, surprisingly, much weaker for the twisted as compared with the untwisted junctions, making the twisted junctions even more attractive for applications. Our work shows that it is possible to push nonvolatile magnetic information storage to the atomically thin limit.

Magnetoresistive random-access memory, which is based on MTJ nonvolatile memory elements, shows excellent scalability, low power consumption and potentially infinite endurance[5]. The typical ferromagnetic (FM) MTJ structure comprises a FM layer pinned by an antiferromagnetic (AF) layer, which acts as a reference layer, and a free FM layer that forms the memory layer, separated by an ultrathin nonmagnetic insulator (Fig. 1b), such as MgO (refs. 6–9). For traditional MTJs, binary resistive states (0 or 1) are accessed by switching the spin orientation of the free layer. The underlying mechanism is a spin-filtering effect that determines the junction conductance ($G$) and whose magnitude is related to the relative angle ($\theta$) of the magnetizations of the free and reference layers, which is simply described by the Slonczewski model[10], $G = \frac{G_P + G_{AP}}{2} + \frac{G_P - G_{AP}}{2}\cos\theta$, in which $G_{AP}$ and $G_P$ are the conductances for the configurations in which the free-layer magnetization is antiparallel and parallel to that of the reference layer, respectively.

AFMs inherently have ultrafast terahertz (THz) spin dynamics owing to the AF exchange coupling that leads to a very low net magnetization and, because of the resulting very low magnetic stray fields, AF spintronic devices can be closely packed. These attributes make them ideal candidates as next-generation, ultrafast, high-density information carriers. However, because ordinary collinear AFMs cannot generate net spin-polarized charge current, all-antiferromagnetic MTJs without any FM components have been rarely realized. Until very recently, using an analogous structure to FM MTJs (Fig. 1b), all-antiferromagnetic MTJs are constructed on the basis of non-collinear AFMs[3,4] (Fig. 1c), such

as $Mn_3Pt$ and $Mn_3Sn$, in which the spin-filtering effect is derived from spin chirality rather than a finite magnetization, as in conventional FM MTJs[3,4,11–15]. Although such AF MTJs largely eliminate magnetic stray fields, a low stray field still exists because of a small net magnetization in these non-collinear AFMs.

Recently explored 2D magnets provide a new platform for exploring new spintronics[16–23]. Indeed, giant magnetoresistance has been demonstrated in 2D MTJs[24–31]. The structure of 2D MTJs is distinct from that of traditional FM MTJs and recent all-antiferromagnetic MTJs in which a nonmagnetic tunnelling barrier is sandwiched by two magnetic electrodes (Fig. 1a). In the 2D MTJs, two nonmagnetic electrodes sandwich an AF 2D bilayer in which there is FM coupling within each layer but AF coupling between each layer. Such a structure can be considered an atomic MTJ in which the entire 2D layer plays the role of the tunnel barrier[26,28] (Fig. 1d). When an external field forces the interlayer-derived antiparallel magnetizations to become parallel, a giant TMR can be realized. Furthermore, multiple-layered flakes are equivalent to cascaded multiple-bilayer barriers[24,26]. Each spin arrangement at successive interfaces contributes to the final tunnelling resistance[26,32]. For example, as shown in the left diagram of Fig. 1e, all three interfaces of a four-layered magnet show antiparallel spin arrangements, resulting in a large tunnelling resistance. By contrast, the right diagram of Fig. 1e exhibits a lower tunnelling resistance owing to the parallel spin arrangement only at the mid-interface. Note that the net magnetization for both configurations vanishes, so that this

[1]Max Planck Institute of Microstructure Physics, Halle, Germany. [2]Department of Physics and Astronomy, University of Nebraska–Lincoln, Lincoln, NE, USA. [3]Nebraska Center for Materials and Nanoscience, University of Nebraska–Lincoln, Lincoln, NE, USA. [4]Institute of Theoretical Physics, Johannes Kepler University, Linz, Austria. ✉e-mail: stuart.parkin@mpi-halle.mpg.de

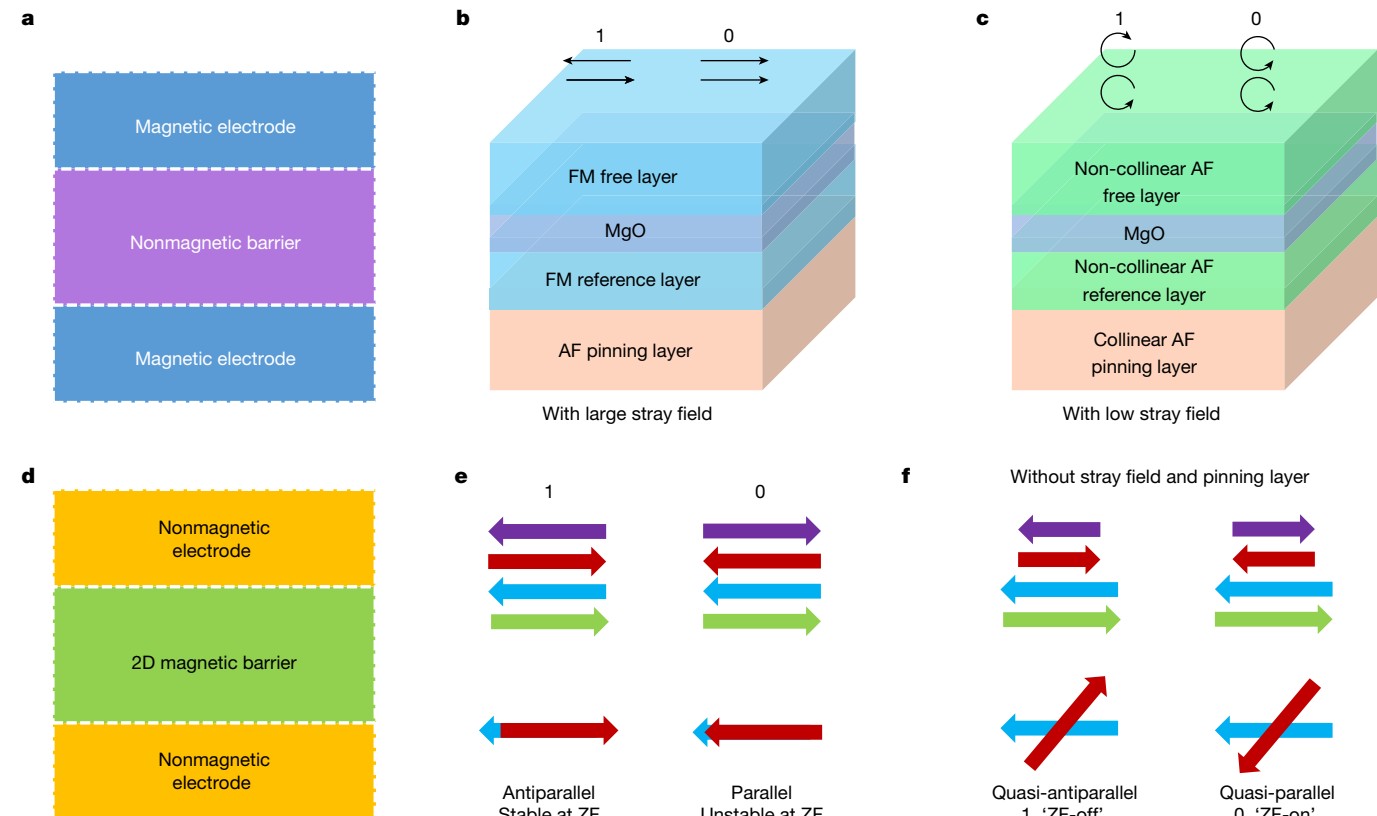

**Fig. 1 | Schematic of an all-antiferromagnetic tunnel junction down to the atomic limit. a**, Schematic structure of a conventional MTJ. **b**, A conventional MTJ with FM electrodes. **c**, A conventional MTJ based on non-collinear AF electrodes. **d**, Schematic structure of a 2D MTJ with nonmagnetic metallic electrodes and an insulating 2D magnetic tunnel barrier. **e**, Schematic diagram of the two magnetic states ('1' and '0') of an untwisted 2D MTJ. The MTJ is formed from two AF bilayers. Upper drawing shows side views of the respective spin configurations. Lower drawing shows top views of the respective spin configurations at the mid-interface. **f**, Schematic diagram of the two magnetic states (1, 'ZF-off' and 0, 'ZF-on') of a twisted 2D MTJ. Upper drawing shows side views of the respective spin configurations. Lower drawing shows top views of the respective spin configurations at the twisted interface.

layer-dependent magnetism gives rise to a promising strategy for developing all-antiferromagnetic MTJs based on 2D AFMs down to the atomic limit.

However, the right configuration shown in Fig. 1e does not exist at ZF owing to a strong interlayer exchange interaction in intrinsic 2D AFMs. Therefore, 2D MTJs formed from intrinsic AF flakes are volatile at ZF (refs. 24–28). To make a 2D all-antiferromagnetic MTJ nonvolatile at ZF, the main challenges are to pin the spin arrangement in the bottom bilayer and eliminate the inherent interlayer AF coupling at the mid-interface (Fig. 1e). Once these are met, the two spin configurations associated with each other by time-reversal can exist at ZF in the top bilayer (Fig. 1e). Here we demonstrate that twisting double bilayers (Fig. 1f) can solve the challenge of preparing nonvolatile 2D all-antiferromagnetic MTJs.

## Magnetic anisotropy of CrSBr

To use twisting to create two nonvolatile states in a 2D all-antiferromagnetic MTJ, a 2D AF system with an in-plane uniaxial magnetic anisotropy is needed. An ideal candidate is CrSBr, an A-type interlayer AF van der Waals (vdW) $n$-type semiconductor[33–35]. The paramagnetic (PM)-to-AF transition includes a substantial intralayer FM coupling that develops above the Néel temperature ($T_N$, approximately 130 K) at a characteristic temperature ($T_C$, approximately 150 K)[33,36]. CrSBr crystallizes in the orthorhombic space group *Pmmn* with crystal axes ($a \neq b \neq c$). CrSBr exhibits a strong magnetocrystalline anisotropy derived from the crystal structure such that the $b$ axis

is the easy axis and the $a$ axis is the hard axis within the $a$–$b$ plane. The uniaxial in-plane magnetic anisotropy is robust down to the monolayer limit[37].

Magnetic anisotropy is closely associated with the tunnelling behaviour in CrSBr-based MTJs. To demonstrate this, a single CrSBr bilayer MTJ is first used. Figure 2a shows a schematic of the MTJ device used here. An exfoliated CrSBr bilayer flake is sandwiched by two thin graphite electrodes, which are crossed to form a vertical tunnelling junction. The sandwich structure is further encapsulated by two hexagonal boron nitride (hBN) flakes to protect against degradation in the ambient atmosphere. Figure 2b shows an optical image of a bilayer device. The temperature-dependent conductance at ZF is plotted in Fig. 2c, with a knee appearing around 133 K manifesting the PM-to-AF transition[34]. With a further decrease in temperature, the conductance quickly falls owing to the semiconducting characteristics of CrSBr. At 2 K, we source a fixed voltage between the two graphite electrodes and find that the minimum tunnelling current appears at ZF and a large one in the high-field regime, corresponding to the spin antiparallel and parallel arrangements, respectively. The shape of the curves is relevant to the orientation of the applied field in the $a$–$b$ plane with a saturation field of about 0.18 T along the $b$ axis but progressively increasing to about 0.75 T along the $a$ axis owing to the uniaxial in-plane magnetic anisotropy of CrSBr (Fig. 1d,e and Extended Data Fig. 1a). We also measured several other few-layered CrSBr MTJs and all of them show similar properties (Supplementary Figs. 3 and 4). A giant TMR is observed in all devices, following the earlier studies on MTJs formed from CrI₃ and, similarly, all devices are volatile

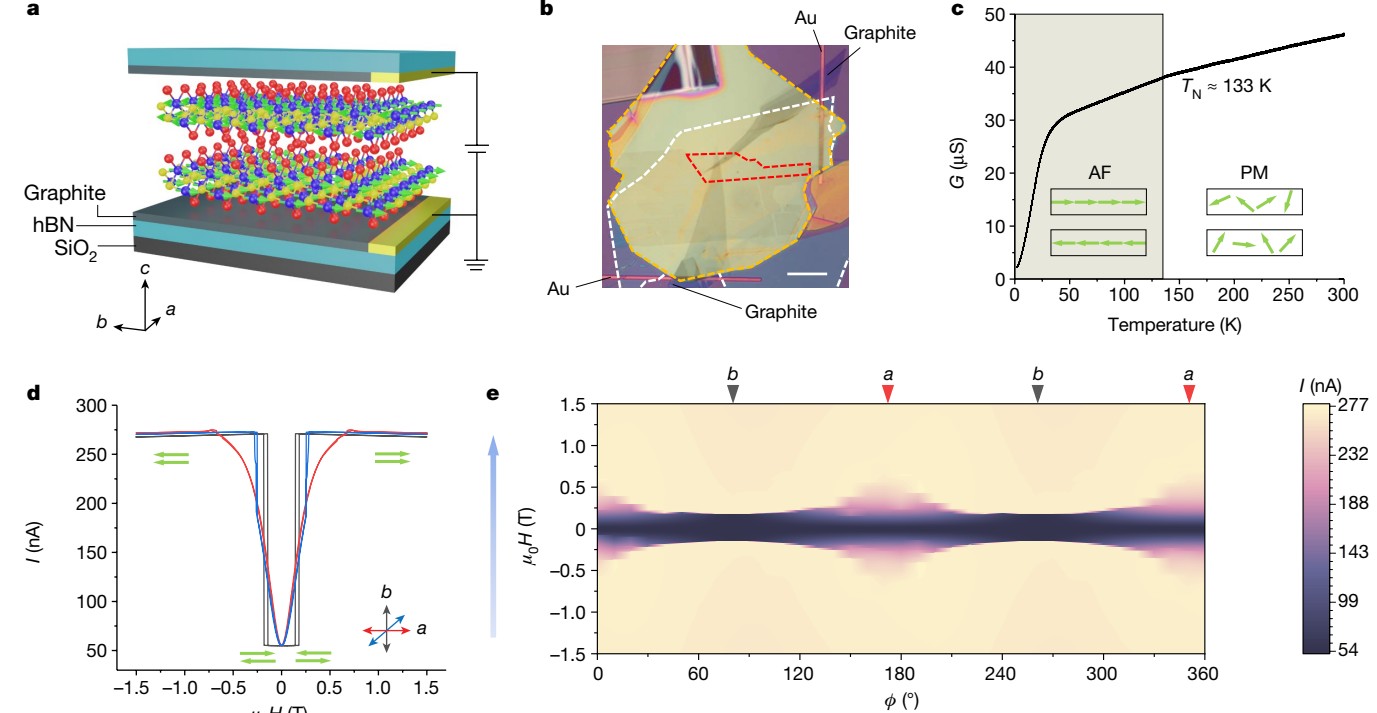

**Fig. 2 | Examining the magnetic anisotropy of CrSBr through electron tunnelling. a**, Schematic of a CrSBr-based MTJ showing a CrSBr bilayer sandwiched between thin graphite contacts. Blue, yellow and red balls correspond to Cr, S and Br, respectively. Green arrows depict spins that lie in-plane. *a*, *b* and *c* crystal axes of the CrSBr bilayer are indicated. **b**, Optical image of a device. The red dashed line outlines the CrSBr bilayer and the yellow and white dashed lines outline the top and bottom hBN layers, respectively. The graphite electrodes can be seen as the two grey regions in the figure. These overlap each other only within the area of the CrSBr flake. The Au wires that are connected to the graphite regions are only 1 μm wide and can be seen as thin reddish lines. Scale bar, 10 μm. **c**, Conductance versus temperature at ZF. The knee at 133 K indicates the Néel temperature. Drawings of the spin order in each state are given in the insets. **d**, Tunnelling current versus field at 2 K with field oriented along different directions as indicated in the inset, in which *a* and *b* are the crystal axes. A constant DC bias of 10 mV is applied. **e**, Field-orientation dependence of the tunnelling current in the *a*–*b* plane at 2 K. A constant DC bias of 10 mV is applied. The blue arrow along the *y* axis indicates the sweeping direction of the field. The angles corresponding to the crystal axes are marked by inverted triangles.

at ZF (refs. 24–27). By contrast, the TMR is negligible in a monolayer CrSBr device owing to the absence of any interlayer AF coupling (Supplementary Fig. 5). Note that all of the field-direction-dependent transport measurements in this study use a local coordinate ($\phi$) of the measurement system (see Methods). All magneto-transport experiments were conducted at 2 K, except as otherwise noted.

The large magnetic anisotropy in the *a*–*b* plane is a useful means of realizing nonvolatile 2D all-antiferromagnetic MTJs if two bilayer CrSBr flakes are stacked, misaligned with respect to each other (Fig. 1f). This results in a very different behaviour for these twisted flakes when a unidirectional field is applied. For example, when an external field is applied along the easy (hard) axis of one flake, flipping the magnetization of the other flake becomes harder (easier), namely, one of the flakes is naturally pinned without the need for the usual AF pinning layers as in conventional MTJs (Fig. 1b,c). Rotating the external field can even alternate the pinning of the two flakes. Moreover, the atomic arrangements at the twisted interface are altered, thereby largely eliminating the inherent interlayer AF coupling at the untwisted interface and, thereby, allowing for bistable spin configurations at the twisted interface at ZF, either quasi-parallel or quasi-antiparallel, as illustrated in Fig. 1f. The prefix 'quasi' means that the parallel and antiparallel configurations are not aligned perfectly as in conventional MTJs but, rather, are misaligned by means of an acute angle and an obtuse angle correlated with the twist angle. According to $G = \frac{G_P + G_{AP}}{2} + \frac{G_P - G_{AP}}{2}\cos\theta$, these bistable spin configurations should, thereby, have different tunnelling conductances.

## Twisted CrSBr bilayer/bilayer MTJs

Figure 3a is a schematic illustration of two CrSBr bilayers stacked at a 35° misalignment angle. The same device structure of Fig. 2a is used except that the single bilayer flake is replaced with two bilayers twisted to one another. The inset of Fig. 3b is the optical image of the fabricated device. The PM-to-AF transition is maintained in the twisted flake, as suggested by conductance–temperature measurements at ZF (Fig. 3b). We first sweep the field between ±1.5 T in the *a*–*b* plane and find that the inherent magnetic anisotropy in each CrSBr bilayer is robust irrespective of the twisted alignment (Extended Data Fig. 2a,b). Notably, nonvolatility (NV) emerges at ZF even though it is unstable for subsequent field sweeping (Extended Data Fig. 2a–f), which is attributed to the spin arrangement in the pinned bilayer becoming flipped after a suitably strong field is applied (see details of strong field behaviours in Methods).

To better lock the spin arrangement of the pinned bilayer, we reduce the amplitude of the sweeping field. Figure 3c shows the field-dependent tunnelling currents as a function of the angle of the applied field within the *a*–*b* plane for field swept between ±0.3 T. The top panel is backward-sweeping and the bottom panel is forward-sweeping. The inverted triangles at the top of Fig. 3c indicate the positions of the *a* and *b* axes of the top and bottom flakes. Comparing the tunnelling currents at ZF in Fig. 3c, we find that two groups of ZF NV emerge related to the *a* and *b* axes of the bottom flake, labelled by the gold and green double-headed arrows, respectively. Similar groups labelled by the purple and blue double-headed arrows are related to the

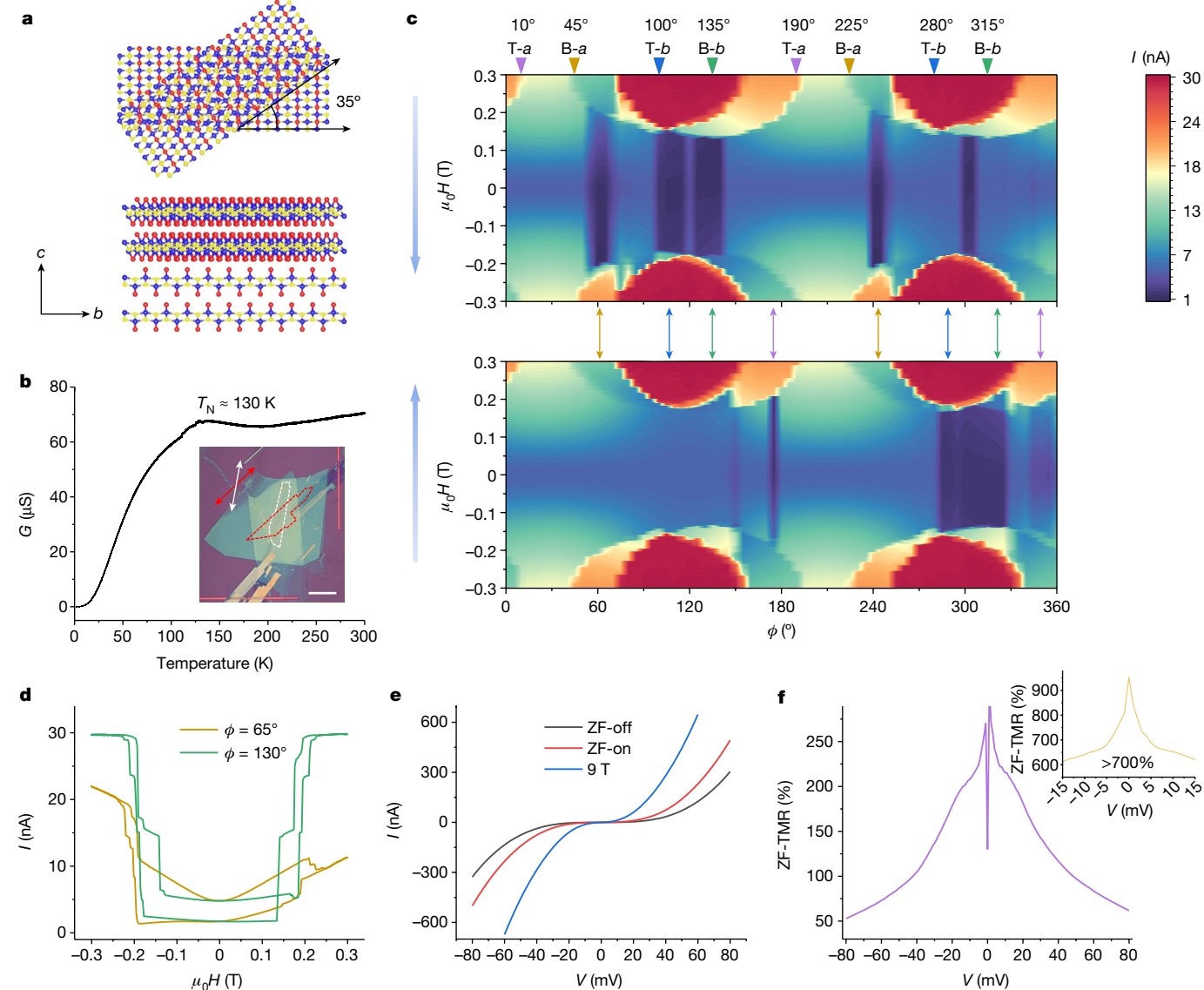

**Fig. 3 | Electrical-transport results for a 35° twisted CrSBr bilayer/bilayer MTJ. a**, Schematic of twisted CrSBr bilayers. Top view in the top panel and side view in the bottom panel. $b$ and $c$ crystal axes of the bottom bilayer are indicated. Blue, yellow and red balls correspond to Cr, S and Br, respectively. **b**, Conductance versus temperature at ZF. Inset, optical image of the device. The red and white dashed lines outline the bottom and top CrSBr bilayer flakes, respectively, and the $a$ axis of each flake is indicated by the corresponding arrows. Scale bar, 10 μm. **c**, Field-orientation dependence of the tunnelling current for field oriented within the $a$–$b$ plane. ±0.3 T field sweep range and 15 mV DC bias are used. The two vertical blue arrows along the $y$ axes indicate the sweeping direction of the field, backward sweeping for the top panel and forward sweeping for the bottom panel. The inverted triangles with angles indicate the position of the crystal axes. T-$a$, the $a$ axis of the top flake; B-$b$, the $b$ axis of the bottom flake; and so on. The double-headed arrows with different colours mark the positions at which ZF NV appears. **d**, Representatives of the two groups of ZF NV related to the easy axis ($\phi$ = 130°) and the hard axis ($\phi$ = 65°), respectively, extracted from **c**. **e**, $I$–$V$ curves at ZF and 9 T. **f**, Extracted ZF-TMR ratio as a function of bias based on the ZF $I$–$V$ curves in **e**. Inset, ZF-TMR ratio of a 10° twisted device.

top flake, although the signals are weaker, possibly because of asymmetries introduced through device fabrication. These symmetrical appearances of ZF NV can be accounted for from alternate pinning of the bottom and top flakes. Representatives of the two groups of ZF NV are plotted in Fig. 3d, which are very stable, being reproduced in ten continuous field-sweeping loops (Extended Data Fig. 2g,i). However, the $\phi$-dependent characters of the two groups are distinct. Whereas the groups related to the two $b$ axes are closely aligned along the $b$ axes, the groups relevant to the $a$ axes appear with up to an approximately 10° deviation from the precise positions of the $a$ axes. The reason for this is that, if the field is along the $a$ axis of one flake, ±0.3 T is not large enough to flip the spin arrangement in the other flake owing to the relatively small 35° twist angle, which is confirmed by the low

tunnelling currents when ±0.3 T fields are oriented along the two $a$ axes (see Fig. 3c), respectively. By contrast, a field applied at an approximately 10° deviation from one $a$ axis corresponding to roughly 45° to the other one makes a ±0.3 T field strong enough to flip the spins in the latter. To further confirm the ZF NV, we fabricated another 40° twisted device and similar results were obtained (Extended Data Fig. 3). When the sweeping field is increased to ±0.35 T, ZF NV appears at the position at which ±0.3 T cannot produce ZF NV (Extended Data Fig. 3f,g). Also note that, in these twisted bilayer/bilayer devices, when the field is oriented between the two $b$ axes, the ZF NV is unstable (Extended Data Fig. 2j) because of the weak pinning strength (see details about the twist-angle-dependent and $\phi$-dependent pinning effect in Supplementary Note 1).

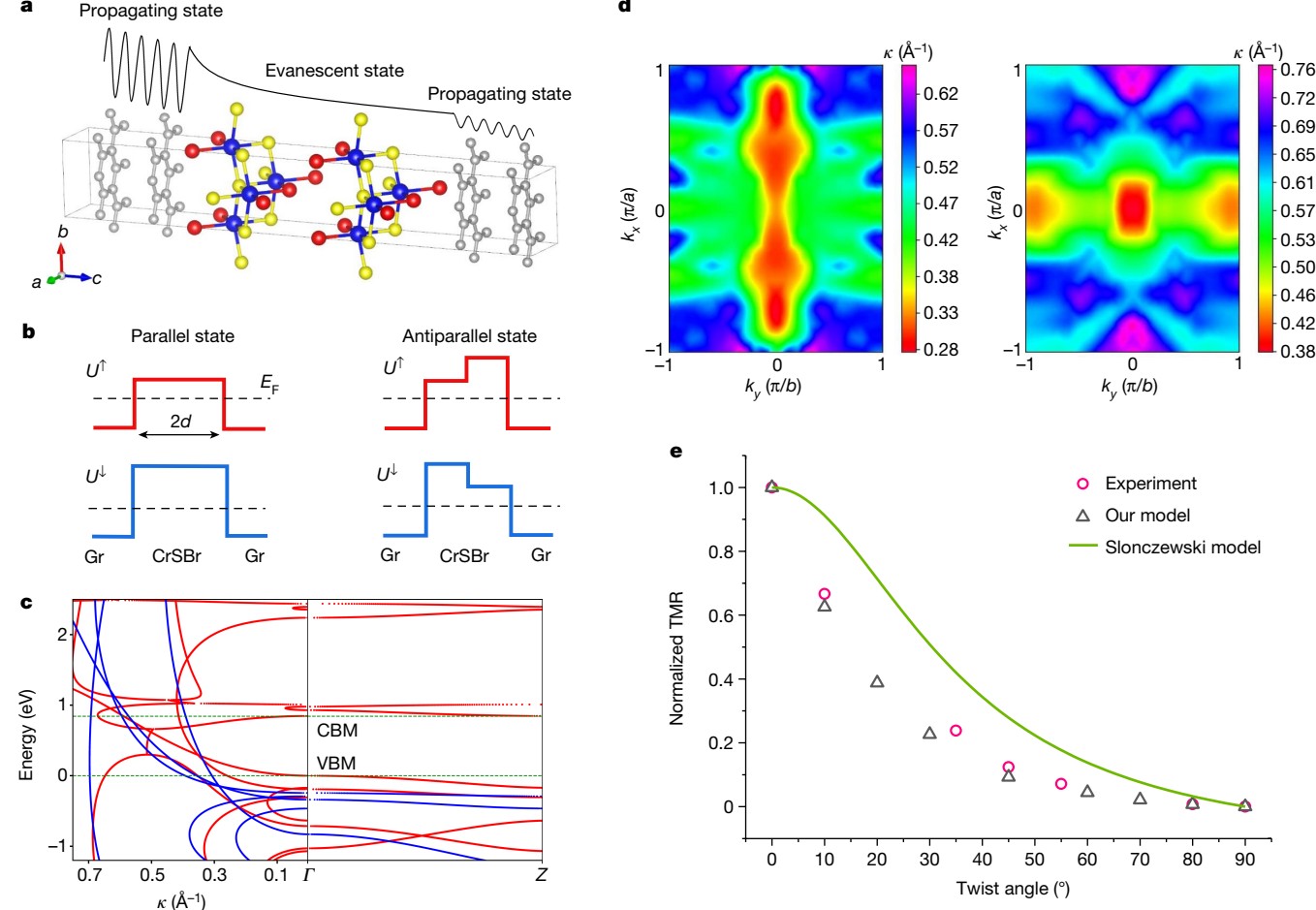

**Fig. 4 | Physical mechanism of the giant TMR effect. a**, Schematic of propagating state transmitting across the CrSBr barrier. Blue, yellow, red and grey balls correspond to Cr, S, Br and C, respectively. **b**, Potential profile for a CrSBr bilayer in parallel (left panel) and antiparallel (right panel) magnetization configurations. Gr, graphite. **c**, Complex band structure of bulk CrSBr in ferromagnetic configuration for spin-up (red dots) and spin-down (blue dots) electrons. Complex bands (left panel) connect to the real bands along the $\Gamma$–$Z$ direction (right panel) at the $\Gamma$ point. The green dashed lines show the positions

of the valence band maximum (VBM) and conduction band minimum (CBM). The VBM is set to zero energy. No real bands appear in the bandgap. **d**, Distribution of the lowest decay rates in the 2D Brillouin zone for spin-up (left) and spin-down (right) electrons calculated at $E_F$. **e**, Normalized TMR as a function of twist angle $\theta$. Grey triangles show the theoretically estimated TMR using our model and pink circles show the experimentally measured TMR. The green curve shows the calculated results based on the Slonczewski model.

Next, we investigate the performance of the 2D all-antiferromagnetic MTJ. Figure 3e is the $I$–$V$ feature of the two NV states at ZF, named 'ZF-off' and 'ZF-on' for the low-conductance state and high-conductance state, respectively, and corresponding to the magnetization configurations of '1' and '0' in Fig. 1f, respectively. The $I$–$V$ curve at 9 T is also presented for better comparison, which shows the highest tunnelling current because of parallel spin arrangements at all three interfaces. The ZF-TMR ratio calculated from the results of ZF-on and ZF-off is up to 200% (Fig. 3f) and even more than 700% in a 10° twisted device (inset of Fig. 3f and Supplementary Fig. 6a–h). By contrast, in a twisted 80° bilayer/bilayer MTJ, the ZF NV is observed with a lower ratio of 8% (Supplementary Fig. 6i–k). In a twisted 90° bilayer/bilayer MTJ, the ZF NV disappears[23] (Extended Data Fig. 5), which is expected because the spins are always orthogonally arranged at the twisted interface at ZF (Extended Data Fig. 5a). A smaller twist angle means that the quasi-antiparallel and quasi-parallel spin configurations at the twisted interface approach more closely to the perfect antiparallel and parallel arrangements, respectively, which results in enhanced ZF-TMR. However, there is a trade-off in that a smaller twist angle has a lower pinning strength (Supplementary Note 1). Moreover, high-temperature measurements demonstrate that the ZF-TMR in the twisted MTJs is robust up to close to $T_N$

(Extended Data Fig. 4). Nevertheless, the TMR ratio calculated from the conductance at 9 T and ZF-on quickly decays with increasing temperature (Extended Data Fig. 4k). These distinct temperature-dependent behaviours strongly imply the uniqueness of the twisted interface in contrast to the untwisted interface (see the reason in Methods).

## Physical mechanisms

Figure 1f shows that the twisted interface is the source of the ZF NV in the 2D all-antiferromagnetic MTJs. To confirm this, we focus on the twisted interface by fabricating twisted CrSBr monolayer/monolayer MTJs, in which ZF NV and distinct temperature-dependent behaviours are also observed (Extended Data Fig. 6 and Supplementary Note 2). Furthermore, magnetic coupling at the twisted interface must be eliminated to realize the bistable configurations of Fig. 1f at ZF, which is affirmed by the persistent ZF NV even if inserting a hBN layer into the twisted interface (Extended Data Fig. 7) and negligible interlayer exchange interaction at the twisted interface obtained from density functional theory (DFT) calculations (Methods) and by the reproduced magnetization process using the Stoner–Wohlfarth model (Supplementary Note 3).

To then uncover the tunnelling mechanism, we use an elegant model of tunnelling through a spin-dependent potential barrier ($U$) whose profile depends on the magnetization of two CrSBr monolayers, parallel or antiparallel, as schematically shown in Fig. 4b. This model is based on the band structures of CrSBr, which are greatly spin-polarized with different bandgaps (and thus tunnelling barrier heights) for up-spin and down-spin electrons (Extended Data Fig. 8). We assume that the lowest imaginary part $\kappa^{\uparrow,\downarrow}$ of the evanescent states of the CrSBr barrier largely determine the decay rates and fully control the probability of tunnelling $\approx e^{-2\kappa^{\uparrow,\downarrow}d}$, in which $d$ is half of the barrier thickness (that is, here equal to one monolayer; Fig. 4a) and the indices $\uparrow$ and $\downarrow$ denote spin up and down, respectively. The $\kappa^{\uparrow,\downarrow}$ can be obtained by calculating the complex band structure of bulk CrSBr (refs. 7,38–41). For example, Fig. 4c shows the calculated spin-dependent complex bands of CrSBr for in-plane vector $\vec{k}_{\parallel} = (k_x, k_y) = 0$. Figure 4d further shows the distribution of the lowest $\kappa^{\uparrow,\downarrow}$ over the 2D Brillouin zone at the Fermi energy ($E_F$). Using these decay rates and assuming that $\vec{k}_{\parallel}$ is conserved in the tunnelling process (that is, coherent tunnelling), we then calculate the tunnelling transmission ($T$) and TMR for the spin-dependent tunnelling barriers as follows.

For the untwisted CrSBr bilayer, when the magnetization of two CrSBr monolayers is parallel (P), the total transmission is approximated by

$$T_P \propto \frac{1}{N_k} \sum_{k_{\parallel}} \left[ e^{-2\kappa^{\uparrow}(\vec{k}_{\parallel})d} e^{-2\kappa^{\uparrow}(\vec{k}_{\parallel})d} + e^{-2\kappa^{\downarrow}(\vec{k}_{\parallel})d} e^{-2\kappa^{\downarrow}(\vec{k}_{\parallel})d} \right] \qquad (1)$$

in which $N_k$ is the total number of $k$-points in our calculation configuration. For the antiparallel (AP) configuration of the two CrSBr monolayer magnetizations, the total transmission is given by

$$T_{AP} \propto \frac{1}{N_k} \sum_{k_{\parallel}} \left[ e^{-2\kappa^{\uparrow}(\vec{k}_{\parallel})d} e^{-2\kappa^{\downarrow}(\vec{k}_{\parallel})d} + e^{-2\kappa^{\downarrow}(\vec{k}_{\parallel})d} e^{-2\kappa^{\uparrow}(\vec{k}_{\parallel})d} \right] \qquad (2)$$

Using equations (1) and (2) and the calculated decay rates $\kappa^{\uparrow,\downarrow}(\vec{k}_{\parallel})$, we estimate TMR $= \frac{T_P - T_{AP}}{T_{AP}}$ to be 1,790%, which is in qualitative agreement with the experimentally measured value of about 1,050% (Extended Data Fig. 1d).

Combining the above model and the Slonczewski model[10] (Methods), we further take into account the effect of twisting with an angle $\theta$ on the transmission probability and obtain

$$\text{TMR}(\theta) = \frac{T(\theta) - T(\pi - \theta)}{T(\pi - \theta)} = \frac{[T_P(\theta) - T_{AP}(\theta)]\cos\theta}{T_{AP}(\theta) + [T_P(\theta) - T_{AP}(\theta)]\sin^2\frac{\theta}{2}} \qquad (3)$$

in which $T(\theta)$ is the total transmission as a function of $\theta$, which manifests that the spin-dependent potential barrier of CrSBr is twist-angle-dependent owing to the rotations of the in-plane wavevector and spin (see Methods). Extended Data Table 1 shows the fitted tunnelling barriers from the experimental $I-V$ curves[42,43]. Figure 4e shows the calculated normalized TMR $= \frac{\text{TMR}(\theta)}{\text{TMR}(0)}$ in comparison with the experimental results, indicating excellent agreement for twist angles varying from 0° to 90°. By contrast, a large deviation from the experimental results is found when using the standard Slonczewski model, which instead shows the important role of coherent tunnelling in our CrSBr MTJs (see Methods).

Our work establishes a compelling strategy to achieve all-antiferromagnetic MTJs in the atomic limit. Using CrSBr, ZF NV is achieved up to around 120 K. Although the materials used in our MTJs preclude operation above about 125 K, we anticipate that the concepts and physical models we present in our paper can be extended to other 2D magnets with higher magnetic transition temperatures in the future.

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

## Methods

### Device fabrication

We prepared electrical contacts on Si/SiO$_2$ (285 nm) substrates by standard electron-beam lithography (Raith PIONEER Two) followed by Ti (2 nm)/Au (10 nm) or Ti (2 nm)/Au (30 nm) deposition (scia Coat 200) to form the electrodes. These electrodes are distributed in a 'T' shape. All flakes (HQ Graphene) were exfoliated onto Si/SiO$_2$ (285 nm) substrates using Scotch Magic Tape. The exfoliation for thin graphite and hBN was carried out under ambient conditions and for CrSBr was carried out under inert conditions in a N$_2$-filled glovebox with <3 ppm O$_2$ and <1 ppm H$_2$O content. The thickness of the CrSBr flakes was initially identified by optical contrast, further examined by Raman spectroscopy (Horiba LabRAM HR Evolution) and atomic force microscopy (Bruker Dimension Icon); see Supplementary Fig. 1. The layer-dependent magnetism deriving from magneto-transport measurements confirmed the thickness of the flakes. The vdW assembly of the hBN/thin graphite/CrSBr/thin graphite/hBN heterostructure was carried out in the same glovebox by sequentially picking up each flake using a poly(bisphenol A carbonate)/polydimethylsiloxane stamp[44]. To insert a hBN monolayer into the twisted interface, an extra pick-up process is required. We used two strategies to align the twist angles. For large exfoliated CrSBr flakes, the tear-and-stack method was used[45,46] (Supplementary Fig. 2a–c). On the other hand, we noticed that the exfoliated CrSBr flakes are always in a ribbon shape with a long edge along the a axis and a short edge along the b axis[33,37], so we aligned small flakes with respect to their long edges (Supplementary Fig. 2d–f). The twist angles based on both of these strategies match well with the experimental results, confirming their reliability. The finished assembly on the stamp was released onto the SiO$_2$/Si substrate with pre-prepared electrodes and letting the thin graphite electrodes make contact with two of them. The residual poly(bisphenol A carbonate) was dissolved in chloroform before the electrical-transport measurements were carried out.

### Electrical-transport measurements

All electrical measurements were conducted in a PPMS DynaCool cryostat (Quantum Design) with a base temperature of 1.8 K and a magnetic field up to 9 T. The conductance–temperature results were measured in a two-terminal configuration using a Model 7270 DSP lock-in amplifier to source a 1 mV AC voltage and to measure the current using a 13.333 Hz reference frequency, with a warm-up recipe at 1 K min$^{-1}$ being used. The other measurements were performed in a two-terminal configuration using a Keithley 2450 SourceMeter to source a DC voltage and measure current. The DC results were verified by AC measurements showing no substantial difference. The tunnelling junction dominates the final magnetoconductance owing to the high conductivity of the thin graphite contacts. The thickness of the prepared Ti/Au electrodes does not influence the experimental results. Note that a two-terminal configuration is widely used in 2D MTJs[26,27]. The Shubnikov–de Haas oscillations of graphite electrodes observed in CrI$_3$ MTJs induced by out-plane field sweeping[25] were well suppressed in our experiments because we were only concerned with in-plane magneto-transport studies. The TMR ratio is given by $(G_h - G_l)/G_l$, in which $G_l$ and $G_h$ are the low conductance and high conductance, respectively. All field-direction-dependent transport measurements used the pre-calibrated local coordinates of the PPMS system and the angle is defined as $\phi$. We used a unified method to mount the devices onto the electrical-transport puck by taking the 'T'-distributed Ti/Au electrodes as the reference object. The '|' electrode and the other two '−' electrodes were aligned along the side edge and bottom edge of the puck by hand, respectively, resulting in the direction of the '|' electrode being almost along the direction of the magnetic field when $\phi = 0°$. This unified custom method helped us to examine the twist angles and locate the crystal axes of CrSBr flakes. We estimate that the uncertainty in angle is about ±5°.

### DFT calculations for tunnelling mechanism

The electronic structure and the optimal interlayer separation are carried out based on DFT using the plane-wave projector augmented wave method[47], as implemented in the Vienna Ab initio Simulation Package (VASP)[48,49]. We used the Perdew–Burke–Ernzerhof exchange-correlation functional[50] within the generalized gradient approximation (GGA). The electron–electron correlation effects beyond the GGA are taken into account using the Hubbard U correction through the GGA + U method[51] with U value of 4.0 eV and Hund's coupling energy J of 0.8 eV on the Cr 3d orbitals. For the self-consistent calculations, a plane-wave basis set with a plane-wave cutoff of 500 eV and a k-point mesh of 12 × 8 × 4 and 12 × 8 × 1 is used for the bulk CrSBr and thin films, respectively. The structural optimization is carried out using VASP maintaining the symmetry of the heterostructure. The positions of the atoms are relaxed towards equilibrium until the Hellmann–Feynman forces become less than 0.01 eV Å$^{-1}$.

Calculations of the complex band structure are performed using the non-equilibrium Green's function formalism (DFT + NEGF approach)[52,53], as implemented in Synopsys QuantumATK[54]. In QuantumATK, the nonrelativistic Fritz Haber Institute pseudopotentials are used with a single-zeta-polarized basis and the cutoff energy is set to 150 Ry. The spin-polarized GGA + U method is used in our calculations with the same U and J values on the Cr 3d orbitals as in VASP. A k-point mesh of 18 × 16 × 10 is used for bulk CrSBr. Periodic boundary conductions are assumed for the transverse direction (x−y plane) and open boundary conditions along the transport z direction. Experimentally measured lattice constants of a = 3.540 Å, b = 4.755 Å and c = 8.394 Å for the bulk CrSBr are assumed in the calculations[33]. Figure 4d shows the distribution of the decay rates calculated at the energy of 0.1 eV below the conduction band minimum (CBM), that is, $E_F = E_{CBM} - 0.1$ eV, considering that CrSBr behaves as an n-type semiconductor. We also calculated the distribution of the decay rates in the 2D Brillouin zone at other energies and obtained TMR ratios ranging from about 200% to about 20,000% (Extended Data Fig. 8), which corroborates that our system inherently has a large TMR effect.

To take into account the effect of twisting on TMR, we assume that the in-plane wavevector $\vec{k}_\parallel$ characterizing the Bloch states of the two CrSBr layers, one twisted with respect to the other, is conserved in the tunnelling process, that is, coherent tunnelling. This approximation neglects the difference between the momentum and quasi-momentum and is applicable to a sufficiently thick vacuum barrier. When the top CrSBr monolayer is twisted with respect to the bottom monolayer, the wavevectors of the bottom monolayer remain unchanged as the reference, whereas the wavevectors of the twisted top monolayer are transformed as follows

$$k'_x = k_x\cos\theta + k_y\sin\theta \tag{4}$$

$$k'_y = -k_x\sin\theta + k_y\cos\theta \tag{5}$$

Then the total transmission for the P and AP states as a function of twist angle $\theta$ can be obtained from

$$T_P(\theta) \propto \frac{1}{N_k} \sum_{k_\parallel} \left[ e^{-2\kappa^\uparrow(\vec{k}_\parallel)d}e^{-2\kappa^\uparrow(\vec{k}_\parallel)d} + e^{-2\kappa^\downarrow(\vec{k}_\parallel)d}e^{-2\kappa^\downarrow(\vec{k}_\parallel)d} \right] \tag{6}$$

$$T_{AP}(\theta) \propto \frac{1}{N_k} \sum_{k_\parallel} \left[ e^{-2\kappa^\uparrow(\vec{k}_\parallel)d}e^{-2\kappa^\downarrow(\vec{k}_\parallel)d} + e^{-2\kappa^\downarrow(\vec{k}_\parallel)d}e^{-2\kappa^\uparrow(\vec{k}_\parallel)d} \right] \tag{7}$$

Equations (6) and (7) reflect the effect of twist on the transmission owing to the rotation of the in-plane wavevector but with collinear magnetizations. In the experiment, however, the magnetizations

of the two CrSBr monolayers are non-collinear owing to the strong magnetocrystalline anisotropy. Therefore, the transmissions need to be calculated when the angle between the magnetizations of the two monolayers is $\theta$ (the same angle as for the structural rotation) and $\pi - \theta$ (magnetization of the top monolayer switched by 180°). Using the same idea as the Slonczewski model[10], we quantize the spin of the top monolayer with respect to the spin axis of the bottom layer, obtaining

$$T(\theta) \propto T_{\mathrm{P}}(\theta)\cos^2\frac{\theta}{2} + T_{\mathrm{AP}}(\theta)\sin^2\frac{\theta}{2} \tag{8}$$

$$T(\pi - \theta) \propto T_{\mathrm{P}}(\theta)\sin^2\frac{\theta}{2} + T_{\mathrm{AP}}(\theta)\cos^2\frac{\theta}{2} \tag{9}$$

We can see that, at $\theta = 0°$, equations (8) and (9) become equations (1) and (2), respectively. The TMR ratio for the twisted case as a function of $\theta$ is then given by

$$\mathrm{TMR}(\theta) = \frac{T(\theta) - T(\pi - \theta)}{T(\pi - \theta)} = \frac{[T_{\mathrm{P}}(\theta) - T_{\mathrm{AP}}(\theta)]\cos\theta}{T_{\mathrm{AP}}(\theta) + [T_{\mathrm{P}}(\theta) - T_{\mathrm{AP}}(\theta)]\sin^2\frac{\theta}{2}}$$

that is, equation (3). From the calculated distribution of the lowest decay rates as a function of $\vec{k}_{\parallel} = (k_x, k_y)$ at $E_{\mathrm{F}} = E_{\mathrm{CBM}} - 0.1$ eV, we compute the distribution of $\vec{k}'_{\parallel} = (k'_x, k'_y)$ as a function of different twist angle $\theta$. Using equations (6), (7) and (3), we then calculate TMR as a function of twist angle $\theta$.

Apart from non-collinear magnetizations, our model also considers coherent tunnelling, that is, conserved in-plane wavevector as discussed above. By contrast, in the Slonczewski model, $G = \frac{G_{\mathrm{P}} + G_{\mathrm{AP}}}{2} + \frac{G_{\mathrm{P}} - G_{\mathrm{AP}}}{2}\cos\theta$, only the former is considered. Applying the experimental TMR ratio of the untwisted bilayer $\frac{G_{\mathrm{P}} - G_{\mathrm{AP}}}{G_{\mathrm{AP}}} = 1{,}050\%$ (Extended Data Fig. 1d) in $G = \frac{G_{\mathrm{P}} + G_{\mathrm{AP}}}{2} + \frac{G_{\mathrm{P}} - G_{\mathrm{AP}}}{2}\cos\theta$, we obtain the green curve in Fig. 4e.

To investigate the optimal interlayer separation, that is, the vdW gap, for the untwisted CrSBr bilayer, initially, in the AF configuration, we used the bulk lattice parameters and the bulk vdW gap of 2.78 Å at room temperature. By varying the vdW gap, we observed an optimal separation of around 2.39 Å at 0 K, which is 0.39 Å smaller than the bulk gap at room temperature, which is reasonable. Furthermore, we also explored the FM configuration, although it is not the experimental ground state of CrSBr. Notably, we found that the optimum vdW gap remained unchanged, indicating consistency across different magnetic configurations in CrSBr.

We further extended similar calculations to a twisted CrSBr structure. To facilitate the calculation, we used the following formula[55]

$$\theta = \cos^{-1}\left[\frac{(q^2 + p^2)\cos 2\delta + q^2 - p^2}{(q^2 - p^2)\cos 2\delta + q^2 + p^2}\right] \tag{10}$$

and selected $\theta = 37.12°$, a commensurate twist angle suitable for an orthorhombic structure determined by setting $p = 1$, $q = 2$ and $2\delta$ as the angle indicated by the blue lines in Supplementary Fig. 7d. When analysing the interlayer separation versus energy for the twisted structure, we find that the vdW gap at 0 K is greatly increased to 3.36 Å. This outcome indicates that the twist operation induces a decoupling effect on the layers, thereby reducing the interlayer AF coupling and electron hopping. To further illustrate the weakening of interlayer coupling, we conducted projected density of states (DOS) calculations for a layer within this twisted CrSBr bilayer while maintaining the other layer unchanged (Supplementary Fig. 8e). For simplicity, we ensured that the magnetic atoms within each layer had the same spin direction. The projected DOS of an individual monolayer of CrSBr was also calculated with a similar spin configuration (Supplementary Fig. 8d).

## DFT calculations for exchange interactions

The calculations were performed using a self-consistent Green function code Hutsepot based on the multiple-scattering theory, specially designed for semi-infinite systems real-space clusters[56,57]. The calculations were performed within the GGA[50]. Strongly localized Cr 3d electrons were treated using a combination of a self-interaction correction and the Slater transition-state methods as implemented within the multiple-scattering method[58–61]. Similar results were also obtained using a GGA + U method[51,62]. We used the maximum angular momentum cutoff of $L_{\max} = 3$. The Brillouin zone integration was done using a tetrahedron method adapted for 2D geometry[63]. The other parameters needed are similar to the previous calculation for the tunnelling mechanism. There were estimated exchange parameters $J_{ij}$ entering the Heisenberg model

$$H = -\frac{1}{2}\sum_{i,j} J_{ij}\,\mathbf{e}_i\mathbf{e}_j \tag{11}$$

Here the unit vectors are $\mathbf{e}_i = \mathbf{S}_i/|\mathbf{S}_i|$ at site $i$, in which $\mathbf{S}_i$ is the localized spin moment. The exchange parameters were calculated using the magnetic force theorem as it is implemented within the multiple-scattering theory[64,65]

$$J_{ij} = \frac{1}{8\pi}\int_{-\infty}^{E_{\mathrm{F}}} dE\,\mathrm{Im}\mathrm{Tr}_L\left(\Delta_i G_{\uparrow}^{ij}\Delta_j G_{\downarrow}^{ji} + \Delta_i G_{\downarrow}^{ij}\Delta_j G_{\uparrow}^{ji}\right) \tag{12}$$

in which $G_{\uparrow(\downarrow)}^{ij}$ is a Kohn–Sham Green's function between the sites $i$ and $j$ for the spin-up (spin-down) channel and $\Delta_i$ is a magnetic interaction vertex function. Supplementary Fig. 9 shows the schematic of the magnetic exchange interactions. $J_1$ to $J_7$ denote the intralayer exchange interactions and $J_{z1}$ and $J_{z2}$ denote the interlayer exchange interactions. The calculated exchange parameters are shown in Extended Data Table 2. The first column shows that the magnetic interaction within each monolayer of CrSBr bulk is strongly ferromagnetic ($J_1$ to $J_7$ are mainly positive) and, between the neighbouring layers, it is antiferromagnetic ($J_{z1} < 0$, $J_{z2}$ is small). The magnetic coupling is mainly mediated by superexchange: within a monolayer through S $sp$ orbitals (S has an induced magnetic moment of $0.08\mu_{\mathrm{B}}$) and between two monolayers through Br $sp$ orbitals (with the induced magnetic moment of $0.03\mu_{\mathrm{B}}$). Cr and Br atoms form a long linear bond under the angle of 180°, Cr–Br–Br–Cr (Supplementary Fig. 9b), which is responsible for an AF interaction ($J_{z1} = -0.25$). In the same time, $J_{z2}$ remains relatively small in magnitude because: (1) the distance between Cr moments over the vdW gap is large for a direct coupling and (2) the bonds between Cr and the neighbouring anions (S and Br) are rotated by a large angle (the bond strength between two monolayers in this direction is rather weak; see Supplementary Fig. 9b). The theoretically obtained Néel temperature 132 K using a random-phase approximation was found to be in good agreement with the experiment. We also calculated the exchange parameters for CrSBr monolayer and bilayer as shown in Extended Data Table 2. The values near the exchange parameters of the CrSBr bulk prove that FM coupling is robust to the monolayer and AF coupling is robust to the bilayer.

Then we investigated the exchange interactions in twisted CrSBr bilayers with twist angles of 45° and 90°, respectively. To calculate exchange parameters in a twisted bilayer, the Green function in equation (12) was calculated using the transformation

$$\widetilde{G}_{LL'}^{ij}(E) = \sum_{L''L'''} U_{LL''}(s_i; E)\, G_{L''L'''}^{ij}(E)\, U_{L'L'''}(s'_j; E) \tag{13}$$

in which $G_{LL'}^{ij}(E)$ are Green function matrix elements and $U_{LL'}(s_i; E)$ is the transformation matrix for a displacement $s_i$ (refs. 66,67). The calculated exchange parameters are also shown in Extended Data Table 2. The

intralayer coupling is changed only marginally, whereas the interlayer coupling was substantially suppressed. The reason for this is that the Cr–Br–Br–Cr bonds are weakened on twist operation: Cr and Br are not lying on one line but are tilted; see Supplementary Fig. 9c,d. This leads to a marked decrease of superexchange between Cr atoms through Br $sp$ orbitals across the vdW gap.

Overall, the DFT results show a negligible interlayer magnetic coupling at the twisted interface and a strong intralayer FM coupling, which support that both quasi-parallel and quasi-antiparallel spin alignments are stable at ZF, as shown in Fig. 1f. Therefore, such bistable spin alignments at the twisted interface can be operated by sweeping external field, being well described by Stoner–Wohlfarth model calculations (Supplementary Note 3). Moreover, the DFT results rather establish a strong interlayer AF coupling at the untwisted interfaces, which means that the net magnetization is zero in the twisted bilayer/bilayer MTJs.

## Strong field behaviours of twisted bilayer/bilayer MTJs

For the 35° twisted bilayer/bilayer MTJ, forward and backward sweeping between −1.5 T and 1.5 T is applied in the $a–b$ plane (Extended Data Fig. 2a,b). Compared with Fig. 2e and Extended Data Fig. 1a, a twofold rotational symmetry is preserved, but the smooth relationship between the orientation of the field and the saturation field in the case of single bilayer MTJ is broken, as shown in Extended Data Fig. 2a,b. A distinct feature is that two tiny humps appear in the vicinity of the two $a$ axes. The angle between the positions of the two humps precisely matches the twist angle. These results demonstrate that the original magnetic anisotropy in the intrinsic CrSBr bilayer is robust, irrespective of the twisted alignment, and strongly support that each bilayer in the twisted structure is largely independent and magnetically decoupled from each other. Apart from the inherent physical properties of CrSBr, the decoupling may be a result of the large twist angle as distinct from small-angle-twisted moiré superlattices (usually less than 10°)[46,68–72], in which the interlayer coupling mechanism is prevalent. To further experimentally confirm whether there is substantial interlayer magnetic coupling in our system, we also performed control experiments on a 30° twisted CrSBr bilayer/hBN monolayer/CrSBr bilayer MTJ (Supplementary Fig. 10), together with the twisted monolayer/hBN monolayer/ monolayer MTJ (Extended Data Fig. 7). The inserted hBN layer would reduce or eliminate the hypothetical coupling at the twisted interface, but we still observe ZF NV in both cases, which affirms the aforementioned independence and interlayer decoupling in line with the DFT calculations. Further experimental evidence is that the symmetry of the tunnelling current curve is mirrored, as well as the values of the two ZF tunnel currents being maintained after a large field application (Extended Data Fig. 2g,h) owing to the spin configuration being flipped to its time-reversal copy in the pinned bilayer, which is reproduced by Stoner–Wohlfarth model calculations in Supplementary Note 3. This is also the reason why the ZF NV is unstable using strong field sweeping (Extended Data Fig. 2c–f).

## Distinct temperature-dependent behaviours between twisted and untwisted interfaces

In the twisted bilayer/bilayer MTJ (Extended Data Fig. 4k), the high-temperature measurements demonstrate that the ZF-TMR calculated from the conductances of ZF-on versus ZF-off resulting from the twisted interface is robust up to close to $T_N$. By contrast, the TMR ratio calculated from the conductance at 9 T and ZF-on quickly decays with increasing temperature. Note that the twisted bilayer/bilayer has three interfaces, that is, one twisted interface and two untwisted interfaces (within each bilayer). Because the spin alignments are parallel at all three interfaces of the twisted bilayer/bilayer MTJ at 9 T and ZF-on corresponds to quasi-parallel spin alignment at only the twisted interface (the right side of Fig. 1f), the TMR from 9 T versus ZF-on mainly originates from the two untwisted interfaces. Thus, these distinct temperature-dependent behaviours strongly imply the uniqueness

of the twisted interface in contrast to the untwisted interface. We then confirmed this discrepancy by comparing an untwisted bilayer MTJ and a twisted bilayer MTJ (Extended Data Fig. 6f). Note that the marked differences in the temperature-dependent variations of TMR for the twisted and untwisted interfaces are independent of the applied bias (Supplementary Figs. 11 and 12).

The reason for this is ascribed to the decoupling mechanism at the twisted interface. In our 2D MTJs, the whole CrSBr stack plays the role of a tunnelling barrier. As a semiconductor, CrSBr is insulating at low temperatures, which well serves as the tunnelling barrier, whereas CrSBr becomes conductive with increasing temperature (Fig. 2c), resulting in degradation of the tunnelling barrier, which can thereby naturally explain the fast decay of the TMR in the untwisted MTJs. Nevertheless, this cannot account for the slow decay behaviour in the twisted MTJs. We have already noted the important difference between the untwisted and twisted interfaces from DFT calculations of the interlayer exchange interactions (Extended Data Table 2). Now we consider the energy-band structure. Supplementary Fig. 8a,b shows the energy-band structure of the untwisted CrSBr bilayer, in which the energy band of the top and bottom monolayers (Supplementary Fig. 8c) hybridizes or overlaps well, which can be understood from a more intuitive picture. Supplementary Fig. 8d shows the projected DOS calculated from an individual CrSBr monolayer, from which it is found that Cr $d$ orbitals dominate the DOS near the CBM. For the case of two untwisted monolayers, the Cr $d$ orbitals in each monolayer are well aligned to allow for an effective interlayer orbital interaction by means of the intervening Br atoms, resulting in superexchange across Cr–Br–Br–Cr. Thus, increasing temperature can effectively enhance the electron interlayer hopping across the vdW gap. Such thermal activation is spin-independent and we surmise that this mechanism progressively dominates the vertical conductance of the untwisted bilayer on warming. However, because the Cr $d$ orbitals are highly spatially anisotropic, rotating one monolayer with respect to the other should substantially reduce the effective interlayer orbital interaction[73]. Supplementary Fig. 8e shows the projected DOS of the top monolayer of a twisted bilayer. Comparing Supplementary Fig. 8e with Supplementary Fig. 8d, the slight difference indicates that interlayer orbital interaction with the bottom monolayer is indeed suppressed at the twisted interface in accordance with the negligible interlayer magnetic coupling suggested by DFT calculations. Accordingly, we suggest that the thermally activated spin-independent interlayer hopping is less in the twisted MTJs and the spin-dependent tunnelling mechanism still contributes part of the vertical conductance. Experimental evidence for this is a slower increase of conductance with warming (Extended Data Fig. 6b) compared with the conductance–temperature curve of the untwisted MTJ (Fig. 2c). On the other hand, we find that the vdW gap becomes physically thicker in the twisted MTJs (Supplementary Fig. 13). DFT calculations confirm that a twisted structure with a thicker vdW gap is energetically favourable (Supplementary Fig. 7). Therefore, the thicker vdW gap can still serve as a robust tunnelling barrier (even though the CrSBr sheets themselves become more conducting) up to close to $T_N$, giving a robust weakly temperature-dependent TMR in the twisted MTJs.

## Data availability

Source data are available at Zenodo repository[74] (https://doi.org/10.5281/zenodo.12209639) and from the corresponding author on request.

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

**Acknowledgements** We acknowledge financing from the Deutsche Forschungsgemeinschaft (DFG, German Research Foundation) – project no. 443406107, Priority Programme (SPP) 2244. The research work at University of Nebraska–Lincoln (theoretical modelling of the transport properties of twisted CrSBr bilayers) was supported by grant number DE-SC0023140 financed by the U.S. Department of Energy, Office of Science, Basic Energy Sciences (K.S., N.A.S. and E.Y.T.). A.E. acknowledges funding from the Fonds zur Förderung der Wissenschaftlichen Forschung (FWF) under grant number I 5384. Part of the calculations was performed at the Max Planck Computing and Data Facility.

**Author contributions** Y.C. and S.S.P.P. conceived the project. Y.C. fabricated the devices and performed most of the experiments and data analysis. H.Z. performed the Raman spectroscopy measurements and drew the crystal schematics. C.F. supplied instructions for magneto-transport measurements. Y.C. performed the Stoner–Wohlfarth model calculations. K.S., N.A.S. and E.Y.T. performed theoretical modelling of the transport properties of twisted CrSBr bilayers on the basis of the DFT calculations. A.E. performed the DFT calculations for exchange interactions. All authors discussed the results. Y.C. and S.S.P.P. wrote the manuscript.

**Funding** Open access funding provided by Max Planck Society.

**Competing interests** The authors declare no competing interests.

**Additional information**
**Correspondence and requests for materials** should be addressed to Stuart S. P. Parkin.

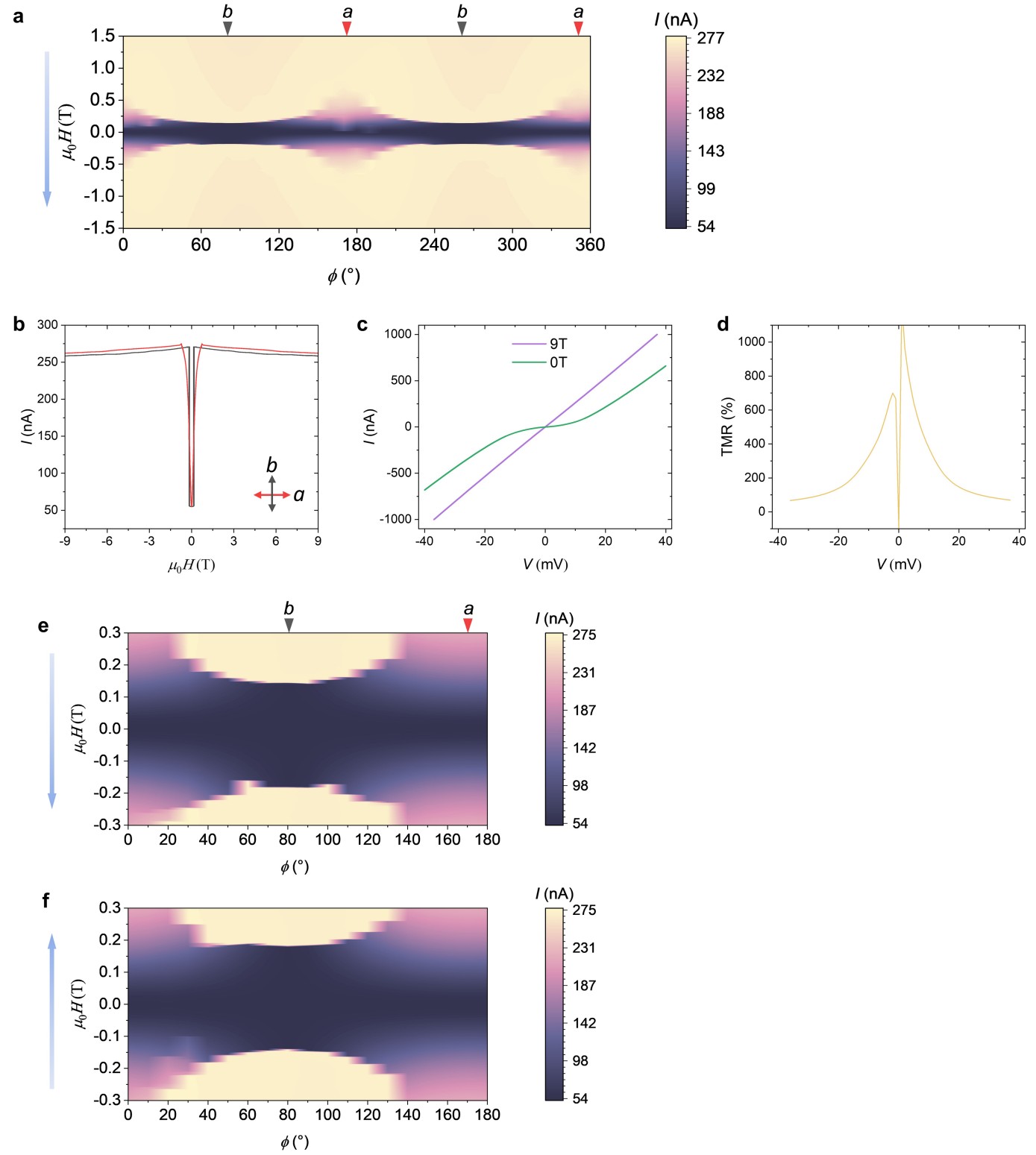

**Extended Data Fig. 1 | Further electrical-transport results of a single bilayer CrSBr MTJ. a**, Field-orientation dependence of the tunnelling current in the $a$–$b$ plane at 2 K. A constant DC bias of 10 mV is applied. The blue arrow indicates the field-sweeping direction. **b**, Tunnelling current versus field at 2 K with field oriented along different directions, as indicated in the inset. A constant DC bias of 10 mV is applied. **c**, $I$–$V$ curves at ZF and 9 T. **d**, Extracted TMR ratio as a function of bias, based on the $I$–$V$ curves in **c**. **e**–**f**, Field-orientation dependence of the tunnelling current in the $a$–$b$ plane at 2 K. The field is swept between ±0.3 T. A constant DC bias of 10 mV is applied. The two blue arrows indicate the field-sweeping direction, backward sweeping in **e** and forward sweeping in **f**.

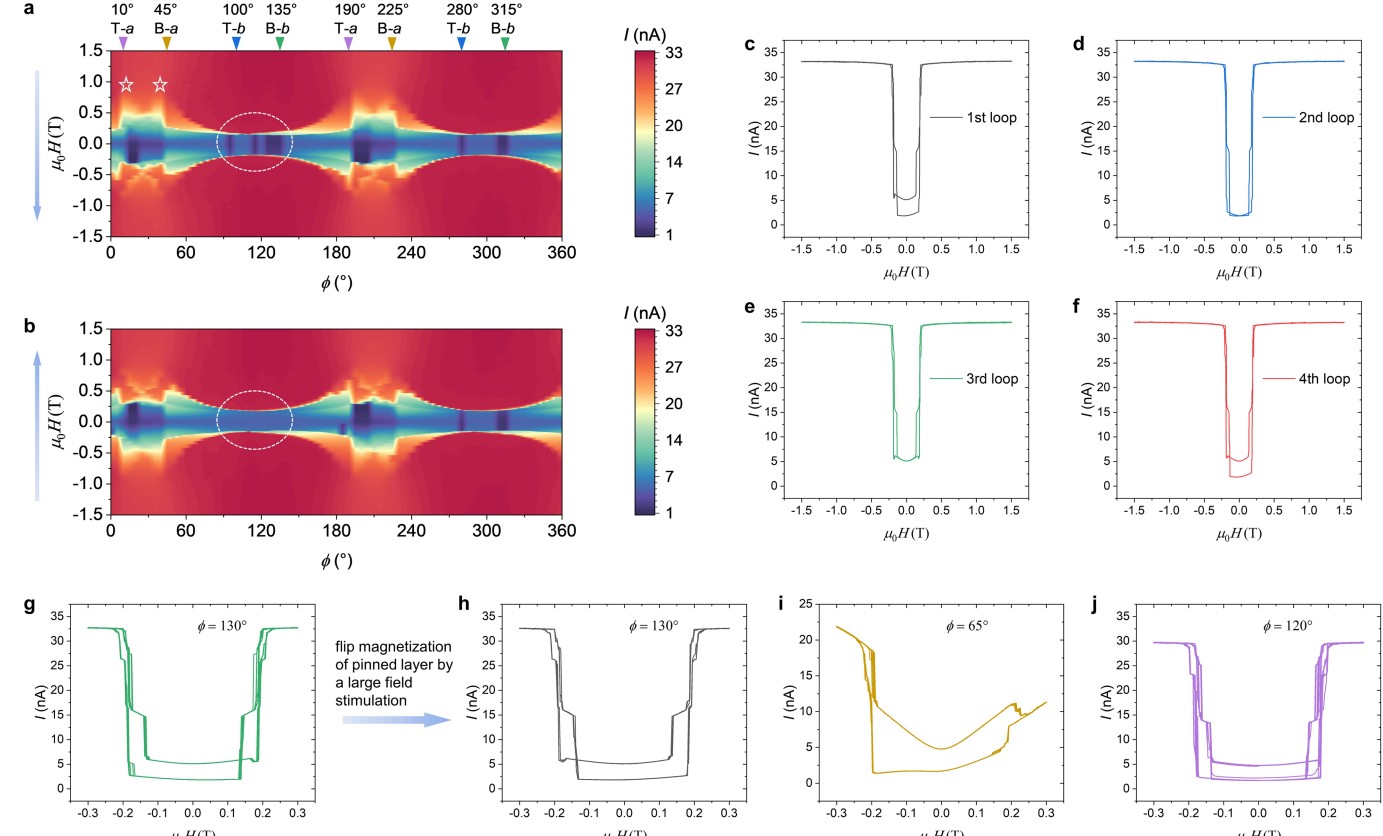

**Extended Data Fig. 2 | Further electrical-transport results of a 35° twisted bilayer/bilayer CrSBr MTJ at 2 K. a,b**, Field-orientation dependence of the tunnelling current in the *a–b* plane. The two blue arrows indicate the field-sweeping direction, backward sweeping in **a** and forward sweeping in **b**. ±1.5 T field and 15 mV DC bias are used. Two humps appear in the vicinity of the *a* axes of each flake, as marked by stars. ZF NV related to *b* axes of each flake is observed in the range outlined by the white dotted ellipse. **c**–**f**, Four current versus magnetic field hysteresis loops at $\phi$ = 130°. ZF NV appears in the first loop (**c**) and fourth loop (**f**). However, the second loop (**d**) and third loop (**e**)

both show similar low and high conductivities, respectively. This is because of the pinned bilayer being flipped after a sufficiently strong field is applied. ±1.5 T field and 15 mV DC bias are used. **g**–**j**, Several current versus magnetic field hysteresis loops. **g**, Ten loops for the field oriented along $\phi$ = 130°. **h**, Three loops for the field oriented along $\phi$ = 130° after a large field stimulation. **g** and **h** are mirror images of each other. **i**, Ten loops for the field oriented along $\phi$ = 65°. **j**, Ten loops for the field oriented along $\phi$ = 120°, showing unstable ZF NV. ±0.3 T field and 15 mV DC bias are used.

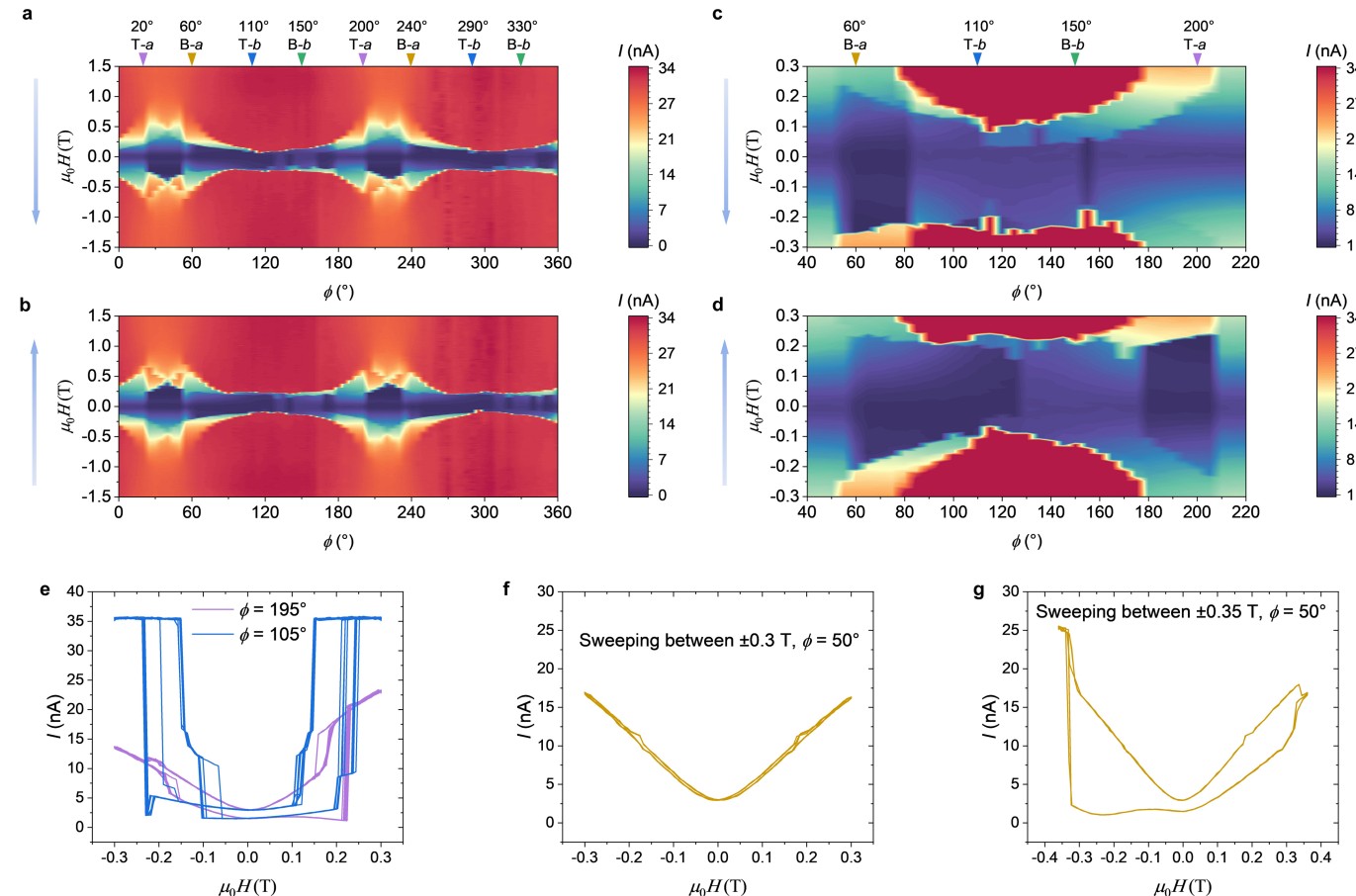

**Extended Data Fig. 3 | Electrical-transport results for a 40° twisted bilayer/bilayer CrSBr MTJ at 2 K. a,b**, Field-orientation dependence of the tunnelling current in the $a$–$b$ plane. The two blue arrows indicate the field-sweeping direction, backward sweeping in **a** and forward sweeping in **b**. **c**,**d**, Same as **a** and **b** except for field swept between ±0.3 T. **e**, Measured tunnelling current for ten repeated hysteresis loops for the field oriented at various angles. **f**,**g**, Measured tunnelling current for varying maximum swept field. A constant DC bias of 1 mV is applied in all measurements.

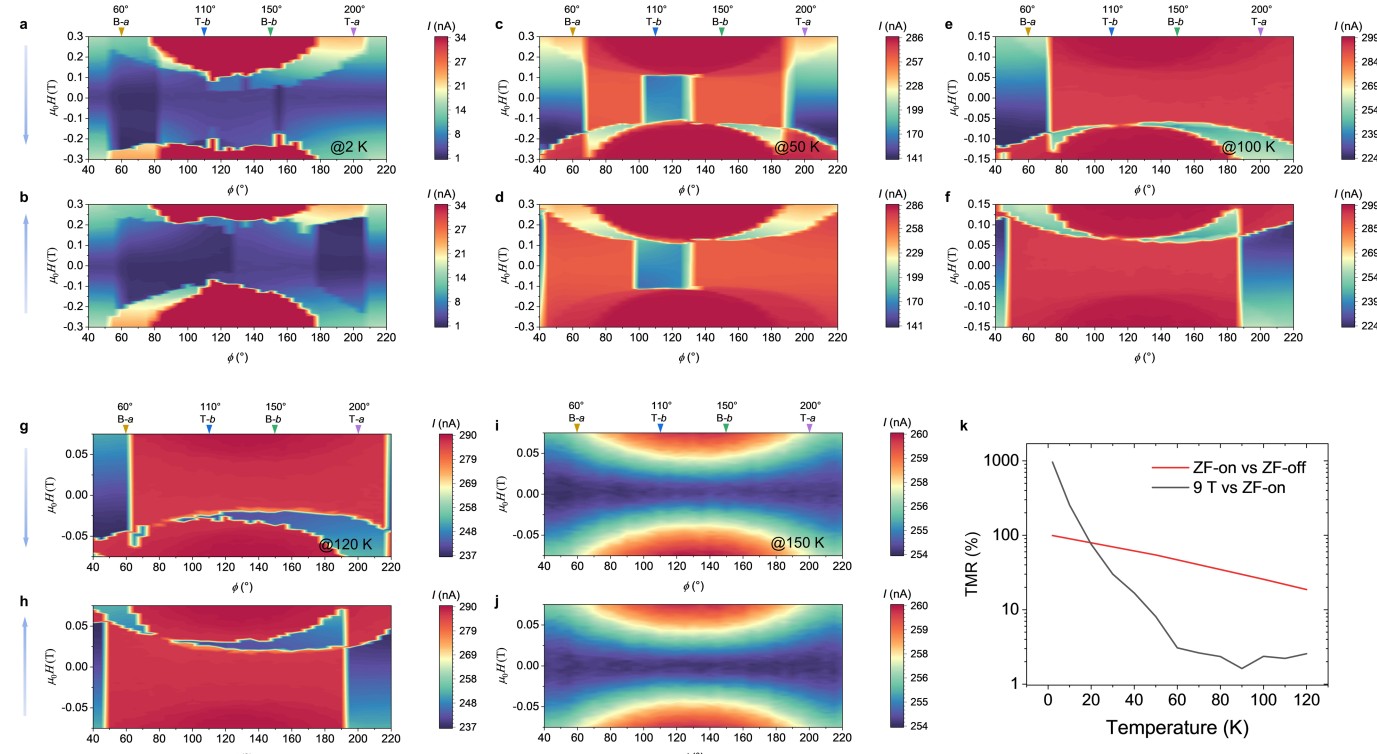

**Extended Data Fig. 4 | Electrical-transport results of a 40° twisted bilayer/ bilayer CrSBr MTJ at different temperatures. a**–**j**, Field-orientation dependence of the tunnelling current in the *a*–*b* plane at different temperatures. 2 K in **a** and **b**, 50 K in **c** and **d**, 100 K in **e** and **f**, 120 K in **g** and **h** and 150 K in **i** and **j**. The blue arrows indicate the field-sweeping direction. A constant DC bias of 1 mV is applied in all measurements. **k**, TMR ratio with 1 mV DC bias at different temperatures.

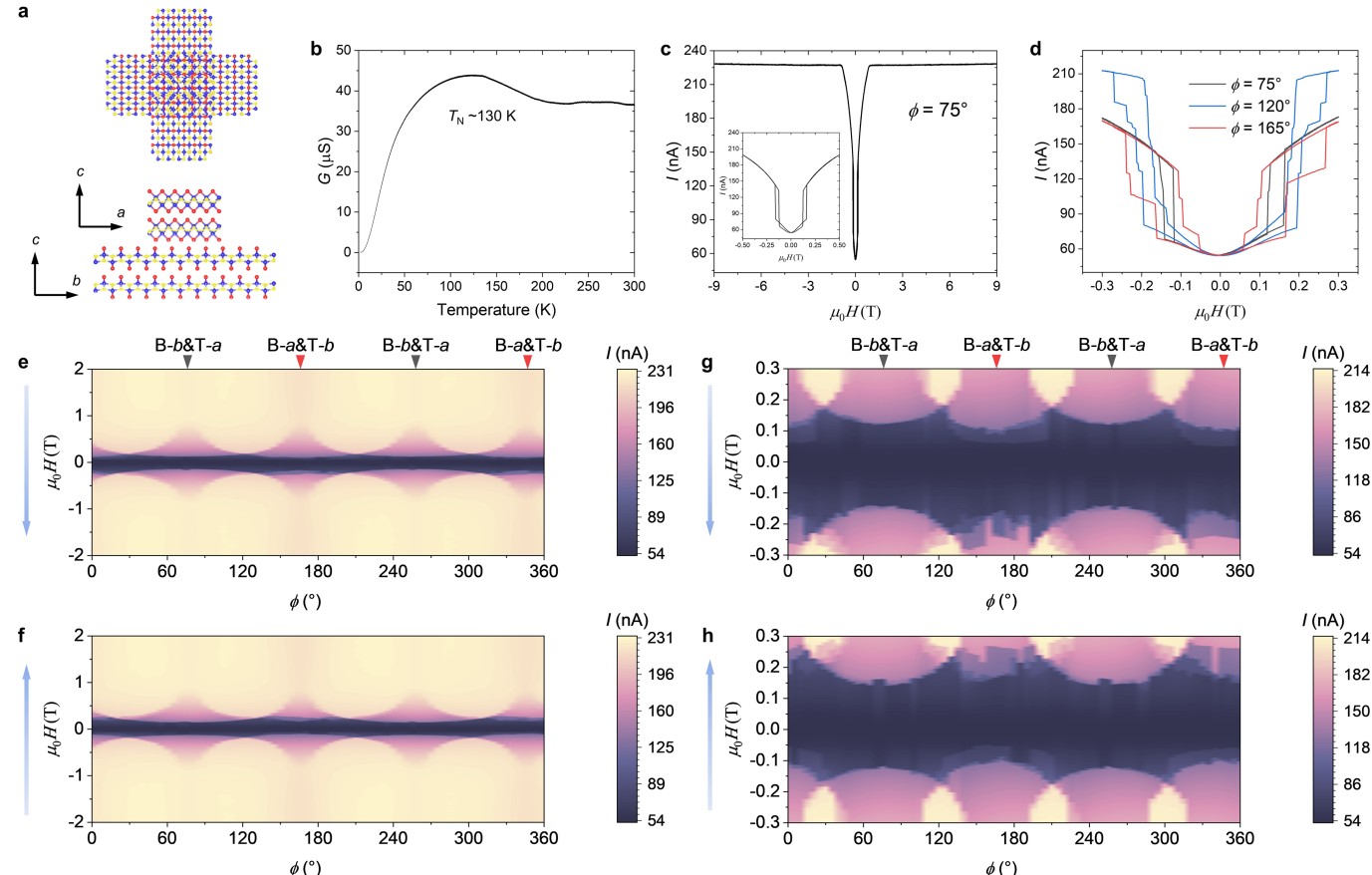

**Extended Data Fig. 5 | Electrical-transport results of a 90° twisted bilayer/bilayer CrSBr MTJ at 2 K. a**, Schematic of orthogonally stacked CrSBr bilayers. Top view in top panel and side view in bottom panel. The *b* (*a*) and *c* axes of the bottom (top) bilayer are indicated. **b**, Conductance versus temperature at ZF. **c**, Tunnelling current versus field with field oriented along the easy (hard) axis of the bottom (top) bilayer. The inset shows a close-up near ZF. ±9 T field is used. **d**, Tunnelling current versus field for field oriented along various angles. ±0.3 T field is used. **e**,**f**, Field-orientation dependence of the tunnelling current in the *a*–*b* plane. The two blue arrows indicate the field-sweeping direction. ±2 T field is used. **g**,**h**, Same as **e** and **f** except for field swept between ±0.3 T. A constant DC bias of 40 mV is used in **c**–**h**.

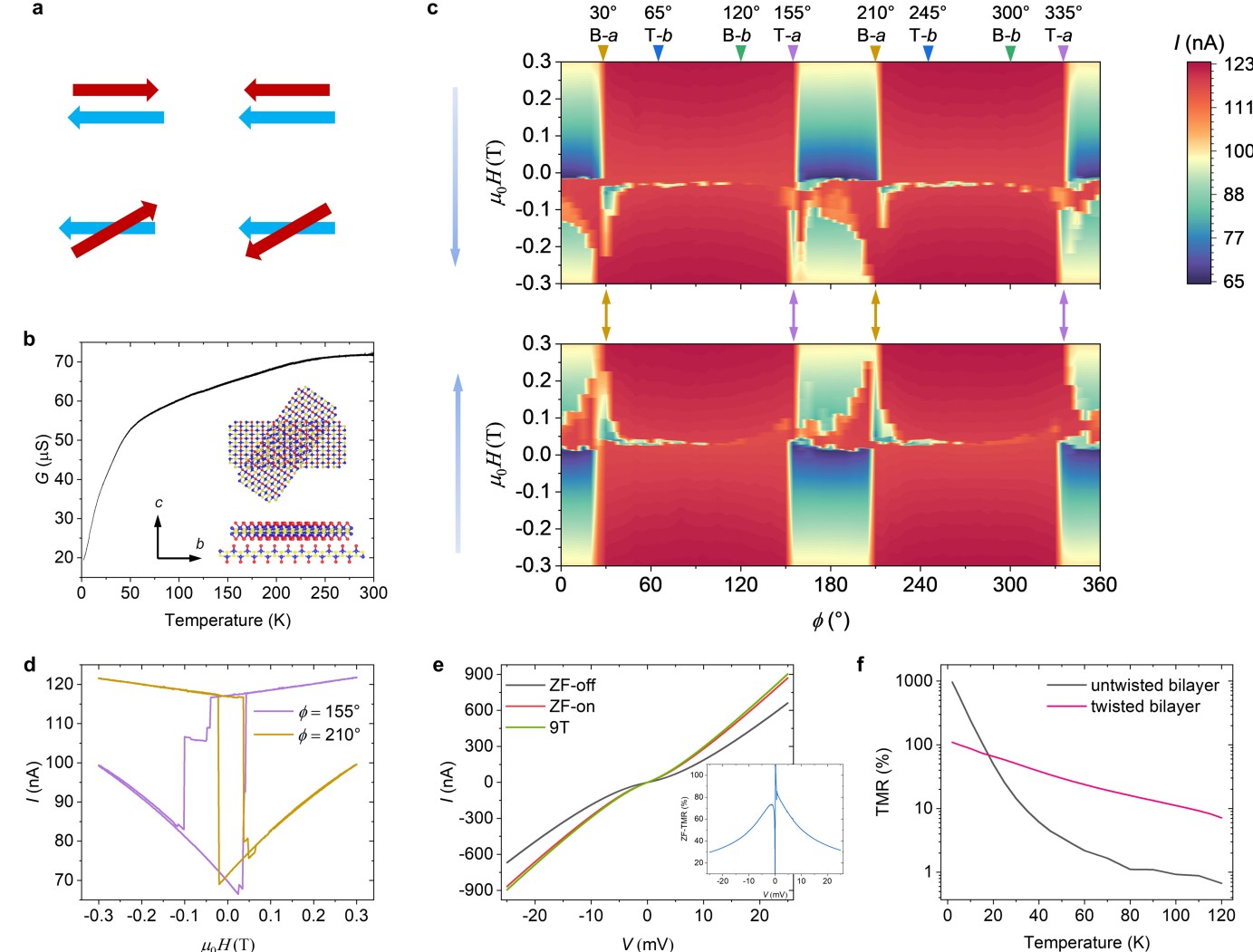

**Extended Data Fig. 6 | Electrical-transport results for a 55° twisted CrSBr monolayer/monolayer MTJ. a**, Schematic of twisted spins. Top panel, side view; bottom panel, top view. **b**, Conductance versus temperature at ZF. Inset, schematic of twisted CrSBr monolayers. **c**, Field-orientation dependence of the tunnelling current for field oriented within the $a$–$b$ plane. ±0.3 T field and 5 mV DC bias are used. **d**, Representatives of the two groups of ZF NV related to the $a$ axis of the top ($\phi = 155°$) and bottom ($\phi = 210°$) flakes, extracted from **c. e**, $I$–$V$ curves at ZF and 9 T. Inset, extracted ZF-TMR ratio as a function of bias, based on the ZF $I$–$V$ curves. **f**, TMR ratio as a function of temperature. 2 mV DC bias is used. The pink curve is ZF-TMR obtained from a 45° twisted CrSBr monolayer/ monolayer MTJ (Supplementary Note 2). The grey curve is TMR obtained from an untwisted bilayer MTJ (Extended Data Fig. 1) using its 1.5 T and ZF conductances.

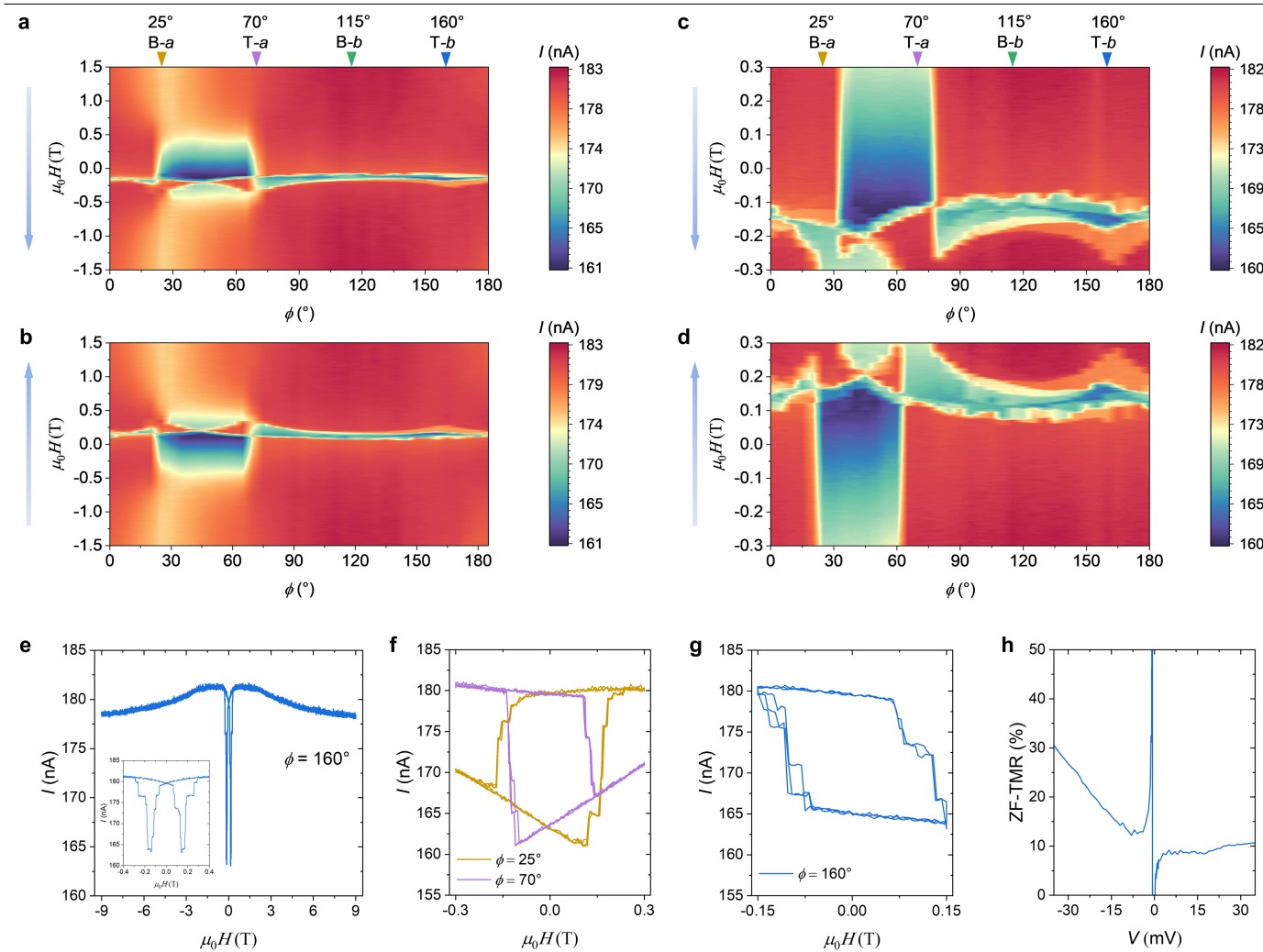

**Extended Data Fig. 7 | Electrical-transport results for a 45° twisted CrSBr monolayer/hBN monolayer/CrSBr monolayer MTJ. a**,**b**, Field-orientation dependence of the tunnelling current in the *a*–*b* plane. The two blue arrows indicate the sweeping direction of field. ±1.5 T field is used. **c**,**d**, Same as **a** and **b** except for sweeping field between ±0.3 T. **e**, Tunnelling current versus field with field oriented along the easy axis of the top CrSBr flake. The inset shows a close-up of ZF adjacency. ±9 T field is used. **f**, Demonstration of ZF NV related to the hard axes. ±0.3 T field is used. **g**, Demonstration of ZF NV related to the easy axis. ±0.15 T field is used. **h**, ZF-TMR. A constant DC bias of 25 mV is applied in **a**–**g**.

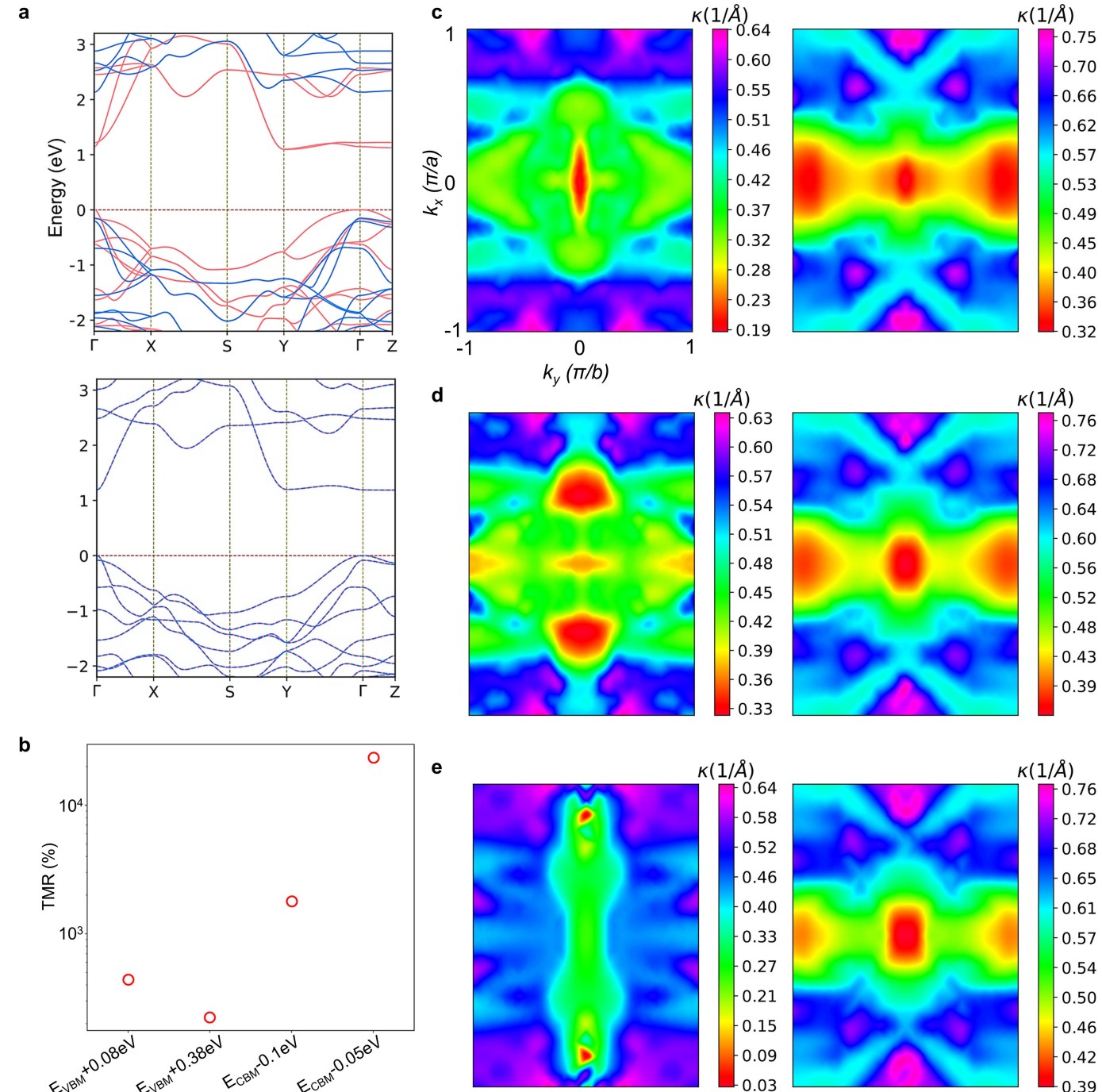

**Extended Data Fig. 8 | Band structure of CrSBr bulk and distribution of the decay rates in the 2D Brillouin zone at other energies. a**, Spin-polarized band structure of bulk CrSBr in the (001) plane of the 2D Brillouin zone for parallel (upper panel) and antiparallel (lower panel) magnetization. The valence band maximum is set to have zero energy (horizontal dashed line). In the upper panel, the red curves represent the up-spin bands and the blue curves represent the down-spin bands. **b**–**e**, TMR ratio $\frac{T_P - T_{AP}}{T_{AP}}$ (**b**) calculated at four different energies using the decay rates in **c**–**e** and Fig. 4d. The energies are $E = E_{VBM} + 0.08$ eV (**c**), $E = E_{VBM} + 0.38$ eV (**d**) and $E = E_{CBM} - 0.05$ eV (**e**). In **c**–**e**, the left panels are for up-spin and the right panels are for down-spin.

**Extended Data Table 1 | Fitted barrier height of the devices**

| | untwisted 2L | 45° twisted 1L/1L | 55° twisted 1L/1L | 10° twisted 2L/2L | 35° twisted 2L/2L | 80° twisted 2L/2L |
|---|---|---|---|---|---|---|
| Junction area (um²) | 0.26 | 2.51 | 1.81 | 0.40 | 0.28 | 2.29 |
| Fitted barrier height (eV) | 0.525 (P) 0.761 (AP) | 0.804 (qP) 0.914 (qAP) | 0.766 (qP) 0.833 (qAP) | 0.224 (qP) 0.297 (qAP) | 0.256(qP) 0.301 (qAP) | 0.251(qP) 0.254(qAP) |

Magnetization configuration at the twisted interface. P, parallel; AP, antiparallel; qP, quasi-parallel; qAP, quasi-antiparallel.

**Extended Data Table 2 | Calculated exchange parameters ($J_i$ in meV; see Supplementary Fig. 9) for CrSBr bulk, monolayer (ML), bilayer (BL) and twisted bilayers (TBL) with twist angles of 45° and 90°**

| $J_i$ | bulk | ML | BL | TBL (45°) | TBL (90°) |
|---|---|---|---|---|---|
| $J_1$ | 2.59 | 3.01 | 2.74 | 2.83 | 2.91 |
| $J_2$ | 6.13 | 5.83 | 6.74 | 6.52 | 6.43 |
| $J_3$ | 1.50 | 1.93 | 1.82 | 1.71 | 1.67 |
| $J_4$ | 0.03 | -0.07 | 0.17 | 0.11 | 0.19 |
| $J_5$ | 0.05 | -0.01 | -0.02 | 0.01 | -0.01 |
| $J_6$ | -1.19 | -0.89 | -1.23 | -1.13 | -1.37 |
| $J_7$ | 0.96 | 1.23 | 0.83 | 1.01 | 0.94 |
| $J_{z1}$ | -0.25 | | -0.31 | -0.005 | -0.0003 |
| $J_{z2}$ | 0.01 | | -0.01 | -0.002 | 0.0001 |