## [Peer Review File · Nature]

Manuscript Title: Twist-assisted all-antiferromagnetic tunnel junction in the atomic limit

Editorial Notes:

Redactions – Third Party Material

Reviewer Comments & Author Rebuttals

Reviewer Reports on the Initial Version:

Referees' comments:

Referee #1 (Remarks to the Author):

In this manuscript, Chen et al. introduce a pioneering approach to constructing all-antiferromagnetic tunnel junctions at the atomic level by employing a twisting strategy. This work focuses on the utilization of two-dimensional antiferromagnet, CrSBr bilayers. Remarkably, a significant nonvolatile tunneling magnetoresistance ratio exceeding 700% is achieved at zero field, with the van der Waals gap functioning as the tunnel barrier. The results strongly suggest the presence of a spin-filtering effect at the twisted interface, an observation further bolstered by experiments involving twisted CrSBr monolayers. This investigation holds immense promise for the advancement of nonvolatile magnetic information storage at the atomic dimension.

Despite the inherent merits of the work, there are some queries and points of contention that ought to be addressed:

- 1) The introduction of the twisting way for constructing antiferromagnetic tunnel junctions is lacked. It would be beneficial if the authors can elucidate more on the underlying principles and mechanisms governing this strategy, especially in comparison to traditional ways.
- 2) The manuscript claims an impressive nonvolatile tunneling magnetoresistance ratio with CrSBr bilayers. How does this result compare with other known antiferromagnetic materials? Are there specific advantages to using CrSBr for this purpose?
- 3) The van der Waals gap is highlighted as the tunnel barrier in this study. Could the authors provide further insight into its role and significance? How does it contribute to the observed tunneling magnetoresistance ratio, and what advantages does it offer over other potential tunnel barriers?
- 4) The observed spin-filtering effect at the twisted interface is intriguing. It would be valuable if the authors could delve deeper into its origins and implications (using Density Functional Theory calculations to further investigate the twisted/non-twisted interfaces' electronic properties would be one of the useful means). How does this effect influence the overall performance of the all-antiferromagnetic tunnel junctions? Moreover, are there conditions or factors that might enhance or mitigate this effect?

5) In pushing the boundaries to achieve atomic-level storage, there could be potential challenges and limitations. It would be informative if the authors can address any technical or theoretical hurdles faced during their study or those they anticipate in future applications of this technology.

This manuscript holds considerable significance in the realm of antiferromagnetic spintronics. It offers a fresh perspective and promises groundbreaking advancements in the domain of high-density, ultrafast information devices. I am inclined to endorse this work for publication in Nature, provided the points raised above are adequately addressed and clarified.

Referee #2 (Remarks to the Author):

The authors report a tunneling magnetoresistance study of twisted 2D antiferromagnet CrSBr. A careful in-plane magnetic field orientation dependence on the tunneling current is measured in twisted CrSBr double bilayers as well as double monolayers. Two groups of magnetic states are identified as the magnetic field is swept up and down. The observation is attributed to a twisted interface mechanism, and zero field nonvolatility is claimed. The results are interesting, and the data quality is high, but several concerns and questions are listed below.

(1) The main discovery is attributed to the twisted interface, and this is certainly a popular topic recently. However, a solid and conclusive explanation of the twisted interface mechanism is missing in this study. Although the twisted interface can potentially tune the interlayer magnetic coupling, the dominating effect is still the intrinsic magnetic anisotropy in individual layers of CrSBr. Based on the tunneling current, the authors try to assign “quasi-AFM coupling” and “quasi-FM coupling”, however, this data can only help assign a spin configuration, which is mainly determined by magnetic anisotropy and crystal axes here, but cannot identify the interlayer magnetic coupling. The direct evidence for the twisting effect on the interlayer magnetic coupling is missing.

(2) There are also no theoretical calculations or simulations to support the twisted interface mechanism and the assigned “quasi-FM/FM coupling”, which will involve a careful study of the local layer stacking and interlayer magnetic interactions. Moreover, in all the magnetic field orientation dependence data, multiple transitions and kinks have been measured and shown in the colormap. However, no magnetic states or spin configurations are assigned. The authors should add theoretical calculations and simulations to identify, support, and label these magnetic states.

(3) Several orientation data look just like an overlapping pattern of the two individual bilayers, which seems to suggest that the two bilayers may be completely magnetic decoupled, for example, in the Extended Data Fig. 16. In other words, it can be just a result of two separate groups of spin configurations, which are controlled by the external magnetic field, magnetic anisotropy, and their crystal axes (the relative twist angle), but not the twisted interface. It will be worth a careful examination of the role of the twisted interface, which only induces the relative twist angle, or also leads to an exotic interlayer magnetic coupling. This question can be potentially answered by adding a thin layer of BN between the two CrSBr layers as a control experiment, where magnetic coupling will be suppressed.

(4) If the current experimental results can indeed be explained based on just the spin configurations determined by the relative crystal axes and magnetic anisotropy without involving the twisted interface, it will highly resemble the concept of the conventional FM MTJ.

Based on the above concerns and comments, the novelty of observations and the evidence provided for the main claims in this study are not enough for the broad readership and high quality of Nature. Therefore, I cannot recommend the publication.

Referee #3 (Remarks to the Author):

The authors fabricated “twist-induced all-antiferromagnetic tunnel junction” consisting of two bilayers of 2-dimensional(2D) antiferromagnetic (AFM) CrSBr, in which in-plane crystallographic axes of the bilayers are twisted by about $10^\circ - 90^\circ$. They observed magnetoresistance (MR) ratios $> 700\%$ at 2K and non-volatile memory function(bistable magnetic alignments) at zero magnetic field. The study is well conducted, and the methods used are appropriate. The topic “antiferromagnetic tunnel junction” is timely and of interest to researchers of the field. The experimental results are basically convincing, except for some points described below. The manuscript may be recommended for publication in Nature if the following points are correctly revised or clarified.

(General Comments)

Comment 1:

Latter half of the manuscript (related to Figs.3-4 and Extended Data Figs.7-14) is not well organized and therefore is difficult to follow. I had to read back and forth several times to understand the contents. I am sure general readers of Nature will have even more difficulty.

Comment 2:

The main novelty of this manuscript with respect to previous works (Refs.18-23, 26-28) is the “twisted” 2D-AFM. The methodology of crystallographically twisted 2D material is interesting from technical point of view. However, the impact of the work on technology and engineering is limited because the material and device structure used in this manuscript is not useful for practical applications. Therefore, I feel that the novelty of the manuscript is not sufficient for Nature journal. To warrant the publication in Nature, the authors should clearly describe the impact of the work on basic science by emphasizing interesting physics of the twisted 2D-AFM tunnel junctions. This point is related to my comment 6.

(Technical Comments)

Comment 3:

Discussions on the magneto-transport properties (magnetization process) of the twisted 2D-AFM tunnel junctions are just qualitative and speculative. The speculative interpretation of the magnetization process is another reason why the latter half of the manuscript is difficult to follow. I carefully read the

qualitative explanations in the manuscript. But I still cannot understand why relatively robust bistable states at zero field appear in the vicinity of $\phi = 65^\circ$ and 130° , which are not exactly the easy axis nor hard axis. Moreover, it is not explicitly mentioned in the manuscript that bistable states appear at around 140° , 175° , 240° , 300° and 360° . Figs. 3c-d are very complex.

The magnetization process should be relatively simple with only a few parameters (in-plane uniaxial magnetic anisotropy, magnetic moment of each monolayer, and small interlayer exchange coupling). Therefore, the magnetization process should be reproduced by using the Stoner-Wohlfarth model or micro-magnetic simulation. Have the authors tried to reproduce the magnetization process related to Figs. 2d-e, 3c-e, 4c-e by using such simulations with appropriate magnetic parameters?

Comment 4:

The magnetic moment of all CrSBr layers should be aligned exactly parallel to the magnetic applied field H when $\mu_0 H$ is higher than about 1 T, resulting in angle(ϕ) - independent current (i.e., conductivity). Such ϕ - independent current is observed in Figs.2d-e and Extended Data Figs.3a-b, 4b-e. However, ϕ -dependence of current is seen in Figs. 5a-e, 7a-b, 10a-b, 11b-e, even for $H > 1$ T. The authors should explain the origin of the ϕ - dependence of conductivity at high magnetic fields.

Comment 5:

Previous works (Refs.18, 19, 21, 22, 26, 27, 28) report some data at higher temperatures. In this manuscript, on the other hand, basically all the experimental results are the data at 2K. I am especially curious about the ϕ - dependence of magneto-transport at higher temperatures, especially above T_N and T_c .

Comment 6:

The van der Waals gap between 2D CrSBr layers acts as the tunnel barrier. If the tunnel barrier simply behaves as a vacuum potential, the physics is not very interesting. But there may be more than that. For example, coupling of the tunneling states across the van der Waals gap may sensitively depend on the twist angle. Regarding the temperature dependence of conductance G above T_N , G monotonically increases with temperature in Figs. 2c (without twist), 4b (with twist), Extended Data 4a (without twist). In Fig. 3b (with twist), on the other hand, G decreases just above T_N . What is the origin of the different behaviours?

(Minor Comments)

Comment 7:

The term "all-antiferromagnetic tunnel junction" in the title is somewhat overstatement because the sample for Fig.4, which shows the best nonvolatility at zero field, is not antiferromagnetic.

Comment 8:

The optical images in Fig.2b, Extended Data Figs. 3b(inset), 11a should be explained more in detail. Only the contour of CrSBr flake is indicated. The authors should indicate where are hBN, graphite and Ti-Au electrodes.

Comment 9:

In the first paragraph of page 6, the authors state

“It should be highlighted that the ZF-TMR ratio in our twisted all-AFM MTJs is far beyond the highest TMR of ~138% at 10 K reported to date in non-collinear AFM MTJs[ref.2]” and even approaches the recent record of 1143% for a conventional FM MTJ at 10 K[ref.37].”

This is not current because the highest MR ratio of MTJs at low temperature is 2610% at 4.2K (see K. Moges et al., Physical Review B 93, 134403 (2016)). Ref.37 reports the highest MR ratio at room temperature (630%) but not at low temperature.

Author Rebuttals to Initial Comments:

Response to Referees' comments

Referees' comments in black; our response in blue.

Referee #1 (Remarks to the Author):

In this manuscript, Chen et al. introduce a pioneering approach to constructing all-antiferromagnetic tunnel junctions at the atomic level by employing a twisting strategy. This work focuses on the utilization of two-dimensional antiferromagnet, CrSBr bilayers. Remarkably, a significant nonvolatile tunneling magnetoresistance ratio exceeding 700% is achieved at zero field, with the van der Waals gap functioning as the tunnel barrier. The results strongly suggest the presence of a spin-filtering effect at the twisted interface, an observation further bolstered by experiments involving twisted CrSBr monolayers. This investigation holds immense promise for the advancement of nonvolatile magnetic information storage at the atomic dimension.

--We sincerely thank the referee for the insightful comments and summary of our work. We are particularly encouraged by the referee's remarks that *"This investigation holds immense promise for the advancement of nonvolatile magnetic information storage at the atomic dimension."*

Despite the inherent merits of the work, there are some queries and points of contention that ought to be addressed:

--In the following, we address the referee's comments point-by-point.

1) The introduction of the twisting way for constructing antiferromagnetic tunnel junctions is lacked. It would be beneficial if the authors can elucidate more on the underlying principles and mechanisms governing this strategy, especially in comparison to traditional ways.

--Thanks for the good suggestion. The zero field nonvolatility is ascribed to the spin-filtering effect at the twisted interface, which highly resembles the concept of traditional FM MTJs within a completely different structure. See Fig. R1a,b, conventional MTJs adopt additional AFM layers to produce pinning effects and an MgO layer as the tunneling barrier. Our twist structures are much easier (Fig. R1c). In our twisted structure, the extra AFM is removed, and the pinning effects inherently exist owing to magnetic anisotropy. In addition, the additional MgO tunneling barrier is also removed, and its role is played by the

van der Waals gap, which also inherently exists in the twisted structure. Thereby, differing from the bottom-up multiple-layer deposition recipes in conventional MTJs, the recipes for constructing the twisted AFM MTJs are relatively simple. Picking up each component layer by layer can obtain the final devices. Please see the fabrication details in the Methods of the revised manuscript.

There is a principle about twist angles we would like to present, which is vital for practical applications. We observed zero field nonvolatility over an extensive range of twist angles $10^\circ - 80^\circ$, meaning the twist angle is valuable for tuning device functions. Furthermore, we noticed a twist angle-dependent trade-off effect between zero field TMR ratio and pinning strength, please see Fig. R1d-e. In short, a smaller twist angle (θ_{twist}) favors a higher ZF-TMR ratio (Fig. R1e) and lower pinning strength (Fig. R1d), and vice versa. Please see the detailed discussion in the extended notes 1 and 3. The needed twist angles for constructing antiferromagnetic tunnel junctions can be flexibly chosen according to practical applications.

Fig. R1 | **a**, a conventional FM MTJ. **b**, By analogy to **a**, an all-antiferromagnetic MTJ based on non-collinear AFMs. **c**, Two AFM spin configurations with different magnetoresistances based on layer-dependent magnetism. Top panel: side view. Bottom panel: top view. **d**, α -dependent Δ normalized H_c for quantifying the pinning strength, in which $\alpha = 0^\circ$ (90°) means field is along the b-axis (a-axis) of the reference CrSBr flake. **e**, Comparison between the calculated TMR ratios and the experimental values as the function of twist angle (θ_{twist}). **a-c** are from Fig. 1 of the revised main text. **d-e** are from Fig. S1 of the extended notes.

2) The manuscript claims an impressive nonvolatile tunneling magnetoresistance ratio with CrSBr bilayers. How does this result compare with other known antiferromagnetic materials? Are there specific advantages to using CrSBr for this purpose?

--Thanks for the question about the experimental object. First, we compare our twisted AFM MTJs with the recently reported conventional AFM MTJs. In a 10° twisted CrSBr all-AFM MTJ, we obtained a TMR of more than 700% at zero field. This year, two groups reported all-AFM MTJs constructed with noncollinear AFMs of Mn_3Pt and Mn_3Sn adopting the classic structure of FM MTJs as shown in Fig. R1b (ref. 1 of the revised main text, *Nature* 613, 490-495 (2023); ref. 2 of the revised main text, *Nature* 613, 485-489 (2023)). The highest TMR of $\sim 138\%$ at 10 K is reported, far below our TMR.

Compared with other 2D materials, there is no outstanding advantage if only regarding the TMR value. For example, CrI_3 bilayer shows 530% TMR at 2 K and 9 T (ref. 20 of the revised main text, *Science* 360, 1214-1218 (2018)). Our work's advantage is that the TMR is nonvolatile at zero field, which is the key to practical application. To realize zero field nonvolatility, a van der Waals material has to meet our three requirements: (a) semiconductor or insulator; (b) interlayer AFM coupling and intralayer FM coupling; (c) strong uniaxial in-plane magnetic anisotropy. We chose CrSBr because it is the exclusive 2D magnet that meets all the requirements as far as we know.

3) The van der Waals gap is highlighted as the tunnel barrier in this study. Could the authors provide further insight into its role and significance? How does it contribute to the observed tunneling magnetoresistance ratio, and what advantages does it offer over other potential tunnel barriers?

--We sincerely appreciate the referee for raising the importance of the van der Waals gap. Yes, it plays a very crucial role in our twisted structures. (a) CrSBr is a van der Waals material with no chemical bonds in the van der Waals gaps. We can exfoliate CrSBr bulk down to monolayer and bilayer, allowing us to fabricate MTJ in the atomic limit. (b) To manufacture MTJs, a tunneling barrier in analogy to MgO in conventional MTJs is necessary. The van der Waals gap as a vacuum dielectric layer plays the role of tunneling barrier. (c) In conventional MTJs, besides the role of tunneling barrier, MgO separating two FM metal layers helps the magnetization of each FM metal layer freely flip without mutual interaction, which gives rise to binary tunneling states at zero field due to parallel ($\uparrow\uparrow$, $\downarrow\downarrow$) or antiparallel ($\uparrow\downarrow$, $\downarrow\uparrow$) spin configurations. Similarly, the van der Waals gap in the twisted interface breaks the inherent interlayer AFM coupling (i.e., decoupling) that originally resides in the untwisted CrSBr interface, making it possible to realize quasi-parallel and quasi-antiparallel configurations at zero field. The decoupling will be discussed later.

The above discussion clarifies that the van der Waals gap plays functions similar to MgO in conventional MTJs. MgO is an actual substance, but rather, the van der Waals gap is "nothing", which inherently exists in all twisted CrSBr MTJs regardless of the twist angles. This merit means we can fabricate any MTJ with any twist angle we want. This is an exotic property superior to MgO.

Moreover, the van der Waals gap likely contributes a higher TMR ratio than actual substances as the tunneling barrier. To prove this further, we performed control experiments on a 30° twisted CrSBr bilayer/hBN monolayer/CrSBr bilayer MTJ (Fig. R2). In this device, the inserted hBN monolayer mainly plays the role of tunneling barrier. A zero field nonvolatility is also observed. However, the TMR ratio at zero field is $\sim 45\%$ in this control twisted MTJ, see Fig. R2g, whose value is substantially lower than the

TMR ratio of $>200\%$ on the 35° twisted MTJ (Fig. 3f of the revised main text) without sandwiching an hBN layer. The reason is ascribed to the scattering-induced spin depolarization by the hBN layer, which proves that the van der Waals gap at the twisted interface is superior to actual substances as a tunneling barrier in the twisted CrSBr MTJs. Similar results were also found in another control sample of 45° twisted CrSBr monolayer/hBN monolayer/CrSBr monolayer MTJ (Fig. R3). These control experimental results have been added as Figs. S4-5 in the extended notes.

Fig. R2 | Electrical transport results for a 30° twisted CrSBr bilayer/hBN monolayer/CrSBr bilayer MTJ. a-b, Field orientation dependence of the tunneling current in the ab plane. Two blue arrows indicate the sweeping direction of field. ± 1.5 T field is used. **c-d**, Same as **a**, **b** except for sweeping field between ± 0.3 T. **e**, Representatives of the two groups of ZF NV related to the easy axis ($\Phi = 135^\circ$) and hard axis ($\Phi = 95^\circ$), respectively. **f**, I - V curves at ZF and 9 T. **g**, Extracted ZF-TMR ratio as a function of bias based on the ZF I - V curves in **f**. A constant DC bias of 20 mV is applied in **a-e**. **a-g** are from Fig. S4 of the extended notes.

Fig. R3 | Electrical transport results for a 45° twisted CrSBr monolayer/hBN monolayer/CrSBr monolayer MTJ. a-b, Field orientation dependence of the tunneling current in the *ab* plane. Two blue arrows indicate the sweeping direction of field. ± 1.5 T field is used. **c-d,** Same as **a, b** except for sweeping field between ± 0.3 T. **e,** Tunneling current versus field with field oriented along the easy axis of the top CrSBr flake. The inset shows a close-up of ZF adjacency. ± 9 T field is used. **f,** Demonstration of ZF nonvolatility related to the hard axis. ± 0.3 T field is used. **g,** Demonstration of ZF nonvolatility related to the easy axis. ± 0.15 T field is used. **h,** ZF-TMR. A constant DC bias of 25 mV is applied in **a-g**. **a-h** are from Fig. S5 of the extended notes.

4) The observed spin-filtering effect at the twisted interface is intriguing. It would be valuable if the authors could delve deeper into its origins and implications (using Density Functional Theory calculations to further investigate the twisted/non-twisted interfaces' electronic properties would be one of the useful means). How does this effect influence the overall performance of the all-antiferromagnetic tunnel junctions? Moreover, are there conditions or factors that might enhance or mitigate this effect?

--Thanks for these excellent suggestions. We conducted first-principles calculations and Stoner-Wohlfarth model calculations to delve into the spin-filtering effect at the twisted interface in the revised manuscript.

In Table R1 (where $J_1 - J_7$ represent the intralayer exchange interactions and J_{z1}, J_{z2} represent the interlayer exchange interactions. Please see details in Part A of Extended note 2), for the intrinsic CrSBr bilayer and bulk, the first-principles calculations show superexchange interaction gives rise to an interlayer AFM coupling at the untwisted interface. The first-principles calculations also indicate an intralayer FM coupling in each CrSBr monolayer mediated by superexchange interaction. These results nicely explain the experimental observations of Fig. 2 in the revised manuscript. At low field, the spin configuration is

antiparallel in the intrinsic CrSBr bilayer, resulting in a low tunneling current. When a strong field forces the spin configuration parallel, the tunneling current increases. However, because of the interlayer AFM coupling, the spin configurations must be antiparallel at zero field, either $\vec{\uparrow}\downarrow$ or $\vec{\downarrow}\uparrow$, resulting in a single tunneling state, namely volatility at zero field.

Table R1. Calculated exchange parameters (in meV) in vacuum for CrSBr bulk, monolayer (ML), bilayer (BL) and twisted bilayers (TBL) with twist angles of 45° and 90° . $J_1 - J_7$ represent the intralayer exchange interactions and J_{z1} and J_{z2} represent the interlayer exchange interactions. A copy of Table S1 in the extended notes.

J_i	bulk	ML	BL	TBL (45°)	TBL (90°)
J_1	2.59	3.01	2.74	2.83	2.91
J_2	6.13	5.83	6.74	6.52	6.43
J_3	1.50	1.93	1.82	1.71	1.67
J_4	0.03	-0.07	0.17	0.11	0.19
J_5	0.05	-0.01	-0.02	0.01	-0.01
J_6	-1.19	-0.89	-1.23	-1.13	-1.37
J_7	0.96	1.23	0.83	1.01	0.94
J_{z1}	-0.25		-0.31	-0.005	-0.0003
J_{z2}	0.01		-0.01	-0.002	0.0001

In contrast, the first-principles calculations show negligible magnetic coupling at the twisted interface, but the intralayer FM coupling is maintained. This means the two stacked CrSBr flakes are decoupled via a twist operation. Thus, parallel ($\vec{\uparrow}\uparrow$, $\vec{\downarrow}\downarrow$) or antiparallel ($\vec{\uparrow}\downarrow$, $\vec{\downarrow}\uparrow$) spin configurations at the twisted interface are accessible at zero field, giving rise to binary tunneling states, namely nonvolatility at zero field. We note that the parallel and antiparallel spin configurations are not perfect but instead quasi-parallel and quasi-antiparallel owing to the twist angles. For example, the relative angle (θ) of the two magnetizations of the untwisted CrSBr bilayer with a perfectly parallel (antiparallel) configuration is 0° (180°). For the 10° twisted MTJ, the relative angle with a quasi-parallel (quasi-antiparallel) configuration is 10° (170°).

Fig. R4 | a-b, Calculated tunneling conductance in a dual sweeping external magnetic field (h) based on the Stoner-Wohlfarth model, which are from Fig. S8 of the extended notes.

Twisting has two functions: (1) producing a pinning effect. (2) destroying the original interlayer AFM coupling, which makes it possible to use an external field to switch the spin configuration between quasi-antiparallel and quasi-parallel. The spin-filtering effect depends on the relative angle (θ) (ref. 8 of the revised main text, *Phys Rev B Condens Matter* 39, 6995-7002 (1989)), which further determines the tunneling conductance. We combined the Stoner-Wohlfarth model to calculate the tunneling conductance, see Fig. R4 and the details in Extended note 3. The calculated results reproduce the experimental observations well. Including zero field nonvolatility and time-reversal symmetry, the sloped relationship between tunnel currents and external field is also captured. Furthermore, the experimental results and theoretical calculations here both indicate a smaller twist angle can enhance the spin-filtering effect at the interface, resulting in a larger TMR ratio. But there is a trade-off between the spin-filtering effect and the pinning strength as discussed in responding to Comment 1.

5) In pushing the boundaries to achieve atomic-level storage, there could be potential challenges and limitations. It would be informative if the authors can address any technical or theoretical hurdles faced during their study or those they anticipate in future applications of this technology.

--Thanks for reminding us to consider our work's limitations further.

In the revised manuscript, we further confirm that it is a spin-filtering mechanism governing the observed zero field nonvolatility. Thus, we think no theoretical difficulties are hindering our twisted MTJs from practical applications. However, there are some practical challenges from materials and industrial techniques. CrSBr is the only 2D magnet to date that fits the requirements discussed in response to Comment 2 to produce zero field nonvolatility, but it works in low temperatures. The good news is that our high-temperature experiments indicated the nonvolatility at zero field is robust close to the Néel temperature of CrSBr. Once the temperature is higher than the Néel temperature, nonvolatility at zero field disappears (Extended Data Fig. 12 of the revised manuscript). Because 2D magnetism is an emergent field proposed in 2017 (ref. 14 and ref. 15 of the revised main text, *Nature* 546, 270-273 (2017), *Nature*

546, 265-269 (2017)), more abundant 2D magnets are waiting to be discovered. A room-temperature atomic MTJ will hopefully be developed in the future with a high phase-transition temperature 2D magnet. On the other hand, because we use tape-based exfoliations, the yield of CrSBr monolayer and bilayer is far below the industrial demand. Manual transfer techniques are time-consuming and can not meet industrial demand. Therefore, developing efficient fabrication techniques is desirable.

This manuscript holds considerable significance in the realm of antiferromagnetic spintronics. It offers a fresh perspective and promises groundbreaking advancements in the domain of high-density, ultrafast information devices. I am inclined to endorse this work for publication in Nature, provided the points raised above are adequately addressed and clarified.

--We highly appreciate the referee for the high evaluation and recommendation publication.

Referee #2 (Remarks to the Author):

The authors report a tunneling magnetoresistance study of twisted 2D antiferromagnet CrSBr. A careful in-plane magnetic field orientation dependence on the tunneling current is measured in twisted CrSBr double bilayers as well as double monolayers. Two groups of magnetic states are identified as the magnetic field is swept up and down. The observation is attributed to a twisted interface mechanism, and zero field nonvolatility is claimed. The results are interesting, and the data quality is high, but several concerns and questions are listed below.

--We sincerely thank the referee for the insightful comments and summary of our work. We are particularly encouraged by the referee's remarks that "The results are interesting, and the data quality is high". We achieved bistable states at zero field in twisted double CrSBr bilayers as well as double monolayers because of a spin-filtering effect at the twisted interface.

After reading the referee's comments carefully, we realized that we used some ambiguous descriptions in the previous manuscript, especially using "quasi-AFM coupling" and "quasi-FM coupling" to assign the bistable states at zero field, which likely brought the referee to consider interlayer magnetic coupling mediated by exchange interactions. We sincerely apologize for those ambiguous descriptions, which have been corrected in the revised manuscript.

We are impressed and grateful that the referee was able to capture the essence of our work that our work highly resembles the concept of the conventional FM MTJ with a novel twisted structure in the atomic limit, which has giant significance for the conventional MTJ domain, which will be discussed in the responding to Comment 4. Our twisted structure is unique via using large twist angles from 10° to 90° . The mechanism is a spin-filtering effect at the twisted interface instead of coupling mediated by moiré potential in other twisted systems with small twist angles (usually less than 10°). Here, we present two comments before more detailed discussions.

(1) The possibility of observing zero field nonvolatility from 10° - 80° is very low if there is a significant interlayer coupling governed by exchange interactions or moiré potential at the twisted interface. Our twist angles are large, which suppresses the formation of moiré superlattices well. On the other hand, to produce meaningful moiré superlattices, the twist angles usually need to be precisely tuned. For example, in the magic-angle twisted graphene system, once the critical angle of 1.1° is missed, superconductivity disappears (ref. 32 of the revised main text, *Nature* 556, 43-50 (2018)). However, we observed zero field nonvolatility over a large twist angle range of 10° to 80° .

(2) The bistable states at zero field support no or negligible interlayer coupling at the twisted interface. First, let us consider the intrinsic or untwisted CrSBr bilayer. When a magnetic field is swept up and down, and then back to zero, one of the antiparallel magnetizations configuration, either $\vec{\uparrow}\downarrow$ or $\vec{\downarrow}\uparrow$, has to return because of the interlayer AFM exchange interaction. The antiparallel magnetization configurations result in a single tunneling state, namely volatility, which has been verified by our experiments and refs. 18-22 of the revised main text. Therefore, assuming there is a similar exchange interaction at the twisted

interface, one between “quasi-AFM coupling” and “quasi-FM coupling” should be more favored at zero field instead of both as the experimental observations, namely a single tunneling state rather than bistable states. We agree with the referee’s opinion that these bistable states should be assigned “quasi-antiparallel configuration” and “quasi-parallel configuration”, which have been used to replace “quasi-AFM coupling” and “quasi-FM coupling” in the revised manuscript.

Below, we will further explain our work’s mechanism using the theoretical and experimental results.

(1) The main discovery is attributed to the twisted interface, and this is certainly a popular topic recently. However, a solid and conclusive explanation of the twisted interface mechanism is missing in this study. Although the twisted interface can potentially tune the interlayer magnetic coupling, the dominating effect is still the intrinsic magnetic anisotropy in individual layers of CrSBr. Based on the tunneling current, the authors try to assign “quasi-AFM coupling” and “quasi-FM coupling”, however, this data can only help assign a spin configuration, which is mainly determined by magnetic anisotropy and crystal axes here, but cannot identify the interlayer magnetic coupling. The direct evidence for the twisting effect on the interlayer magnetic coupling is missing.

--Thanks a lot for the constructive comments, which encouraged us to consider the role of the twisted interface further. We agree with “this data can only help assign a spin configuration, which is mainly determined by magnetic anisotropy and crystal axes here, but cannot identify the interlayer magnetic coupling.”. Based on the beginning discussions and subsequent discussions in responding to other comments, the referee will find that magnetic coupling at the twisted interface cannot support our experimental observations. However, a spin-filtering effect resembling the mechanism of conventional FM MTJs can explain our results nicely. We are sorry for using the ambiguous “quasi-AFM coupling” and “quasi-FM coupling” to assign the bistable states at zero in our previous manuscript. We have corrected them and other ambiguous texts in the revised manuscript.

[REDACTED]

We also agree that “the dominating effect is still the intrinsic magnetic anisotropy in individual layers of CrSBr” and further highlight the maintained in-plane uniaxial magnetic anisotropy surviving from twist operations is very important to produce pinning effects. The pinning effect is one of the necessary preconditions for an MTJ to work functionally. If there is no pinning, for example, in a conventional FM MTJ, nonvolatility cannot be produced at zero field (Fig. R1a) because sweeping external magnetic field produces the same influence on the magnetizations of both magnetic layers.

In conventional FM MTJs, the pinning effects are usually from exchange bias produced by an additional AFM pinning layer, see Fig. R1b. The pinning AFM layers usually adopt synthetic antiferromagnets (SAFs), which are composed of 3 layers at least (Fig. R1c). In contrast, our twisting structure inherently possesses a pinning effect arising from the strong magnetic anisotropy of CrSBr without needing the usual AFM pinning layers (Fig. R1d). A detailed discussion about pinning effects in the twisted CrSBr structures is available in Extended note 1.

(2) There are also no theoretical calculations or simulations to support the twisted interface mechanism and the assigned “quasi-FM/FM coupling”, which will involve a careful study of the local layer stacking and interlayer magnetic interactions. Moreover, in all the magnetic field orientation dependence data, multiple transitions and kinks have been measured and shown in the colormap. However, no magnetic states or spin configurations are assigned. The authors should add theoretical calculations and simulations to identify, support, and label these magnetic states.

--Thanks for emphasizing the importance of theoretical calculations. We have carried out first-principles calculations. We confirmed that it is a superexchange interaction resulting in FM coupling in the CrSBr intralayer and AFM coupling in the CrSBr interlayer. The results also indicate negligible magnetic coupling at the twisted interface.

Table R1. Calculated exchange parameters (in meV) in vacuum for CrSBr bulk, monolayer (ML), bilayer (BL) and twisted bilayers (TBL) with twist angles of 45° and 90°. A copy of Table S1 of the extended notes.

J_i	bulk	ML	BL	TBL (45°)	TBL (90°)
J_1	2.59	3.01	2.74	2.83	2.91
J_2	6.13	5.83	6.74	6.52	6.43
J_3	1.50	1.93	1.82	1.71	1.67
J_4	0.03	-0.07	0.17	0.11	0.19
J_5	0.05	-0.01	-0.02	0.01	-0.01
J_6	-1.19	-0.89	-1.23	-1.13	-1.37
J_7	0.96	1.23	0.83	1.01	0.94
J_{z1}	-0.25		-0.31	-0.005	-0.0003
J_{z2}	0.01		-0.01	-0.002	0.0001

In detail, in Table. R1, J_1 to J_7 represent the intralayer exchange interactions, and J_{z1} , J_{z2} represent the interlayer exchange interactions. The approximate $J_1 - J_7$ in all the cases of Table. R1 indicates the intralayer FM coupling is robust regardless of the twist operations because the intralayer atom arrangements are the same. AFM coupling is obtained at the untwisted interface (see the J_{z1} and J_{z2} of bulk and bilayer), but is suppressed at the twisted interface (see the J_{z1} and J_{z2} of 45° -twisted and 90° -twisted cases) because the interlayer atom arrangements are changed. Therefore, the observed bistable states at zero field should be assigned to “quasi-antiparallel configuration” and “quasi-parallel configuration” rather than magnetic coupling. Please see more details of the first-principles calculations in Part A of Extended note 2.

The multiple transitions and kinks in the colormaps at finite fields derive from magnetic domains in the tunneling junction area since they always show randomness in multiple sweeping loops, for example, see Fig. R2. If their origin is magnetic coupling, the multiple transitions and kinks should prefer stable appearance instead of randomness. There are two main reasons for forming magnetic domains at finite fields. (1) Internal factors, multiple mechanisms’ competition, such as external field, magnetocrystalline anisotropy, demagnetization energy, and thermal fluctuation. (2) External factors, asymmetries introduced via device fabrications, for example, local strains and defects. Comparing Fig. R2a with Fig. R2b, ideally, their shape should be equal regarding the symmetry of the twisted structure but rather are different, in fact, owing to such external factors. In the revised manuscript, we have included texts about the multiple transitions and kinks in Part C of Extended note 2.

Fig. R2 | Tiny plateaus and kinks at finite fields in multiple loops of sweeping field. a-b, The same data as Extended Data Fig. 14 of the revised manuscript measured from the 55° twisted monolayer/monolayer CrSBr MTJ. The arrows mark the tiny plateaus and kinks.

(3) Several orientation data look just like an overlapping pattern of the two individual bilayers, which seems to suggest that the two bilayers may be completely magnetic decoupled, for example, in the Extended Data Fig. 16. In other words, it can be just a result of two separate groups of spin configurations, which are controlled by the external magnetic field, magnetic anisotropy, and their crystal axes (the relative twist angle), but not the twisted interface. It will be worth a careful examination of the role of the twisted interface, which only induces the relative twist angle, or also leads to an exotic interlayer magnetic coupling. This question can be potentially answered by adding a thin layer of BN between the two CrSBr layers as a control experiment, where magnetic coupling will be suppressed.

--Thanks for the careful observation of the experimental results. Indeed, several orientation data look like an overlapping pattern of the two individual bilayers, especially Extended Data Fig. 16 of the previous manuscript measured on a 90° CrSBr twisted bilayer/bilayer MTJ. The overlapping pattern can also be found in Extended Data Fig. 7 of the revised manuscript, a 35° twisted CrSBr bilayer/bilayer MTJ, though the pattern is more obscure owing to the smaller relative twist angle. The overlapping pattern is mainly from the two untwisted interfaces of the two individual CrSBr bilayers (Fig. R3), which indicates that the magnetic anisotropy of the CrSBr is maintained in the twisted MTJs and each magnetization of the two individual CrSBr bilayer evolves with an external magnetic field independently, which is the key to producing pinning effects as discussed in response to Comment 2. Moreover, the overlapping pattern also suggests that the interlayer AFM coupling is maintained within each CrSBr bilayer, so the always antiparallel spin arrangements at the two untwisted interfaces do not contribute to the zero field nonvolatility. In addition to the two untwisted interfaces in the 4-layer structure of twisted CrSBr bilayer/bilayer, the third interface is twisted (Fig. R3), and a spin-filtering effect at this interface is responsible for the observed zero field nonvolatility, which will be discussed in detail in response to Comment 4.

Fig. R3 | Schematic of the twisted 4-layer structure of double CrSB bilayers. There are three interfaces. Two of them are untwisted, and the third one is twisted.

More convincingly, we carried out the referee's excellent suggestion. The control experiments on a 30° twisted CrSB bilayer/hBN monolayer/CrSB bilayer MTJ also show an overlapping pattern of the two individual bilayers and a zero field nonvolatility (Fig. R4), which further confirms the bistable states at zero field resulting from a spin-filter effecting instead of magnetic coupling at the twisted interface. Similar results were also found in another control sample of 45° twisted CrSB monolayer/hBN monolayer/CrSB monolayer MTJ (Fig. R5). These control experimental results have been added in the revised manuscript as Figs. S4 and 5 of extended notes.

Fig. R4 | Electrical transport results for a 30° twisted CrSB bilayer/hBN monolayer/CrSB bilayer MTJ. a-b, Field orientation dependence of the tunneling current in the ab plane. Two blue arrows indicate the sweeping direction

of field. ± 1.5 T field is used. **c-d**, Same as **a**, **b** except for sweeping field between ± 0.3 T. **e**, Representatives of the two groups of ZF NV related to the easy axis ($\Phi = 135^\circ$) and hard axis ($\Phi = 95^\circ$), respectively. **f**, I - V curves at ZF and 9 T. **g**, Extracted ZF-TMR ratio as a function of bias based on the ZF I - V curves in **f**. A constant DC bias of 20 mV is applied in **a-e**. **a-g** are from Fig. S4 of the extended notes.

Fig. R5 | Electrical transport results for a 45° twisted CrSBr monolayer/hBN monolayer/CrSBr monolayer MTJ. a-b, Field orientation dependence of the tunneling current in the ab plane. Two blue arrows indicate the sweeping direction of field. ± 1.5 T field is used. **c-d**, Same as **a**, **b** except for sweeping field between ± 0.3 T. **e**, Tunneling current versus field with field oriented along the easy axis of the top CrSBr flake. The inset shows a close-up of ZF adjacency. ± 9 T field is used. **f**, Demonstration of ZF nonvolatility related to the hard axis. ± 0.3 T field is used. **g**, Demonstration of ZF nonvolatility related to the easy axis. ± 0.15 T field is used. **h**, ZF-TMR. A constant DC bias of 25 mV is applied in **a-g**. **a-h** are from Fig. S5 of the extended notes.

(4) If the current experimental results can indeed be explained based on just the spin configurations determined by the relative crystal axes and magnetic anisotropy without involving the twisted interface, it will highly resemble the concept of the conventional FM MTJ.

--We highly appreciate the referee for the condensed conclusion, yes, our work highly resembles the concept of the conventional FM MTJ, and indeed, our work can be nicely explained by the spin-filter effect **involving the twisted interface**, where the spin alignments are determined by the relative crystal axes and magnetic anisotropy. To further demonstrate the concept, we carried out Stoner-Wohlfarth model

calculations. The basic idea is based on the discussion in response to Comment 3, “the magnetic anisotropy of the CrSBr is maintained in the twisted MTJs and each magnetization of the two individual CrSBr bilayer evolves with an external magnetic field independently”, which means it is possible to calculate the relative angle between the two magnetizations at the twisted interface. Because the relative angle determines the transmission of the spin-filtering effect (ref. 8 of the revised main text, *Phys Rev B Condens Matter* 39, 6995-7002 (1989)), which further determines the tunneling conductance (G), we then obtained the results of Fig. R6, please see the details in Extended note 3. The calculated results reproduce the experimental observations well. Including zero field nonvolatility and time-reversal symmetry, the sloped relationship between tunnel currents and external field is also captured.

Fig. R6 | a-b, Calculated tunneling conductance in a dual sweeping external magnetic field (h) based on the Stoner-Wohlfarth model, which are from Fig. S8 of the extended notes.

We note that our work realized atomic MTJs based on the basic concept of conventional MTJs with a completely different structure. Our work offers new perspectives in the conventional MTJ domain, especially in developing all-AFM MTJs.

(1) Our work is the first MTJ that demonstrated zero field nonvolatility down to the atomic limit. Atomic MTJs are one of the ultimate goals in spintronics to realize ultrahigh-density information storage.

(2) The twisted structure is unprecedented in the conventional MTJ domain. Conventional MTJs adopt additional AFM layers to produce pinning effects and an MgO layer as the tunneling barrier. In our twisted structure, the extra AFM is removed, and the pinning effects inherently exist owing to magnetic anisotropy. In addition, the extra MgO tunneling barrier is removed, and its role is played by the van der Waals gap, which also inherently exists in the twisted structure. The classic MTJ structure was proposed more than 30 years (e.g., ref. 8 of the revised main text, *Phys Rev B Condens Matter* 39, 6995-7002 (1989)). Our work introduces new inspirations into the conventional MTJ domain.

(3) Realizing ultrahigh-density information storage also faces a physical constraint, that is, the stray field problem. Conventional MTJs are formed with FM layers, which inevitably produce stray fields. If FM MTJs

are placed tightly, the stray field of each FM MTJ will strongly influence each other. To overcome this physical constraint, developing all-AFM MTJs without any FM component is a frontier in the domain of conventional MTJs. This year, two groups reported all-AFM MTJs constructed with noncollinear AFMs based on the classic structure of FM MTJs as shown in Figs. 1a,b of the revised main text (ref. 1 of the revised main text, *Nature* 613, 490-495 (2023); ref. 2 of the revised main text, *Nature* 613, 485-489 (2023)). The highest TMR of ~138% at 10 K is reported. Note that the noncollinear AFMs still have low stray field. Our work reports atomic all-AFM MTJs with TMR up to 700%, not only in the spatial dimensions but also in the physical properties, showing superiority over conventional MTJs for developing ultrahigh-density information storage. On the other hand, because AFM materials inherently possess ultrafast terahertz (THz) spin dynamics, our twisted MTJs are also promising for ultrafast information devices.

(4) In the revised manuscript, we confirm that it is a spin-filtering mechanism governing the observed zero field nonvolatility instead of a particular magnetic coupling mechanism exclusively existing in CrSBr system. We chose CrSBr because it is the only 2D magnet that meets our requirements as far as we know. Our requirements are (a) semiconductor or insulator, (b) interlayer AFM coupling and intralayer FM coupling, (c) strong uniaxial in-plane magnetic anisotropy. Moreover, in the revised manuscript, we demonstrated that the nonvolatility at zero field is robust close to the Néel temperature (Fig. R7). Once the temperature is higher than the Néel temperature, nonvolatility at zero field disappears. Because 2D magnetism is an emergent field first reported in 2017 (ref. 14 and ref. 15 of the revised main text, *Nature* 546, 270-273 (2017), *Nature* 546, 265-269 (2017)), more abundant 2D magnets are waiting to be discovered. A room-temperature atomic MTJ will hopefully be developed in the future if a 2D magnet meets the above requirements and has a high phase-transition temperature. Thus, we firmly believe our twisted MTJs are very promising for practical applications. We also note that cryo-electronics is a very exciting technological field that has become of great interest just recently, motivated, in part, by the intense research efforts to explore quantum computing systems that operate at cryogenic temperatures (*IEEE Microwave Magazine* 22, 60-78 (2021)), and, at the same time, by several recent discoveries in proximity induced superconductivity such as, for example the superconducting and Josephson diode effects (*Nature Physics* 19, 1379-1380 (2023)). Cryogenic electronics requires advanced memories so that our all-AFM MTJs could be useful for such applications.

Fig. R7 | Electrical transport results of a 40° twisted bilayer/bilayer CrSBr MTJ at different temperatures. a-j, Field orientation dependence of the tunneling current in the ab plane at different temperatures. 2 K in a-b, 50 K in c-d, 100 K in e-f, 120 K in g-h, and 150 K in i-j. The two blue arrows indicate the field sweeping direction. A constant DC bias of 1 mV is applied in all measurements. k, ZF-TMR ratio with 1 mV DC bias at different temperatures. a-k are from Extended Data Fig. 12 of the revised manuscript.

Based on the above concerns and comments, the novelty of observations and the evidence provided for the main claims in this study are not enough for the broad readership and high quality of Nature. Therefore, I cannot recommend the publication.

--Thanks for mentioning the novelty and evidence problems. We have performed additional experiments, first-principles calculations, and the Stoner-Wohlfarth model calculations in the revised manuscript to support our claims further. For the novelty problem, besides the significance in the conventional MTJ domain discussed in response to Comment 4, we want to further display our work's significance in 2D nanoelectronics, especially 2D twistrionics.

(1) Our work is the first report of zero field nonvolatile atomic MTJs. We believe it is a milestone in the atomic MTJ field towards practical applications. In 2017, two groups first reported 2D magnets, i.e., CrI_3 and $\text{Cr}_2\text{Ge}_2\text{Te}_6$ (ref. 14 and ref. 15 of the revised main text, *Nature* 546, 270-273 (2017), *Nature* 546, 265-269 (2017)). Then, several groups immediately developed MTJs down to the atomic limit from 2018 to 2019 (refs. 18-22 of the revised main text). These pioneering works have great significance because atomic MTJs are one of the ultimate goals in spintronics to realize ultrahigh-density information storage. However, after 2019, there has been no significant progress in atomic MTJs. One of the important reasons is that all previous atomic MTJs are volatile at zero field.

(2) Our work will extend 2D twistrionics. Twistrionics is an emergent field in 2D nanoelectronics that was energized by the reports of magic-angle twisted graphene in 2018 (refs. 31 and 32 of the revised main text, *Nature* 556, 80-84 (2018), *Nature* 556, 43-50 (2018)), after which the concept of the twist angle became popular for tuning device functions. However, from magic-angle twisted graphene with superconductivity to the recent twisted MoTe₂ with a fractional quantized anomalous Hall effect (*Nature* 622, 74-79 (2023)), most of these previous reports use small twist angles (less than 10°), because moiré superlattices usually appear at small twist angles, which is useful to further tailoring energy bands. This is the mainstream concept in the 2D twistrionics community. Because of this, twists with large twist angles are rarely explored. Nevertheless, our work focuses on large twist angles and we find zero field nonvolatility due to a spin-filtering effect at the twisted interface. We believe our work will extend 2D twistrionics by using large twist angles and new mechanisms in addition to small angles and moiré superlattices.

Referee #3 (Remarks to the Author):

The authors fabricated “twist-induced all-antiferromagnetic tunnel junction” consisting of two bilayers of 2-dimensional(2D) antiferromagnetic (AFM) CrSBr, in which in-plane crystallographic axes of the bilayers are twisted by about $10^\circ - 90^\circ$. They observed magnetoresistance (MR) ratios $> 700\%$ at 2K and non-volatile memory function(bistable magnetic alignments) at zero magnetic field. The study is well conducted, and the methods used are appropriate. The topic “antiferromagnetic tunnel junction” is timely and of interest to researchers of the field. The experimental results are basically convincing, except for some points described below. The manuscript may be recommended for publication in Nature if the following points are correctly revised or clarified.

--We sincerely thank the referee for the insightful comments and summary of our work. We are particularly encouraged by the referee’s remark that “The study is well conducted, and the methods used are appropriate.” and our work is “timely and of interest to researchers of the field”. We also appreciate the referee spending much time on our work, and we are very impressed that the referee noticed some important details that we ignored. We have carefully addressed the referee’s comments. The comments have helped us to improve the quality of our work further. Below, we respond to the referee’s comments one by one.

(General Comments)

Comment 1:

Latter half of the manuscript (related to Figs.3-4 and Extended Data Figs.7-14) is not well organized and therefore is difficult to follow. I had to read back and forth several times to understand the contents. I am sure general readers of Nature will have even more difficulty.

--We are very thankful for the referee supplying us with feedback from a reader’s viewpoint. We have reorganized the latter half of the manuscript. Specifically, we adjusted related figures by adding indicators and additional descriptions. Some ambiguous texts have been clarified in the revised manuscript. Additional experimental and theoretical results are also added in the revised manuscript. A detailed discussion follows below.

Comment 2:

The main novelty of this manuscript with respect to previous works (Refs.18-23, 26-28) is the “twisted” 2D-AFM. The methodology of crystallographically twisted 2D material is interesting from technical point of view. However, the impact of the work on technology and engineering is limited because the material and device structure used in this manuscript is not useful for practical applications. Therefore, I feel that

the novelty of the manuscript is not sufficient for Nature journal. To warrant the publication in Nature, the authors should clearly describe the impact of the work on basic science by emphasizing interesting physics of the twisted 2D-AFM tunnel junctions. This point is related to my comment 6.

--We thank the referee for his/her comments about the novelty of our work, which has made us further consider our work's potential impacts in 2D nanoelectronics, especially 2D twistrionics, and conventional MTJs, especially the development of all-AFM MTJs. We have included additional text in the revised manuscript to address these points.

(1) Our work is the first report of zero field nonvolatile atomic MTJs. We believe it is a milestone in the atomic MTJ field towards practical applications. In 2017, two groups first reported 2D magnets, i.e., CrI_3 and $\text{Cr}_2\text{Ge}_2\text{Te}_6$ (ref. 14 and ref. 15 of the revised main text, *Nature* 546, 270-273 (2017), *Nature* 546, 265-269 (2017)). Then, several groups immediately developed MTJs down to the atomic limit from 2018 to 2019, as the referee mentioned refs. 18-22 in the previous manuscript. These pioneering works have great significance because atomic MTJs are one of the ultimate goals in spintronics to realize ultrahigh-density information storage. However, after 2019, there has been no significant progress in atomic MTJs. One of the important reasons is that all previous atomic MTJs are volatile at zero field.

(2) In the revised manuscript, the referee will find that it is a spin-filtering mechanism governing the observed zero field nonvolatility instead of a particular magnetic coupling mechanism exclusively existing in the CrSBr system. To prove this, we inserted a hBN layer into the twisted interface to change the entire environment in the twisted interface, and we still observed zero field nonvolatility (see details in Part B of Extended note 2). We chose CrSBr because it is the only 2D magnet that meets our needed requirements as far as we know. Our requirements are (a) semiconductor or insulator, (b) interlayer AFM coupling and intralayer FM coupling, (c) strong uniaxial in-plane magnetic anisotropy. Refs. 26-28 in the previous manuscript only focus on the physical properties of CrSBr, so our work's importance goes far beyond theirs. Moreover, in the revised manuscript, we have demonstrated that the nonvolatility at zero field is robust even up to close to the Néel temperature (see details in response to Comment 5). Once the temperature is higher than the Néel temperature, nonvolatility at zero field disappears. Because 2D magnetism is an emergent field first reported in 2017 (ref. 14 and ref. 15 of the revised main text, *Nature* 546, 270-273 (2017), *Nature* 546, 265-269 (2017)), more abundant 2D magnets are waiting to be discovered. A room-temperature atomic MTJ will hopefully be developed in the future if a 2D magnet meets the above requirements and has a high phase-transition temperature. Thus, we firmly believe our twisted MTJs are very promising for practical applications. We also note that cryo-electronics is a very exciting technological field that has become of great interest just recently, motivated, in part, by the intense research efforts to explore quantum computing systems that operate at cryogenic temperatures (*IEEE Microwave Magazine* 22, 60-78 (2021)), and, at the same time, by several recent discoveries in proximity induced superconductivity such as, for example the superconducting and Josephson diode effects (*Nature Physics* 19, 1379-1380 (2023)). Cryogenic electronics requires advanced memories so that our all-AFM MTJs could be useful for such applications.

(3) Our work will extend 2D twistrionics. Twistrionics is an emergent field in 2D nanoelectronics that was energized by the reports of magic-angle twisted graphene in 2018 (refs. 31 and 32 of the revised main

text, *Nature* 556, 80-84 (2018), *Nature* 556, 43-50 (2018)), after which the concept of the twist angle became popular for tuning device functions. However, from magic-angle twisted graphene with superconductivity to the recent twisted MoTe₂ with a fractional quantized anomalous Hall effect (*Nature* 622, 74-79 (2023)), most of these previous reports use small twist angles (less than 10°), because moiré superlattices usually appear at small twist angles, which is useful to further tailoring energy bands. This is the mainstream concept in the 2D twistrionics community. Because of this, twists with large twist angles are rarely explored. Nevertheless, our work focuses on large twist angles and we find zero field nonvolatility due to a spin-filtering effect at the twisted interface. We believe our work will extend 2D twistrionics by using large twist angles and new mechanisms in addition to small angles and moiré superlattices. Please see more discussion in response to Comment 6.

(4) Creating ultrahigh-density and ultrafast MTJs is one of the longstanding targets in spintronics, which is facing two difficulties, i.e., spatial extent and physical constraints. For spatial extent, our twisted MTJs are shrunk to the atomic limit, which is a remarkable advance compared with conventional MTJs. One of the physical constraints in conventional FM MTJs is the stray field problem. Conventional MTJs are formed with FM layers, which inevitably produce stray fields. If FM MTJs are placed tightly, the stray field of each FM MTJ will strongly influence each other. To overcome this physical constraint, developing all-AFM MTJs without any FM component is a frontier in the domain of conventional MTJs. This year, two groups reported all-AFM MTJs constructed with noncollinear AFMs based on the classic structure of FM MTJs as shown in Figs. 1a,b of the revised main text (ref. 1 of the revised main text, *Nature* 613, 490-495 (2023); ref. 2 of the revised main text, *Nature* 613, 485-489 (2023)). In contrast, our work reports all-AFM MTJs with a novel twisted structure in the atomic limit. Thus, besides the AFM merit, the spatial extent also shows superiority over conventional FM MTJs for developing ultrahigh-density information storage by our twisted MTJs. On the other hand, because AFM materials inherently possess ultrafast terahertz (THz) spin dynamics, our twisted MTJs are also promising for ultrafast information devices.

(5) Our twisted structure is unique compared with conventional MTJs. Conventional MTJs adopt additional AFM layers to produce pinning effects and an MgO layer typically as the tunneling barrier. In our twisted structure, the extra AFM layers are not needed, and the pinning effects inherently exist owing to magnetic anisotropy. In addition, the extra MgO tunneling barrier is removed, and its role is played by the van der Waals gap, which also inherently exists in the twisted structure. The classic MTJ structure was proposed more than 30 years (e.g., ref. 8 of the revised main text, *Phys Rev B Condens Matter* 39, 6995-7002 (1989)). Our work introduces new inspirations into the conventional MTJ domain.

(Technical Comments)

Comment 3:

Discussions on the magneto-transport properties (magnetization process) of the twisted 2D-AFM tunnel junctions are just qualitative and speculative. The speculative interpretation of the magnetization process is another reason why the latter half of the manuscript is difficult to follow. I carefully read the qualitative explanations in the manuscript. But I still cannot understand why relatively robust bistable states at zero

field appear in the vicinity of $\phi = 65^\circ$ and 130° , which are not exactly the easy axis nor hard axis. Moreover, it is not explicitly mentioned in the manuscript that bistable states appear at around 140° , 175° , 240° , 300° and 360° . Figs. 3c-d are very complex.

--We thank the referee again his/her comments about the difficulty in reading our original manuscript. We have realized the places that we need to improve and correct. Especially a clarified presentation of Figs. 3c,d in the previous manuscript will help readers understand the other results. Moreover, we added new experimental and theoretical results in the revised manuscript to describe the magnetization process, which will be presented later.

Figs. 3c-d in the previous manuscript (Figs. R1a and b) are redrawn as Fig. 3c in the revised manuscript (Fig. R1c). We now place the forward-sweeping and backward-sweeping colormaps into a figure. Some repeated information is removed. In Fig. R1a,b, in the range of $\phi = 95^\circ$ - 140° , we really wanted to elucidate there are two groups of zero field nonvolatility correlating the b -axis of the top CrSBr flake and the b -axis of the bottom CrSBr flake, respectively. However, they are merged and look like one group in this angle range. In Fig. R1c, we replace the nonvolatile data of $\phi = 120^\circ$ and $\phi = 300^\circ$ of Fig. R1a,b with the volatile data to display two groups. This modification is acceptable because when the field is oriented at $\phi = 120^\circ$ and $\phi = 300^\circ$, unstable nonvolatility at zero field is observed, see Fig. R1d. This instability is ascribed to the weak pinning strength when the field is oriented near the angular bisectors of the two magnetizations of the double CrSBr flakes, see details in Extended note 1.

In Fig. R1c, we use the inverted triangles with angles to mark the positions of the crystal axes and set relevant colors for each crystal axis of the two CrSBr flakes. We further set double-headed arrows with different colors between the top panel and bottom panel, which mark the positions where zero field nonvolatility appears. The double-headed arrows with the same color as the corresponding inverted triangles indicate which crystal axis is related by the zero field nonvolatility, for example, the group of zero field nonvolatility marked by the gold double-headed arrows is related to the a -axis of the bottom CrSBr flake. Because of the 180° -relationship between negative field and positive field, two double-headed arrows with the same color indicate the same group of zero field nonvolatility. Thus, the eight double-headed arrows mark four groups of zero field nonvolatility, and each CrSBr flake has two groups.

Fig. R1 | a-d, Electrical transport results of a 35° twisted CrSBr bilayer/bilayer MTJ. **a-c**, Field orientation dependence of the tunneling current for field oriented within the *ab* plane. ± 0.3 T field and 15 mV DC bias are used. The blue arrows indicate the sweeping direction of the field. The inverted triangles with angles mark the position of the crystal axes. T-*a*: the *a*-axis of the top flake, B-*b*: the *b*-axis of the bottom flake, and so on. The double-headed arrows with different colors in **c** mark the position where zero field nonvolatility appears. **a-b** are from Fig. 3c,d of the previous main text. **c** is from Fig. 3c of the revised main text. **d**, 10 loops for the field oriented along $\Phi = 120^\circ$, showing unstable ZF nonvolatility, which is from Extended Data Fig. 9d of the revised manuscript. **e-f**, Measured tunneling current for varying maximum swept field, which are from Extended Data Fig. 10f-g of the revised manuscript.

Now, let us discuss Fig. R1c. First, we focus on the bottom CrSBr flake. There are two groups of zero field nonvolatility marked by the gold and green double-headed arrows, respectively. Then, for the top CrSBr flake, two similar groups are marked by the purple and blue double-headed arrows, respectively. These two groups for each flake are related to the *a* and *b* axes, respectively. The group related to the *b*-axis is closely aligned along the *b*-axis, but the group relevant to the *a*-axis appears with up to a $\sim 10^\circ$ deviation from the precise positions of the *a*-axis. We also found it difficult to understand the deviation when we got the experimental results the first time. Then, we realized the reason is that when the field is along the *a*-axis of one flake, ± 0.3 T is not enough to flip the magnetizations in both flakes due to the relatively small twist angle, see Fig. R1e. If increasing the amplitude of field sweeping, for example, ± 0.35 T, zero field nonvolatility appears, see Fig. R1f. In the revised manuscript, we have edited the related texts to enhance the clarity of our presentation.

The magnetization process should be relatively simple with only a few parameters (in-plane uniaxial magnetic anisotropy, magnetic moment of each monolayer, and small interlayer exchange coupling).

Therefore, the magnetization process should be reproduced by using the Stoner-Wohlfarth model or micro-magnetic simulation. Have the authors tried to reproduce the magnetization process related to Figs. 2d-e, 3c-e, 4c-e by using such simulations with appropriate magnetic parameters?

--Thanks for the excellent suggestion. In the revised manuscript, we performed Stoner-Wohlfarth model calculations, which reproduced our experimental observations nicely, please see Fig. R2 and details in Extended note 3. The basic idea is that the magnetic anisotropy of the CrSBr is maintained in the twisted MTJs, and each magnetization of the two individual CrSBr bilayers evolves with an external magnetic field independently, which makes it possible to calculate the relative angle between the two magnetizations at the twisted interface. Because the relative angle determines the tunneling conductance (G), we then obtained the results of Fig. R2. The calculated results reproduce the experimental observations well. Including zero field nonvolatility and time-reversal symmetry, the sloped relationship between tunnel currents and external field is also captured.

Fig. R2 | a-b, Calculated tunneling conductance in a dual sweeping external magnetic field (h) based on the Stoner-Wohlfarth model, which are from Fig. S8 of the extended notes.

Comment 4:

The magnetic moment of all CrSBr layers should be aligned exactly parallel to the magnetic applied field H when $\mu_0 H$ is higher than about 1 T, resulting in angle(ϕ) - independent current (i.e., conductivity). Such ϕ - independent current is observed in Figs.2d-e and Extended Data Figs.3a-b, 4b-e. However, ϕ - dependence of current is seen in Figs. 5a-e, 7a-b, 10a-b, 11b-e, even for $H > 1$ T. The authors should explain the origin of the ϕ - dependence of conductivity at high magnetic fields.

--Thank you very much for this good question. First, we agree with the referee's opinion that the magnetic moments of all CrSBr layers should be aligned exactly parallel to the external magnetic fields at a high field regime, and ϕ -independent current should appear if only considering magnetization arrangements. Indeed, as the referee observes in our original manuscript, this ϕ -independent current character is displayed in Figs. 2d-e (natural bilayer) and Extended Data Figs. 3a-b (natural bilayer), 4b-e (natural

trilayer). However, an anisotropic magnetoconductance is observed in Extended Data Figs. 5a-e (natural 4 layers), 7a-b (twisted bilayer/bilayer), 10a-b (twisted bilayer/bilayer), 11b-e (twisted monolayer/monolayer). We further highlight the disparity by comparing Extended Data Figs. 11b-e (twisted monolayer/monolayer) with Figs. 2d-e (natural bilayer), and comparing Extended Data Figs. 5a-e (natural 4 layers) with Data Figs. 4b-e (natural trilayer), by which we conclude that the anisotropic magnetoconductance prefers to emerge in thick and twisted devices.

[REDACTED]

In previous reports (e.g., ref. 26, *Adv Mater* 32, e2003240 (2020)), an anisotropic magnetoresistance is reported, see Fig. R3. In CrSBr bulk, the magnetoresistance is larger when a strong magnetic field is oriented along a -axis compared with the case of field oriented along b -axis. These results are consistent with our observations about the anisotropic magnetoconductance in the thick and twisted samples, i.e., the magnetoconductance along a -axis is lower. A reason for the anisotropic magnetoconductance is the hidden order in CrSBr at low temperatures. Telford et al. assign the hidden order to a ferromagnetic ordering along b -axis (ref. 30 of the revised main text, *Nat Mater* 21, 754-760 (2022)), and a strong magnetic field reorienting this hidden order results in anisotropic magnetoconductance. However, the origin of the hidden order is being debated. Telford et al. ascribed it to magnetic defects although Wu et al. consider incoherently coupled 1D electronic chains (*Adv Mater* 34, e2109759 (2022)), and López-Paz et al. trust spin-dimensionality crossover caused by a slowing down of the magnetic fluctuations (*Nat Commun* 13, 4745 (2022)). Our results suggest that the thickness and twisting are tightly associated with the anisotropic magnetoconductance in the strong-field regime. We thank the referee again for proposing this important question, which could be very useful for clarifying the origin of the hidden order in CrSBr.

Comment 5:

Previous works (Refs.18, 19, 21, 22, 26, 27, 28) report some data at higher temperatures. In this manuscript, on the other hand, basically all the experimental results are the data at 2K. I am especially curious about the ϕ -dependence of magneto-transport at higher temperatures, especially above T_N and T_c .

--We thank the referee for this valuable suggestion of high-temperature experiments. Please see Fig. R4. In a twisted CrSBr bilayer/bilayer device, we demonstrate that an increased temperature reduces the TMR ratio (Fig. R4k) because thermal activation leads to the twisted system being more conductive and fluctuant, for example, the maximal tunneling current is ~ 30 nA at 2 K (Fig. R4a,b), but ~ 280 nA at 50 K (Fig. R4c,d). Despite the reduced TMR ratio with increasing temperature, the nonvolatility at zero field is maintained up to TMR $\sim 20\%$ at 120 K (Fig. R4k). Remarkably, we did not observe any nonvolatility at 150 K (Fig. R4i,j). Because the T_N and T_C of CrSBr are ~ 130 K and ~ 150 K, respectively, the experimental results indicate the limitation of our twisted MTJs is not low-temperature conditions but rather the phase transition temperatures. The revised manuscript has added the temperature-dependent results as Extended Data Fig. 12.

Fig. R4 | Electrical transport results of a 40° twisted bilayer/bilayer CrSBr MTJ at different temperatures. **a-j,** Field orientation dependence of the tunneling current in the ab plane at different temperatures. 2 K in **a-b**, 50 K in **c-d**, 100 K in **e-f**, 120 K in **g-h**, and 150 K in **i-j**. The two blue arrows indicate the field sweeping direction. A constant DC bias of 1 mV is applied in all measurements. **k,** ZF-TMR ratio with 1 mV DC bias at different temperatures. **a-k** are a copy of Extended Data Fig. 12 in the revised manuscript.

Comment 6:

The van der Waals gap between 2D CrSBr layers acts as the tunnel barrier. If the tunnel barrier simply behaves as a vacuum potential, the physics is not very interesting. But there may be more than that. For example, coupling of the tunneling states across the van der Waals gap may sensitively depend on the twist angle.

--We appreciate the referee for raising the importance of van der Waals gap and twist angle. Yes, they play very important roles in our twisted structures. First, we discuss the van der Waals gap's necessities. (a) CrSBr is a van der Waals material and there are no chemical bonds in van der Waals gap. We can exfoliate CrSBr bulk down to monolayer and bilayer, allowing us to fabricate MTJ in the atomic limit. (b) To manufacture MTJs, a tunneling barrier such as MgO in conventional MTJs is necessary. The van der Waals gap as a vacuum dielectric layer plays the role of a tunneling barrier. (c) In conventional MTJs, besides the role of tunneling barrier, MgO separating two FM metal layers helps the magnetization of each FM metal layer freely flip without mutual interaction, which gives rise to binary tunneling states at zero field of either parallel ($\uparrow\uparrow$, $\downarrow\downarrow$) or antiparallel ($\uparrow\downarrow$, $\downarrow\uparrow$) configurations. A good counter-example is that the magnetization configurations of intrinsic CrSBr bilayer are always antiparallel at zero field ($\uparrow\downarrow$ or $\downarrow\uparrow$), resulting in a single tunneling state because of the inherent interlayer AFM coupling at the untwisted interface. Accordingly, another crucial role the van der Waals gap played in the twisted CrSBr is to suppress the inherent interlayer AFM coupling in the untwisted CrSBr, which is proved by (1) the negligible exchange interaction at the twisted interlayer from the first-principles calculations and (2) the observed zero field nonvolatility in the control devices with an hBN monolayer inserted at the twisted interface (see details in extended note 2).

Then, differing from the usual bottom-up deposition recipes in conventional MTJs suffering from the limitation of crystalline lattice matching, van der Waals gap makes it possible to fabricate different MTJs with various twist angles, which is the reason we can fabricate MTJs over an extensive twist angle range of 10° - 90° . Furthermore, we noticed a twist angle-dependent trade-off effect between TMR ratio and pinning strength. A smaller twist angle favors a higher TMR ratio and a lower pinning strength, and vice versa, which means we can design different MTJs with varying twist angles according to practical applications. We note that this twist angle-dependent property is ascribed to the uniaxial magnetic anisotropy and angle-dependent spin-filtering effects rather than twist angle-dependent magnetic coupling at the twisted interface (see details in Extended notes 1 and 3).

We further note this non-coupling mechanism makes our work more unique. There are indeed abundant reports reporting twist angle-dependent physical properties, and most of them use small twist angles (less than 10°). One of the most famous examples is the superconductivity in magic-angle (1.1°) twisted graphene bilayer (ref. 32 of the revised main text, *Nature* 556, 43-50 (2018)). Once missing 1.1° a little bit, superconductivity disappears, in which coupling at the twisted interface mediated by moiré period potential is the physical mechanism. To create meaningful physics, particular angles are always needed and usually predicted by theoretical calculations, e.g., the theoretical work about magic-angle graphene, *PNAS* 108, 12233-12237 (2011). People typically think large twist angles are boring because moiré period potential always appears at small twist angles. This is the mainstream concept in the community of 2D

twistronics until now. On the contrary, we focused on twisting with such “boring” large angles and observed nonvolatilities at zero field with angles of 10° - 80° instead of particularly small angles. We believe our work will extend 2D twistronics by using large twist angles and new mechanisms in addition to small angles and moiré period potential.

Finally, we apologize to the referee because of some ambiguous words such as “quasi-AFM coupling” and “quasi-FM coupling” in the previous manuscript (we have replaced them with “quasi-antiparallel configuration” and “quasi-parallel configuration” in the revised manuscript), which possibly confused the referee with considering magnetic coupling at the twisted interface mediated by exchange interactions.

Regarding the temperature dependence of conductance G above T_N , G monotonically increases with temperature in Figs. 2c (without twist), 4b (with twist), Extended Data 4a (without twist). In Fig. 3b (with twist), on the other hand, G decreases just above T_N . What is the origin of the different behaviours?

--Thanks for bringing this issue to our attention. The non-monotonic conductance-temperature relationship has been widely reported (e.g., ref. 30 of the revised main text, *Nat Mater* 21, 754-760 (2022)) in the natural CrSBr if fabricating electrodes separately without crossing to measure planar conductance (Fig. R5a). For example, the black curve in Fig. R5b displays the planar conductance-temperature results of CrSBr bilayer, which indicates an overall semiconducting behavior with conductance (G) decreasing with decreasing temperature (T). Nevertheless, in the range of ~ 215 K to T_N (~ 130 K), the G increases with decreasing T and reaches a local maximum at T_N due to reduced scattering caused by spin fluctuations as the CrSBr bilayer becomes magnetically ordered. Fig. R5c shows a similar G - T curve of CrSBr monolayer.

[REDACTED]

In our cases, we use crossed electrodes to form a tunneling junction. Out of the tunneling junction, the electrodes also come into contact with CrSBr sheets. Besides the planar conductance, the vertical conductance contributes to the final conductance. In the previous manuscript, in Figs. 2c (2L without

twist), 4b (2L with twist) and Extended Data Fig. 4a (3L without twist), because the flakes are thin, the vertical conductance dominates the final conductance, resulting in a monotonic G - T relationship. In contrast, in Fig. 3b (4L with twist), the vertical conductance is relatively suppressed owing to the thicker flake, which gives rise to an increased proportion from planar conductance for the final conductance, and thereby, a non-monotonic behavior of planar conductance is observed in Fig. 3b (Fig. R5d here). It should be emphasized that the local maximum in Fig. 3b is relatively low compared with Fig. R5b,c because of the contribution of vertical conductance.

(Minor Comments)

Comment 7:

The term “all-antiferromagnetic tunnel junction” in the title is somewhat overstatement because the sample for Fig.4, which shows the best nonvolatility at zero field, is not antiferromagnetic.

--Thanks for pointing out the question about the title. Indeed, the 55° twisted CrSBr monolayer/monolayer showing 70% zero field TMR ratio in Fig. 4 can be taken as an atomic FM MTJ because CrSBr monolayer is ferromagnetic. In twisted CrSBr bilayer/bilayer devices, because the CrSBr bilayer is an AFM magnet, we call them with “all-antiferromagnetic tunnel junction”. In a 10° twisted CrSBr bilayer/bilayer MTJ, a more than 700% zero field TMR was obtained. We researched twisted CrSBr monolayer/monolayer for a more important reason, to explain the origin of the nonvolatility in the twisted CrSBr bilayer/bilayer MTJs. The twisted CrSBr bilayer/bilayer AFM MTJ is the theme of our work, thus we prefer to keep the present title.

Comment 8:

The optical images in Fig.2b, Extended Data Figs. 3b(inset), 11a should be explained more in detail. Only the contour of CrSBr flake is indicated. The authors should indicate where are hBN, graphite and Ti-Au electrodes.

--Thanks for the advice. We have labeled graphite and Au electrodes in the following figure (updated as new Fig. 2b in the revised main text). The yellow and white dashed curves outline the top and bottom hBN.

Fig. R6 | Optical image of a device and the red dashed curve outlines the CrSBr bilayer. The yellow and white dashed curves outline the top and bottom hBN, respectively. Scale bar, 10 μm . A copy of Fig. 2b in the revised main text.

Comment 9:

In the first paragraph of page 6, the authors state

“It should be highlighted that the ZF-TMR ratio in our twisted all-AFM MTJs is far beyond the highest TMR of $\sim 138\%$ at 10 K reported to date in non-collinear AFM MTJs[ref.2]” and even approaches the recent record of 1143% for a conventional FM MTJ at 10 K[ref.37].”

This is not current because the highest MR ratio of MTJs at low temperature is 2610% at 4.2K (see K. Moges et al., *Physical Review B* 93, 134403 (2016)). Ref.37 reports the highest MR ratio at room temperature (630%) but not at low temperature.

--Thanks for reminding us of the new record. We have updated the new record in the revised main text (the 2nd paragraph of page 6), and *Physical Review B* 93, 134403 (2016) has been cited as ref. 38 in the revised manuscript.

Reviewer Reports on the First Revision:

Referees' comments:

Referee #1 (Remarks to the Author):

I appreciate the efforts made to address the concerns raised in the initial review. However, I find the resolution provided for Issue 4 to be insufficiently comprehensive. Merely calculating the intralayer exchange interactions does not adequately address the core of the issue.

It is recommended that the authors extend their computational analysis to include the calculation of the band structure and density of states for the system under investigation. Furthermore, employing density functional theory in conjunction with the non-equilibrium Green's function method to investigate the electronic transport properties of the devised device at various twist angles would be highly beneficial. Analyses focusing on spin injection efficiency, transmission spectra, transmission pathway/eigenstate, and k-resolved transmission spectra are likely to offer significant insights.

Additionally, the description of the computational methods used in the manuscript is overly general. To ensure the reproducibility of your results, it is crucial to provide more detailed information regarding the computational procedures employed. This includes specifying the software packages, the version used, the exchange-correlation functionals, the basis set, and any other relevant computational parameters. Providing such details will not only enhance the credibility of your findings but also allow other researchers in the field to replicate your work, thereby contributing to the advancement of knowledge in this area.

Referee #2 (Remarks to the Author):

We thank the authors for the detailed response, the updated result of a control experiment, and the resultant correction of the twisted interface mechanism. Our main previous concern about this work was the novelty of the underlying mechanism in this system, which can be explained by either twisted/moire-induced interlayer coupling or a concept that highly resembles the conventional MTJ, so we proposed a control experiment to answer this question. If it is the first exotic case, the magnetic coupling will be suppressed by the non-magnetic hBN, while the spin-valve effect should still survive. The authors' new data clearly shows that it is not the first case, and the main observation remains in the new device with hBN layer. In the reponse, the authors have also agreed with this and changed it accordingly that it can be nicely explained as a result of the spin alignments which are just determined by the relative crystal axes and magnetic anisotropy, and no magnetic coupling at the twisted interface is involved. On the other hand, the observed effect is robust in a wide angle range of 10-90, which basically means a random angle (>10) will work, and the twist is not crucial as long as the two layers are

decoupled. We believe this control experiment rules out the exotic mechanism and confirms a mechanism highly resembling the conventional MTJ, which, in our opinion, significantly lowers the novelty in this work.

In the response, the authors also emphasized the significance of the zero field nonvolatility in all-AFM MTJs realized in this work. However, several important works have already been made and published in this direction, but was missed in the reference.

(1) Early work on spin tunnel field-effect transistors based on a 2D magnet that realized gate-controlled non-volatile functions but with magnetic field. <https://www.nature.com/articles/s41928-019-0232-3>

(2) Magnetic field free non-volatile memristive effect has been later achieved by electrostatic gating in a few-layer 2D magnet. <https://pubs.acs.org/doi/10.1021/acs.nanolett.3c03926>

(3) Non-volatile function is also achieved through interfacial multiferroic effect in 2D magnet/polymer heterostructure. <https://www.nature.com/articles/s41928-023-00931-1>

(4) The same system of twisted CrSBr has been studied, and multistep magnetization and hysteresis were reported. <https://www.nature.com/articles/s41563-023-01735-6>

Considering the corrected mechanism discussed above and the listed existing works that have been done, we think the novelty and impact of this work are not sufficient for Nature journal, and more technical journals should be considered.

Referee #3 (Remarks to the Author):

The authors fabricated “twist-induced all-antiferromagnetic tunnel junction” consisting of two bilayers of 2-dimensional(2D) antiferromagnetic (AFM) CrSBr, in which in-plane crystallographic axes of the bilayers are twisted by about 10 – 90 degree. They observed magnetoresistance (MR) ratios > 700% at 2K and non-volatile memory function(bistable magnetic alignments) at zero magnetic field. The study is well conducted, and the methods used are appropriate. The topic “antiferromagnetic tunnel junction” is timely and of interest to researchers of the field. The authors clearly answered to my questions and comments. In the revised manuscript, the authors put more emphasis on “2D twistronics” by using additional experimental data and analyses. The revisions may strengthen the novelty of this work for the broad readers of Nature.

Author Rebuttals to First Revision:

Response to Referees' comments

Referees' comments in black; our response in blue.

Referees' comments:

Referee #1 (Remarks to the Author):

I appreciate the efforts made to address the concerns raised in the initial review. However, I find the resolution provided for Issue 4 to be insufficiently comprehensive. Merely calculating the intralayer exchange interactions does not adequately address the core of the issue.

It is recommended that the authors extend their computational analysis to include the calculation of the band structure and density of states for the system under investigation. Furthermore, employing density functional theory in conjunction with the non-equilibrium Green's function method to investigate the electronic transport properties of the devised device at various twist angles would be highly beneficial. Analyses focusing on spin injection efficiency, transmission spectra, transmission pathway/eigenstate, and k-resolved transmission spectra are likely to offer significant insights.

Additionally, the description of the computational methods used in the manuscript is overly general. To ensure the reproducibility of your results, it is crucial to provide more detailed information regarding the computational procedures employed. This includes specifying the software packages, the version used, the exchange-correlation functionals, the basis set, and any other relevant computational parameters. Providing such details will not only enhance the credibility of your findings but also allow other researchers in the field to replicate your work, thereby contributing to the advancement of knowledge in this area.

We thank the referee for appreciating our efforts to address Issue 4 but which the referee regards as not being comprehensive enough. We appreciate the detailed comments concerning this point. To calculate the properties, especially the tunneling transport properties, of the twisted CrSBr bilayers is non-trivial especially because a key ingredient to our finding of two distinct states in zero field is the in-plane magnetic anisotropy of the CrSBr monolayers themselves which allows for the magnetic moments at the interface of the twisted CrSBr layers to be oriented at an arbitrary angle (the twist angle) to each other. This makes first principle calculations very difficult so we have developed a new and what we believe is an elegant approach by inviting Prof. Evgeny Tsymbal (University Nebraska), an acknowledged theoretical expert in the field of magnetic tunnel junctions (MTJs), to collaborate with us. Through discussions with Prof. Tsymbal and his group we have come up with a new model which can quantitatively account for the angular dependence of the TMR that we measure experimentally. We discuss this model further below but, with respect to the first comments of the referee, we have added more details of the computational methods, including descriptions of the Vienna ab initio Simulation Package (VASP), the Perdew-Burke-Ernzerhof (PBE) exchange-correlation functional, K-point mesh, and computational parameters. Please refer to the details in the Methods of the revised main text.

To return to our new calculations developed in conjunction with Prof. Evgeny Tsymbal's group, we have theoretically addressed the core point of our work, namely the origin of a large TMR in CrSBr-based MTJs and the mechanism of its twist-angle dependence. We have developed a theoretical model that allows for the calculation of the TMR as a function of the twist angle in our system: this model captures the key findings of our experimental results. The model is based on a key assumption that the tunneling conductance of the MTJs is coherent and is largely controlled by the accumulated transmission of the electrons through the individual CrSBr layers. This is justified by the fact that CrSBr is insulating at low temperatures (e.g. at 2 K, since CrSBr is a semiconductor). Please note that in our earlier manuscript, we argued that the van der Waals gap (vdW gap) was itself the tunnel barrier. Now, we realize this understanding was not correct and that the tunnel barrier is rather formed predominantly from the CrSBr sheets themselves.

For simplicity, we neglect any role of the graphite electrodes (note that the graphite lattice is incommensurate with the CrSBr lattice) and assume that they simply provide electronic states for transport that are uniformly distributed in the two-dimensional Brillouin zone (2DBZ) of CrSBr.

Fig. R1 | Physical mechanism of the giant TMR effect. **a**, Schematic of the propagating state transmitted across a CrSBr barrier formed from two CrSBr monolayers between graphite (Gr) electrodes. Blue, yellow, red and grey balls correspond to Cr, S, Br and C, respectively. **b**, Schematic potential profile for a CrSBr bilayer in parallel (left panel) and antiparallel (right panel) magnetization configurations for spin-up (red) and spin-down (blue) electrons. **c**, Complex band structure of bulk CrSBr in the ferromagnetic configuration for spin-up (red dots) and spin-down (blue dots) electrons. Complex bands (left panel) connect to the real bands along the Γ -Z direction (right panel) at the Γ point. The green dashed lines show the positions of the valence band maximum (VBM) and conduction band minimum

(CBM). The VBM is set to zero energy. No real bands appear in the band gap. **d**, Distribution of the decay rates in the 2D Brillouin zone for spin-up (left figure) and spin-down (right figure) electrons calculated at energy $E = E_{\text{CBM}} - 0.1$ eV. **e**, Normalized TMR as a function twist angle θ . Grey triangles show the theoretically estimated TMR and pink circles show the experimentally measured TMR. Green curve shows the calculated results based on Slonczewski model. All figures are from Fig. 5 of the revised main text.

We now consider, in detail, the case for an MTJ with two CrSBr monolayers that form the tunnel junction that are twisted at an arbitrary angle. (The details of the model are given in the main text of the revised manuscript). We calculate the twist-angle dependence of the TMR by matching the in-plane wave vectors (quasi-momenta) \vec{k}_{\parallel} in the two twisted CrSBr layers.

In our model, we start from an untwisted CrSBr bilayer. A spin-dependent potential barrier (U) of the untwisted bilayer depends on the magnetization of the two CrSBr monolayers, parallel or antiparallel, as schematically shown in Fig. R1b. This model is based on the band structure of CrSBr, which exhibits greatly spin-polarized energy bands and different band gaps (and thus tunneling barrier height) for up(\uparrow)- and down(\downarrow)-spin electrons (see Fig. 20 of the revised Extended Data Figs). We assume that the spin-dependent evanescent states of CrSBr fully control transmission (Fig. R1a). These evanescent states are obtained by calculating the complex band structure of bulk CrSBr along the wave vector k_z (out-plane) as a function of energy (E) and in-plane wave vector $\vec{k}_{\parallel} = (k_x, k_y)$ for the case when the moments in each CrSBr layer are oriented parallel to one another. Fig. R1c shows the result of this calculation for $\vec{k}_{\parallel} = 0$. The complex bands (left panel) are connected to the real bands (right panel) at the Γ point. The complex bands in the band gap of CrSBr determine evanescent states. The evanescent states with the lowest decay rates $\kappa^{\uparrow, \downarrow}$ largely control the transmission probability $T \sim e^{-2\kappa^{\uparrow, \downarrow}d}$, where d is half of the total barrier thickness (corresponding here to 2 CrSBr layers, as mentioned earlier). We therefore calculate the distribution of the decay rates in the 2DBZ of $\vec{k}_{\parallel} = (k_x, k_y)$ at the Fermi energy (E_F) of 0.1 eV below conduction band minimum (CBM), that is, $E_F = E_{\text{CBM}} - 0.1$ eV, considering that CrSBr behaves as an n -type semiconductor; see Fig. R1d, spin-up (left figure) and spin-down (right figure) electrons. Using these decay rates and assuming \vec{k}_{\parallel} is conserved in the tunneling process (i.e. coherent tunneling), we then calculate the tunneling transmission and TMR for the spin-dependent tunneling barriers, as follows.

For the untwisted CrSBr bilayer, when the magnetizations of the two CrSBr monolayers are parallel (P), the total transmission is approximated by

$$T_P \propto \frac{1}{N_k} \sum_{\vec{k}_{\parallel}} \left[e^{-\kappa^{\uparrow}(\vec{k}_{\parallel})d} e^{-\kappa^{\downarrow}(\vec{k}_{\parallel})d} + e^{-\kappa^{\downarrow}(\vec{k}_{\parallel})d} e^{-\kappa^{\uparrow}(\vec{k}_{\parallel})d} \right] \quad (\text{R1})$$

where d is the thickness of a CrSBr monolayer, and N_k is the total number of k -points in our calculation configuration. For the antiparallel (AP) magnetization of the two CrSBr monolayers, the total transmission is given by

$$T_{AP} \propto \frac{1}{N_k} \sum_{\vec{k}_{\parallel}} \left[e^{-\kappa^{\uparrow}(\vec{k}_{\parallel})d} e^{-\kappa^{\downarrow}(\vec{k}_{\parallel})d} + e^{-\kappa^{\downarrow}(\vec{k}_{\parallel})d} e^{-\kappa^{\uparrow}(\vec{k}_{\parallel})d} \right] \quad (\text{R2})$$

Using Eqs. (R1-R2) and the calculated decay rates $\kappa^{\uparrow, \downarrow}(\vec{k}_{\parallel})$, we estimate $TMR = \frac{T_P - T_{AP}}{T_{AP}}$ to be 1790%, which is in qualitative agreement with the experimentally measured value of $\sim 1050\%$ in the untwisted

CrSBr bilayer MTJ (see Fig. 3d of the revised Extended Data Figs). We also calculated the distribution of the decay rates in the 2DBZ at other energies within the band gap of CrSBr and obtained TMR ratios ranging ~200% to ~20000% (see Fig. 21 of the revised Extended Data Figs), which corroborates our system inherently possesses a large TMR effect.

Now, we consider a twisted bilayer with a twist angle θ , which produces effects on wave vector and spin. Taking the bottom monolayer as a reference, the \vec{k}_{\parallel} of the twisted top monolayer are transformed as follows

$$k'_x = k_x \cos\theta + k_y \sin\theta \quad (R3)$$

$$k'_y = -k_x \sin\theta + k_y \cos\theta \quad (R4)$$

Then, the total transmission for the P and AP states as a function of twist angle θ can be obtained from

$$T_P(\theta) \propto \frac{1}{N_k} \sum_{k_{\parallel}} \left[e^{-\kappa^{\uparrow}(\vec{k}_{\parallel})d} e^{-\kappa^{\uparrow}(\vec{k}'_{\parallel})d} + e^{-\kappa^{\downarrow}(\vec{k}_{\parallel})d} e^{-\kappa^{\downarrow}(k'_{\parallel})d} \right] \quad (R5)$$

$$T_{AP}(\theta) \propto \frac{1}{N_k} \sum_{k_{\parallel}} \left[e^{-\kappa^{\uparrow}(\vec{k}_{\parallel})d} e^{-\kappa^{\downarrow}(\vec{k}'_{\parallel})d} + e^{-\kappa^{\downarrow}(\vec{k}_{\parallel})d} e^{-\kappa^{\uparrow}(k'_{\parallel})d} \right] \quad (R6)$$

Eqs. (R5-R6) reflect the effect of twist on the transmission due to the rotation of the in-plane wave vector but with collinear magnetizations. Further considering the influence on the spins, the spins in the twisted bilayer are non-collinear because of the twist angle θ and uniaxial in-plane anisotropy. Using a similar concept to the Slonczewski model (ref.10, *Phys. Rev. B* 39, 6995-7002 (1989)), we quantize the spin of the top monolayer with respect to the spin axis of the bottom monolayer and then obtain

$$T(\theta) \propto T_P(\theta) \cos^2 \frac{\theta}{2} + T_{AP}(\theta) \sin^2 \frac{\theta}{2} \quad (R7)$$

Eqs. (R3-R7) show clearly that the spin-dependent potential barrier of CrSBr is twist angle-dependent due to the rotations of wave vector and spin. Because we have bistable spin configurations in the twisted bilayer with relative angles of θ and $\pi - \theta$ at zero field, we obtain

$$TMR(\theta) = \frac{T(\theta) - T(\pi - \theta)}{T(\pi - \theta)} = \frac{[T_P(\theta) - T_{AP}(\theta)] \cos\theta}{T_{AP}(\theta) + [T_P(\theta) - T_{AP}(\theta)] \sin^2 \frac{\theta}{2}} \quad (R8)$$

Fig. R1e shows the calculated *normalized TMR* = $\frac{TMR(\theta)}{TMR(0)}$ in comparison with the experimental results indicating excellent agreement. By contrast a large deviation from the experimental results is found when using the standard Slonczewski model, as shown in Fig. R1e, which further shows rather the important role of coherent tunneling in our CrSBr MTJs since Slonczewski model only considers the effect of non-collinear magnetizations.

Fig. R2 | Distinct temperature-dependent behaviors between twisted and untwisted interfaces. **a**, TMR ratio corresponding to a 1 mV DC bias at different temperatures from a 40° twisted bilayer/bilayer CrSBr MTJ. Because the spin alignments are all parallel at three interfaces of the twisted bilayer/bilayer MTJ at 9 T and because ZF-on corresponds to quasi-parallel spin alignment at the twisted interface, the TMR from 9 T *versus* ZF-on mainly originates from the two untwisted interfaces. **b**, TMR ratio as a function of temperature. 2 mV DC bias is used. The pink curve is ZF-TMR obtained from a 45° twisted CrSBr monolayer/monolayer MTJ. The grey curve is TMR obtained from an untwisted bilayer MTJ using its 1.5 T and ZF conductances. **c**, Projected DOS calculated from an individual CrSBr monolayer. **d**, Projected DOS of the top monolayer of a twisted CrSBr bilayer. **a** is from Fig. 11k of the revised Extended Data Figs. **b** is from Fig. 4c of the revised main text. **c-d** are from Fig. 23 of the revised Extended Data Figs.

In addition, we have carried out new measurements on the detailed temperature dependence of the TMR in the twisted versus untwisted MTJs and have made an unexpected finding. We have found what we believe is a highly interesting phenomenon, namely distinct temperature-dependent behaviors between the twisted and untwisted interfaces. We have included these new data in our revised manuscript. Specifically, we experimentally find that the TMR decays rapidly with increasing temperature for junctions with untwisted interfaces, but rather that the TMR for twisted interfaces has a much weaker temperature dependence and, indeed, remains robust up to ~ 120 K, almost the Néel temperature (130 K). Our first observation of this new finding is shown in Fig. R2a. The results of the temperature dependence of TMR for twisted bilayer/bilayer MTJs are compared to one another. Note that the twisted bilayer/bilayer has three interfaces, i.e., one twisted interface and two untwisted interfaces (within each bilayer). The zero field TMR (ZF-TMR) calculated from ZF-on *versus* ZF-off is attributed to the twisted interface (across which the exchange coupling is weak). However, the TMR from 9 T *versus* ZF-on mainly originates from the two untwisted interfaces because the spin alignments are parallel for all three interfaces at 9 T, and ZF-on corresponds to quasi-parallel spin alignment at only the twisted interface. We then confirmed this phenomenon by comparing an untwisted bilayer MTJ and a twisted bilayer MTJ, see Fig. R2b. The reason is ascribed to the decoupling mechanism at the twisted interface, as suggested by our DFT calculations of exchange interactions, which can also be intuitively understood from interlayer orbital overlaps. In the following section, we briefly introduce this concept. Please refer to the details in the revised manuscript.

Fig. R2c shows the projected density of states (DOS) calculated from an individual CrSBr monolayer, from which it is found that Cr-*d* orbitals dominate the DOS near CBM. In the untwisted two monolayers, the Cr-*d* orbitals of the top and bottom monolayers are perfectly aligned to form a considerable interlayer orbital overlap via the intervening Br-Br atoms (Fig. 19 of the revised Extended Data Figs; Methods). Increasing temperature effectively enhances the electron interlayer hopping (spin-independent) across the vdW gap, which quickly makes the CrSBr bilayer conductive in the vertical direction. Because CrSBr is the tunneling barrier in our MTJ, the TMR quickly decays with increasing temperature. However, in the twisted bilayer, the two monolayers are decoupled from each other, in addition to our DFT calculations of

exchange interactions, which can also be understood from the orbital overlap. Since the Cr-*d* orbital is highly spatially anisotropic, rotating one monolayer with respect to the other substantially reduces the interlayer orbital overlap again via the Br-Br atoms (Fig. 19 of the revised Extended Data Figs; Methods). Moreover, Fig. R2d shows the projected DOS of the top monolayer of a twisted bilayer. Comparing Fig. R2d with Fig. R2c, the slight difference indicates the orbital overlap from the bottom monolayer is indeed suppressed at the twisted interface. Accordingly, we hypothesize that the thermally activated spin-independent interlayer hopping is less in the twisted MTJs, and the tunneling mechanism still contributes part of the vertical conductance. On the other hand, we find that the vdW gap becomes thicker in the twisted MTJs (see Fig. 25 of the revised Extended Data Figs). DFT calculations confirm that a twisted structure with a thicker vdW gap is energetically favorable (see Fig. 24 of the revised Extended Data Figs). Therefore, the thicker vdW gap can still serve as a tunneling barrier even though CrSBr sheets become conductive close to Néel temperature, giving a robust TMR in the twisted MTJs.

Conclusion and outlook

Please see more details about the physical mechanisms and new experimental results in the revised manuscript. The revised manuscript demonstrates that twist plays three important roles in our 2D all-AFM MTJs.

1. Twist angle-dependent pinning effect, which has been discussed in the last response letter (please refer to Supplementary Note 1 of the present revised manuscript). The usual AFM pinning layer is avoided, and the pinning induced by twisting in our twisted MTJs is inherent.
2. Twist-induced decoupling results in bistable tunneling states at zero field, namely zero field nonvolatility. DFT calculations show that the TMR is considerable and twist angle-dependent. We note that decoupling is not a sufficient condition for producing zero field nonvolatility. In an untwisted CrSBr monolayer/hBN monolayer/CrSBr monolayer MTJ (see Fig. 26 of the revised Extended Data Figs), we do not observe any zero field nonvolatility due to the absence of pinning correlated with twisting even if the inserted hBN layer decouples the interface.
3. The robust TMR up to near the Néel temperature in twisted MTJs further enhances the superiority of twisted MTJs to untwisted MTJs, which indicates the twisted interface is important and necessary.

We believe our new models will be applicable to many other van der Waals MTJs with twisted magnetic tunneling barriers, and room-temperature 2D all-AFM MTJs are very promising.

Referee #2 (Remarks to the Author):

We thank the authors for the detailed response, the updated result of a control experiment, and the resultant correction of the twisted interface mechanism. Our main previous concern about this work was the novelty of the underlying mechanism in this system, which can be explained by either twisted/moire-induced interlayer coupling or a concept that highly resembles the conventional MTJ, so we proposed a control experiment to answer this question. If it is the first exotic case, the magnetic coupling will be suppressed by the nonmagnetic hBN, while the spin-valve effect should still survive. The authors' new data clearly shows that it is not the first case, and the main observation remains in the new device with hBN layer. In the reponse, the authors have also agreed with this and changed it accordingly that it can be nicely explained as a result of the spin alignments which are just determined by the relative crystal axes and magnetic anisotropy, and no magnetic coupling at the twisted interface is involved. On the other hand, the observed effect is robust in a wide angle range of 10-90, which basically means a random angle (>10) will work, and the twist is not crucial as long as the two layers are decoupled. We believe this control experiment rules out the exotic mechanism and confirms a mechanism highly resembling the conventional MTJ, which, in our opinion, significantly lower the novelty in this work.

We appreciate the referee's comments and concerns about the novelty of our work. To address these valid concerns, we have carried out extensive new theoretical calculations that have allowed us to unravel the origin of the giant tunneling magnetoresistance (TMR) of our junctions. Our original statement in our earlier manuscript that the tunnel barrier was formed from the van der Waals (vdW) gap was not correct. We have now established that the origin of the TMR lies in coherent tunneling of the electrons through a tunnel barrier that is essentially formed from the individual CrSBr layers themselves. Using this approximation we have now developed a new and what we believe is a novel model of tunneling in collaboration with Prof. Evgeny Tsymbal's group at the University of Nebraska-Lincoln. They have calculated the band structure of CrSBr when the magnetic moments are oriented parallel to one another from first principles DFT methods. Then they calculate the dependence of the spin-dependent decay of the wavefunctions of electrons as a function of the in-plane parallel momentum (\vec{k}_{\parallel}) of these electrons for electrons with spins parallel and antiparallel to the magnetization of the CrSBr monolayer as they propagate across a CrSBr monolayer. As discussed in the response to referee #1 above (and as discussed in the main text and Methods) we can then calculate the \vec{k}_{\parallel} dependent decay of the electrons through a second monolayer twisted with respect to the first one by an arbitrary twist angle when the moment of this second layer is parallel to its easy axis (defined by the CrSBr orthorhombic lattice) or antiparallel to this direction. In this way, the TMR can be calculated by integrating over \vec{k}_{\parallel} . Very large TMR values are found comparable (but higher) to those that we measured. Most importantly, the functional form of the angular dependence of the TMR is in excellent agreement with our experiments. This agreement shows that the TMR is a result of coherent tunneling through tunnel barriers that are formed from the individual CrSBr layers themselves. Note that there are very few reported cases in the literature where coherent tunneling has been observed in magnetic tunnel junctions, especially rather in twisted atomic tunneling barriers.

We have also carried out extensive new experiments, in particular, to probe the temperature dependence of the TMR in the twisted and untwisted CrSBr junctions. Remarkably, we find that the TMR rapidly falls with increasing temperature for the untwisted junction but that the TMR weakly depends on temperature for the twisted junctions (see Fig. R5a,b, details will be discussed later). This means in the latter case that there is a crossover temperature where the TMR is larger for the twisted junction as compared to the untwisted case. The TMR maintains a high value for the twisted junction nearly up to the magnetic ordering temperature of CrSBr. We propose that this is because, as we show from our detailed DFT calculations and

experimental results, the vdW gap is significantly increased for the twisted junctions and Cr-*d* electrons are highly spatially anisotropic, leading to a weaker overlap of orbitals. Since CrSBr is a semiconductor with increasing temperature, there will be more thermally generated carriers within the CrSBr layers, which can readily hop (spin-independent) across the untwisted interface but with greater difficulty across the twisted interface.

Thus, we are convinced that our work has considerable novelty with respect to the underlying physics but also showing great potential for future technological applications. In particular, we believe our model can readily be applied to other tunnel barriers formed from twisted insulating 2D antiferromagnets to determine the angular dependence of the TMR.

In the following we further discuss the significance of our findings.

1. Significance of practical applications for spintronics

Our group has been exploring MTJs for more than 30 years, and we note that there remain many critical issues in the field of spintronic MTJs for practical applications. In particular, long-range dipole fields are a critical issue: these have to be eliminated to build any practical spintronic device. For the traditional ferromagnetic (FM) MTJs, the coupling of long-range dipole fields increases with inverse size of the device, which substantially hinders the improvements of device density. Currently the only effective way to eliminate the role of dipole fields is to use what is called a “synthetic antiferromagnetic (SAF)” or an “artificial antiferromagnetic” structure that the last author of our paper invented (and patented) long ago (ref. 1, *Nat. Phys.* 14, 217-219 (2018), *PRL* 64.19 (1990): 2304, *PRL* 67.25 (1991): 3598, *PRL* 66.16 (1991): 2152). The SAF structure is the coupling of the moments of two ferro (or ferri) magnetic layers antiparallel by using a spacer layer that couples the two magnetic layers antiferromagnetically via RKKY coupling. The all-antiferromagnetic (AFM) MTJ that we present here needs no such spacer layer. Its innate zero net magnetization eliminates the long-range dipole fields, thereby stands at the forefront of the spintronic MTJ field as the 3rd referee said “*The topic “antiferromagnetic tunnel junction” is timely and of interest to researchers of the field*”, and not until 2023, were all-AFM MTJs reported with the traditional structure of FM MTJs using a MgO tunnel barrier but chiral AFM electrodes (see Fig. 1b,d in the revised main text; see ref. 3-4, *Nature* 613, 485-489 (2023), *Nature* 613, 490-495 (2023)).

Fig. R3 | No zero field nonvolatility in an untwisted CrSBr monolayer/hBN monolayer/CrSBr monolayer. This figure is from Fig. 26 of the revised Extended Data Figs.

Our work presents, for the first time, a novel all-AFM MTJ device structure even down to the atomic limit by twisting two CrSBr bilayers, which is therefore at the intersection of the traditional MTJ field and the emerging 2D twistrionics field. Importantly, this device exhibits two nonvolatile states at zero field which is essential for any practical application for memory or storage. We note that our concept derives not from a simple extension of twistrionics but instead from our long-term exploration and thinking on traditional MTJs. For example, in the traditional MTJ, an additional AFM pinning layer is necessary, whose importance does not appear to be recognized by the referee in his/her comments, “*twist is not crucial as long as the two layers are decoupled.*”. In the revised manuscript, we present experimental results from an untwisted CrSBr monolayer/hBN monolayer/CrSBr monolayer MTJ (Fig. R3), which does not show any zero field nonvolatility due to the absence of pinning correlated with twisting even if the inserted hBN layer decouples the interface. We especially highlight that some sort of pinning is necessary to make an MTJ functional, and our twisted structure inherently possesses a pinning effect (please see the detailed discussion in Supplementary Note 1). Moreover, we do not use the mainstream concept in 2D twistrionics field, i.e., moiré potential produced at particular twisted angles, but instead, as the referee said, “*the observed effect is robust in a wide angle range of 10-90, which basically means a random angle (>10) will work*” (Note that, not as the referee said, 90° did not show zero field nonvolatility, which will be discussed later). In our opinion, such a robust effect in such a wide angle range is a significant advantage for practical applications compared with other moiré systems, while the referee likely thinks it is a disadvantage! Furthermore, our work’s theme is zero field nonvolatility in all-AFM MTJs, which is tightly associated with the critical issues of practical applications in the traditional MTJ field. Hence, we selected the twisted CrSBr bilayer/bilayer as our core device since CrSBr bilayer is antiferromagnetic instead of the simpler twisted monolayer/monolayer structure since the CrSBr monolayer is itself ferromagnetic. To uncover the physical mechanism, we used the twisted monolayer/monolayer devices to demonstrate the twisted interface is the source of such zero field nonvolatility. However, we agree that the origin of the physical mechanism was not sufficient in our previous revised manuscript, which has been solved in the present revised manuscript. Therefore, we very much hope that the referee can take into consideration the significance of practical applications in the spintronic MTJ field when the referee evaluates our present revised manuscript. We believe the referee will reach the same conclusion as the other referees, “*This manuscript holds considerable significance in the realm of antiferromagnetic spintronics. It offers a fresh perspective and promises groundbreaking advancements in the domain of high-density, ultrafast information devices.*”.

2. Interesting physics in our twisted MTJ

2.1 Unconventional giant TMR effect

Now, let us discuss the physical mechanism of why our system has such a large TMR ratio at zero field, even more than 700% (see the inset of Fig. 3f of the revised main text), far beyond the TMR of ~138% and 2% in the very recent traditional all-AFM MTJs (ref. 3-4, *Nature* 613, 485-489 (2023), *Nature* 613, 490-495 (2023)). We further note that large TMR ratios are not prevalent in traditional FM MTJs. After the development of several decades, the records are 1143% from a conventional CoFe/MgO/CoFe FM MTJ at 10 K (ref. 44, *Appl. Phys. Lett.* 122, 112404 (2023)) and 2610% from a quaternary Heusler alloy FM MTJ at 4.2 K (ref. 45, *Phys. Rev. B* 93 (2016)). These traditional FM MTJs and AFM MTJs use MgO as the

tunneling barrier because MgO is superable to other tunneling barriers (first experimentally demonstrated by our group 20 years ago, ref. 8, *Nat. Mater.* 3, 862-867 (2004)), which makes it almost the exclusive choice even till today. However, growing MgO is not easy, and more tunneling barriers are in intense demand. Remarkably, in our twisted MTJ, we do not need such traditional MgO-like barriers anymore. CrSBr becomes insulating at low temperatures, e.g., 2K, since it is a semiconductor. Accordingly, differing from traditional MTJs in which a nonmagnetic MgO barrier is sandwiched by two magnetic electrodes, the magnetic CrSBr stack sandwiched by two nonmagnetic electrodes plays the role of the tunnel barrier in our 2D twisted MTJs (see Fig. R4a), and interestingly, the tunnel barrier correlates with twist angle. Even more interestingly, even if the semiconductive CrSBr stack becomes conductive at high temperatures, the TMR effect is still robust in the twisted CrSBr MTJ but rapidly degrades in the untwisted CrSBr MTJ (will be discussed later). Therefore, our 2D twisted MTJ is totally different from the traditional MTJs. We further note the large TMR in the traditional MgO-based MTJs resulting from the unique energy band structure of the nonmagnetic MgO layer (ref. 6-7, *Phys. Rev. B* 63, 220403 (2001), *Phys. Rev. B* 63, 054416 (2001)). Differently, in the following, you will see a very distinct physical mechanism resulting from the magnetic CrSBr in charge of the considerable zero field nonvolatile TMR and the twist angle-dependent TMR in our experiments.

Fig. R4 | Physical mechanism of the giant TMR effect. **a**, Structure of conventional MTJ (left panel) and our 2D MTJ (right panel). **b**, Schematic of the propagating state transmitted across a CrSBr barrier formed from two CrSBr monolayers between graphite (Gr) electrodes. Blue, yellow, red and grey balls correspond to Cr, S, Br and C, respectively. **c**, Schematic potential profile for a CrSBr bilayer in parallel (left panel) and antiparallel (right panel) magnetization configurations for spin-up (red) and spin-down (blue) electrons. **d**, Complex band structure of bulk CrSBr in the ferromagnetic configuration for spin-up (red dots) and spin-down (blue dots) electrons. Complex bands (left panel) connect to the real bands along the Γ -Z direction (right panel) at the Γ point. The green dashed lines show the positions of the valence band maximum (VBM) and conduction band minimum (CBM). The VBM is set to zero energy. No real bands appear in the band gap. **e**, Distribution of the decay rate in the 2D Brillouin zone for spin-up (left figure) and spin-down (right figure) electrons calculated at energy $E = E_{\text{CBM}} - 0.1$ eV. **f**, Normalized TMR as a function twist angle θ . Grey triangles show the theoretically estimated TMR and pink circles show the experimentally measured TMR. Green curve shows the calculated results based on Slonczewski model. **a** is from Fig. 1 of the revised main text. **b-f** are from Fig. 5 of the revised main text.

We now consider, in detail, the case for an MTJ with two CrSBr monolayers that form the tunnel junction that are twisted at an arbitrary angle. (The details of the model are given in the main text of the revised manuscript). We calculate the twist-angle dependence of the TMR by matching the in-plane wave vectors (quasi-momenta) \vec{k}_{\parallel} in the two twisted CrSBr layers. For simplicity, we neglect any role of the graphite electrodes (note that the graphite lattice is incommensurate with the CrSBr lattice) and assume that they simply provide electronic states for transport that are uniformly distributed in the two-dimensional Brillouin zone (2DBZ) of CrSBr.

In our model, we start from an untwisted CrSBr bilayer. A spin-dependent potential barrier (U) of the untwisted bilayer depends on the magnetization of the two CrSBr monolayers, parallel or antiparallel, as schematically shown in Fig. R4c. This model is based on the band structure of CrSBr, which exhibits greatly spin-polarized energy bands and different band gaps (and thus tunneling barrier height) for up(\uparrow)- and down(\downarrow)-spin electrons (see Fig. 20 of the revised Extended Data Figs). We assume that the spin-dependent evanescent states of CrSBr fully control transmission (Fig. R4b). These evanescent states are obtained by calculating the complex band structure of bulk CrSBr along the wave vector k_z (out-plane) as a function of energy (E) and in-plane wave vector $\vec{k}_{\parallel} = (k_x, k_y)$ for the case when the moments in each CrSBr layer are oriented parallel to one another. Fig. R4d shows the result of this calculation for $\vec{k}_{\parallel} = 0$. The complex bands (left panel) are connected to the real bands (right panel) at the Γ point. The complex bands in the band gap of CrSBr determine evanescent states. The evanescent states with the lowest decay rates $\kappa^{\uparrow, \downarrow}$ largely control the transmission probability $T \sim e^{-2\kappa^{\uparrow, \downarrow}d}$, where d is half of the total barrier thickness (corresponding here to 2 CrSBr layers, as mentioned earlier). We therefore calculate the distribution of the decay rates in the 2DBZ of $\vec{k}_{\parallel} = (k_x, k_y)$ at the Fermi energy (E_F) of 0.1 eV below conduction band minimum (CBM), that is, $E_F = E_{\text{CBM}} - 0.1$ eV, considering that CrSBr behaves as an n -type semiconductor; see Fig. R4e, spin-up (left figure) and spin-down (right figure) electrons. Using these decay rates and assuming \vec{k}_{\parallel} is conserved in the tunneling process (i.e. coherent tunneling), we then calculate the tunneling transmission and TMR for the spin-dependent tunneling barriers, as follows.

For the untwisted CrSBr bilayer, when the magnetization of two CrSBr monolayers is parallel (P), the total transmission is approximated by

$$T_P \propto \frac{1}{N_k} \sum_{k_{\parallel}} \left[e^{-\kappa^{\uparrow}(\vec{k}_{\parallel})d} e^{-\kappa^{\downarrow}(\vec{k}_{\parallel})d} + e^{-\kappa^{\downarrow}(\vec{k}_{\parallel})d} e^{-\kappa^{\uparrow}(\vec{k}_{\parallel})d} \right] \quad (\text{R1})$$

where d is the thickness of a CrSBr monolayer, and N_k is the total number of k -points in our calculation configuration. For the antiparallel (AP) magnetization of the two CrSBr monolayers, the total transmission is given by

$$T_{AP} \propto \frac{1}{N_k} \sum_{k_{\parallel}} \left[e^{-\kappa^{\uparrow}(\vec{k}_{\parallel})d} e^{-\kappa^{\downarrow}(\vec{k}_{\parallel})d} + e^{-\kappa^{\downarrow}(\vec{k}_{\parallel})d} e^{-\kappa^{\uparrow}(\vec{k}_{\parallel})d} \right] \quad (R2)$$

Using Eqs. (R1-R2) and the calculated decay rates $\kappa^{\uparrow,\downarrow}(\vec{k}_{\parallel})$, we estimate $TMR = \frac{T_P - T_{AP}}{T_{AP}}$ to be 1790%, which is in qualitative agreement with the experimentally measured value of $\sim 1050\%$ in the untwisted CrSBr bilayer MTJ (see Fig. 3d of the revised Extended Data Figs). We also calculated the distribution of the decay rates in the 2DBZ at other energies within the band gap of CrSBr and obtained TMR ratios ranging $\sim 200\%$ to $\sim 20000\%$ (see Fig. 21 of the revised Extended Data Figs), which corroborates our system inherently possesses a large TMR effect.

Now, we consider a twisted bilayer with a twist angle θ , which produces effects on wave vector and spin. Taking the bottom monolayer as a reference, the \vec{k}_{\parallel} of the twisted top monolayer are transformed as follows

$$k'_x = k_x \cos\theta + k_y \sin\theta \quad (R3)$$

$$k'_y = -k_x \sin\theta + k_y \cos\theta \quad (R4)$$

Then, the total transmission for the P and AP states as a function of twist angle θ can be obtained from

$$T_P(\theta) \propto \frac{1}{N_k} \sum_{k_{\parallel}} \left[e^{-\kappa^{\uparrow}(\vec{k}_{\parallel})d} e^{-\kappa^{\uparrow}(\vec{k}'_{\parallel})d} + e^{-\kappa^{\downarrow}(\vec{k}_{\parallel})d} e^{-\kappa^{\downarrow}(k'_{\parallel})d} \right] \quad (R5)$$

$$T_{AP}(\theta) \propto \frac{1}{N_k} \sum_{k_{\parallel}} \left[e^{-\kappa^{\uparrow}(\vec{k}_{\parallel})d} e^{-\kappa^{\downarrow}(\vec{k}'_{\parallel})d} + e^{-\kappa^{\downarrow}(\vec{k}_{\parallel})d} e^{-\kappa^{\uparrow}(k'_{\parallel})d} \right] \quad (R6)$$

Eqs. (R5-R6) reflect the effect of twist on the transmission due to the rotation of the in-plane wave vector but with collinear magnetizations. Further considering the influence on the spins, the spins in the twisted bilayer are non-collinear because of the twist angle θ and uniaxial in-plane anisotropy. Using a similar concept to the Slonczewski model (ref.10, *Phys. Rev. B* 39, 6995-7002 (1989)), we quantize the spin of the top monolayer with respect to the spin axis of the bottom monolayer and then obtain

$$T(\theta) \propto T_P(\theta) \cos^2 \frac{\theta}{2} + T_{AP}(\theta) \sin^2 \frac{\theta}{2} \quad (R7)$$

Eqs. (R3-R7) show clearly that the spin-dependent potential barrier of CrSBr is twist angle-dependent due to the rotations of wave vector and spin. Because we have bistable spin configurations in the twisted bilayer with relative angles of θ and $\pi - \theta$ at zero field, we obtain

$$TMR(\theta) = \frac{T(\theta) - T(\pi - \theta)}{T(\pi - \theta)} = \frac{[T_P(\theta) - T_{AP}(\theta)] \cos\theta}{T_{AP}(\theta) + [T_P(\theta) - T_{AP}(\theta)] \sin^2 \frac{\theta}{2}} \quad (R8)$$

Fig. R4f shows the calculated *normalized* $TMR = \frac{TMR(\theta)}{TMR(0)}$ in comparison with the experimental results indicating excellent agreement. By contrast a large deviation from the experimental results is found when using the standard Slonczewski model, as shown in Fig. R4f, which further shows rather the important role

of coherent tunneling in our CrSBr MTJs since Slonczewski model only considers the effect of non-collinear magnetizations.

2.2 Distinct temperature-dependent behaviors between twisted and untwisted interfaces

In addition, we have carried out new measurements on the detailed temperature dependence of the TMR in the twisted versus untwisted MTJs and have made an unexpected finding. We have found what we believe is a highly interesting phenomenon, namely distinct temperature-dependent behaviors between the twisted and untwisted interfaces. We have included these new data in our revised manuscript. Specifically, we experimentally find that the TMR decays rapidly with increasing temperature for junctions with untwisted interfaces, but rather that the TMR for twisted interfaces has a much weaker temperature dependence and, indeed, remains robust up to ~ 120 K, almost the Néel temperature (130 K). Our first observation of this new finding is shown in Fig. R5a. The results of the temperature dependence of TMR for twisted bilayer/bilayer MTJs are compared to one another. Note that the twisted bilayer/bilayer has three interfaces, i.e., one twisted interface and two untwisted interfaces (within each bilayer). The zero field TMR (ZF-TMR) calculated from ZF-on *versus* ZF-off is attributed to the twisted interface (across which the exchange coupling is weak). However, the TMR from 9 T *versus* ZF-on mainly originates from the two untwisted interfaces because the spin alignments are parallel for all three interfaces at 9 T, and ZF-on corresponds to quasi-parallel spin alignment at only the twisted interface. We then confirmed this phenomenon by comparing an untwisted bilayer MTJ and a twisted bilayer MTJ, see Fig. R5b. The reason is ascribed to the decoupling mechanism at the twisted interface, as suggested by the previous DFT calculations of exchange interactions, which can also be intuitively understood from the interlayer orbital overlaps. In the following section, we briefly introduce the main concept. Please refer to the details in the revised manuscript.

Fig. R5 | Distinct temperature-dependent behaviors between twisted and untwisted interfaces. **a**, TMR ratio corresponding to a 1 mV DC bias at different temperatures from a 40° twisted bilayer/bilayer CrSBr MTJ. Because the spin alignments are all parallel at three interfaces of the twisted bilayer/bilayer MTJ at 9 T and ZF-on corresponds to quasi-parallel spin alignment at the twisted interface, the TMR from 9 T *versus* ZF-on mainly originates from the two untwisted interfaces. **b**, TMR ratio as a function of temperature. 2 mV DC bias is used. The pink curve is ZF-TMR obtained from a 45° twisted CrSBr monolayer/monolayer MTJ. The grey curve is TMR obtained from an untwisted bilayer MTJ using its 1.5 T and ZF conductances. **c**, Projected DOS calculated from an individual CrSBr monolayer. **d**, Projected DOS of the top monolayer of a twisted CrSBr bilayer. **a** is from Fig. 11k of the revised Extended Data Figs. **b** is from Fig. 4c of the revised main text. **c-d** are from Fig. 23 of the revised Extended Data Figs.

Fig. R5c shows the projected density of states (DOS) calculated from an individual CrSBr monolayer, from which it is found that Cr-*d* orbitals dominate the DOS near CBM. In the untwisted two monolayers, the Cr-*d* orbitals of the top and bottom monolayers are perfectly aligned to form a considerable interlayer orbital overlap via the intervening Br-Br atoms (Fig. 19 of the revised Extended Data Figs; Methods). Increasing temperature effectively enhances the electron interlayer hopping (spin-independent) across the vdW gap, which quickly makes the CrSBr bilayer conductive in the vertical direction. Because CrSBr is the tunneling barrier in our MTJ, the TMR quickly decays with increasing temperature. However, in the twisted bilayer, the two monolayers are decoupled from each other, in addition to the previous DFT calculations of exchange interactions, which can also be understood from the orbital overlap. Since the Cr-*d* orbital is highly spatially anisotropic, rotating one monolayer with respect to the other substantially reduces the interlayer orbital overlap via the intervening Br-Br atoms (Fig. 19 of the revised Extended Data Figs; Methods). Moreover, Fig. R5d shows the projected DOS of the top monolayer of a twisted bilayer. Comparing Fig. R5d with Fig. R5c, the slight difference indicates the orbital overlap from the bottom monolayer is indeed suppressed at the twisted interface. Accordingly, we hypothesize that the thermally activated spin-independent interlayer hopping is less in the twisted MTJs, and the tunneling mechanism still contributes part of the vertical conductance. On the other hand, we find that the vdW gap becomes thicker in the twisted MTJs (see Fig. 25 of the revised Extended Data Figs). DFT calculations confirm that a twisted structure with a thicker vdW gap is energetically favorable (see Fig. 24 of the revised Extended Data Figs). Therefore, the thicker vdW gap can still serve as a tunneling barrier even though CrSBr sheets become conductive close to Néel temperature, giving a robust TMR in the twisted MTJs.

3. Conclusion and outlook

Please see more details about the physical mechanisms and new experimental results in the revised manuscript. The revised manuscript demonstrates that twist plays three important roles in our 2D all-AFM MTJ.

1. Twisted angle-dependent pinning effect, which has been discussed in the last response letter (please refer to Supplementary Note 1 of the present revised manuscript). The usual AFM pinning layer is avoided, and the pinning induced by twisting in our twisted MTJs is inherent.
2. Twist-induced decoupling results in bistable tunneling states at zero field, namely zero field nonvolatility. DFT calculation suggests the TMR is considerable and twisted angle-dependent. Again, we note that decoupling is not a sufficient condition for producing zero field nonvolatility without pinning produced by twisting.
3. The robust TMR near the Néel temperature further enhances the superiority of twisted MTJs to untwisted MTJs, which indicates the twisted interface is important and necessary.

We believe the above physical models will be applicable to many other van der Waals MTJs with twisted magnetic tunneling barriers, and room-temperature 2D all-AFM MTJs are very promising. Moreover, our work provides new perspectives for traditional MTJs. We unveil that using a twisted magnetic tunneling barrier, namely the structure in the right panel of Fig. R4a, can likely result in a large nonvolatile TMR than the traditional structure of MTJs (left panel of Fig. R4a), which means preparing magnetic semiconductors and insulators will be very promising to replace the nonmagnetic MgO tunneling barrier in the traditional MTJs.

In the response, the authors also emphasized the significance of the zero field nonvolatility in all-AFM MTJs realized in this work. However, several important works have already been made and published in this directions, but was missed in the reference.

(1) Early work on spin tunnel field-effect transistors based on a 2D magnet that realized gate-controlled nonvolatile functions but with magnetic field. <https://www.nature.com/articles/s41928-019-0232-3>

(2) Magnetic field free nonvolatile memristive effect has been later achieved by electrostatic gating in a few-layer 2D magnet. <https://pubs.acs.org/doi/10.1021/acs.nanolett.3c03926>

(3) Nonvolatile function is also achieved through interfacial multiferroic effect in 2D magnet/polymer heterostructure. <https://www.nature.com/articles/s41928-023-00931-1>

(4) The same system of twisted CrSBr has been studied, and multistep magnetization and hysteron were reported. <https://www.nature.com/articles/s41563-023-01735-6>

We thank the referee for these four references, that is,

(1). Jian et al. *Nature Electronics* 2.4 (2019): 159-163: Spin tunnel field-effect transistors based on two-dimensional van der Waals heterostructures.

(2). Fu et al. *Nano Letters* 23.24 (2023): 11866-11873: Nonvolatile Memristive Effect in Few-Layer CrI3 Driven by Electrostatic Gating.

(3). Liang et al. *Nature Electronics* 6.3 (2023): 199-205: Small-voltage multiferroic control of two-dimensional magnetic insulators.

(4). Boix-Constant et al. *Nature Materials* 23.2 (2024): 212-218. Multistep magnetization switching in orthogonally twisted ferromagnetic monolayers.

However, we don't believe that any of these papers, although each is very interesting in its own right, is relevant to our main claim that we first report zero field nonvolatility in a 2D twisted all-AFM MTJ. Our work involves three key aspects, i.e., all-AFM MTJ, zero field nonvolatility, and twist. None of the above references concern an all-AFM MTJ nor twisting except for paper 4 that was published during the review process of our paper. However, paper 4 concerns a CrSBr bilayer that is twisted at 90 deg. We showed in our paper that 90 deg is not an interesting angle for non-volatile device applications and paper 4 does not concern an all-AFM MTJ but rather a device in which one state (only available by applying a magnetic field) has a large magnetization. The novelty in our paper is the creation of a device with 2 states in zero magnetic field with large differences in tunneling resistance and for which both states have zero magnetization. Thus, we believe that reference 4 does not lower our work's novelty but rather enhances it.

Let us briefly discuss each of the papers mentioned by the referee.

(1) This paper reports a spin tunnel field-effect transistor based on a bilayer of CrI3 using a gate voltage as the operational knob in a strong external magnetic field environment (0.76-1.77 T, Fig. 4 of (1)). This transistor cannot work in zero magnetic field: there is only one state in zero field. Moreover, the device uses a single untwisted CrI3 bilayer in which the two layers are switched between parallel and antiparallel magnetization configurations with large magnetic fields. This means, furthermore, that one state is ferromagnetic with a large net magnetization. Thus, this paper has little to do with the novel concept in our paper of a device that is always in zero net magnetization and yet which has two non-volatile tunneling states with giant differences in resistance in zero magnetic field.

(2) This paper concerns a few layers of CrI₃ sandwiched between graphene electrodes. The paper mainly concerns 5 monolayers of CrI₃. This paper reports a very small ~0.15% resistance variation of the device in zero field (calculated from Fig. 3e of (2)) after applying very large voltages (10-70V). The results in this paper, as the title shows, are likely related to voltage induced changes in the physical structure of the device. Moreover, the authors rule out any mechanism related to the magnetization of the MTJs. The very small changes in resistance (~0.15%) as well as the very large voltages preclude the use of this device as a promising memory device. In contrast, we show > 700% resistance variation at zero field, with non-volatility without any breakdown effects.

(3) This paper reports using voltage to switch the electric polarization of an organic P(VDF-TrFE) layer and further control the magnetic state of an insulating 2D ferromagnet (Cr₂Ge₂Te₆). A constant external magnetic field is needed to operate the two remanent states (see Fig. 4b-e of (3)), and reflectance magnetic circular dichroism (RMCD) shows a very weak variation of <0.03% in the magnetism of the two remanent states. This work does not provide any electrical transport data, and the device structure does not involve an MTJ. Moreover, since this device has a thick insulating overlayer across which large voltages (up to 80 V) are applied to affect the magnetic state of a few layers of a 2D ferromagnet, no vertical transport is possible to address any changes in the magnetic state. Furthermore, the device is innately ferromagnetic and thus does not concern the main content of our paper which is a device that has two non-volatile states both of which are antiferromagnetic with no net magnetization. We cannot see any significant connection with our paper.

(4) This paper is we agree somewhat related to our work. We note that this paper was published on 30 November 2023 after we submitted our paper and when our paper was in the review process. This is why we weren't aware of this paper when we submitted our paper. Nevertheless, this paper concerns a single bilayer of CrSBr and the theme of paper (4), as the referee said is "*multistep magnetization and hysteron*" which has little to do with our paper even though we use devices formed from the same material. Most importantly, this paper concerns two CrSBr monolayers that are twisted at 90 deg to one another. The paper shows two states in a magnetic field but the difference in resistance of these states is very small, only up to at most 25 % and most of the paper concerns devices with changes in resistance of only ~8%. Furthermore, this paper concerns a device which shows no non-volatile states at zero field and in which one of states obtained by applying an external magnetic field has a large net magnetization. Thus, we believe our paper in which two bilayers of CrSBr with no net magnetization are twisted at an arbitrary angle between 0 and 90 degrees but not including 90 deg (the only angle considered in paper 4) is novel and goes well beyond paper 4. In particular, we show twisted tunnel junctions formed from twisted bilayers of CrSBr that display two **non-volatile states** with **giant tunneling magnetoresistance (up to ~700 %)**, that have **no net magnetization**. We show the origin of the tunneling magnetoresistance. We also present data on the **twist angle dependence** that matches well with calculations.

Moreover, we find (4) further enhances the novelty of our work. Please allow us to explain in more detail.

- a) The main observations of (4), i.e., multistep magnetization owing to magnetic domains produced by the competition of external field (nonzero field), magnetocrystalline anisotropy, and dipole field, were also observed and discussed in our earlier manuscript about the multiple transitions and kinks in our experimental results. Because such multiple transitions and kinks appearing at a finite external field are not our theme, we simply discussed them in a supplementary note (see the end of Supplementary Note 2). In the present revised manuscript, (4) has been cited as ref. 3 in the supplementary notes file to discuss related phenomena.
- b) (4) merely reports a single twisted angle of 90° and a single twisted structure of monolayer/monolayer. It should be noted we also reported a 90° twisted device in our initial

manuscript. Our theme is zero field nonvolatility in the twisted bilayer/bilayer MTJs, which cannot appear in the 90° twisted device according to our physical model. To prove this, we performed control experiments on a 90° twisted device. Please see Fig. 28 of the revised Extended Data Figs in the revised manuscript. Hence, (4) does not involve any core results of our work. We further note that even our 90° twisted device is different from (4) because we used a twisted bilayer/bilayer structure, and we observed an interesting 4-fold rotational symmetry in the angle-dependent pattern of tunneling current (Fig. 28e,f of the revised Extended Data Figs). More importantly, we found that (4) also shows no zero field nonvolatility, which means the control results of our work can be independently reproduced by another group, and our model is correct. In the present revised manuscript, (4) has been cited as ref. 23 to discuss related texts. We highlight to realize zero field nonvolatility, 90° twisted angle must be avoided!

- c) Our work goes well beyond paper (4). Our twisted angle ranges from 10° to 90° . (4) merely shows a single twisted angle of 90° and a single twisted structure of monolayer/monolayer. It should be noted that our core structure is the twisted bilayer/bilayer structure instead of the twisted monolayer/monolayer structure in (4). Because CrSBr monolayer is ferromagnetic and bilayer is antiferromagnetic, the significance of the twisted bilayer/bilayer is far beyond the twisted monolayer/monolayer structure in the spintronics field, as previously discussed. To explain the physical origin of zero field nonvolatility in the twisted bilayer/bilayer structure, we also fabricated the twisted monolayer/monolayer devices and inserted an hBN layer into the twisted interface. Moreover, (4) does not discuss the physical origin of TMR at the twisted interface. By contrast, in our revised manuscript, we established the physical models of the tunneling mechanism and the different temperature-dependent manners between twisted and untwisted MTJs. The novel and comprehensive results and understanding in our work will stimulate much work in twisted all-AFM structures.

Based on the above discussion, the references of (1-3) mentioned by the referee demonstrate that 2D magnets provide a new platform to explore exotic nanodevices. We have cited them as ref. 22, ref. 20, ref. 21 in the revised manuscript, respectively.

Considering the corrected mechanism discussed above and the listed existing works that have been done, we think the novelty and impact of this work are not sufficient for Nature journal, and more technical journals should be considered.

We have substantially rewritten our paper so as to include both our new theoretical models and our new experimental findings. We believe that we have fully addressed the comments by the referee. We hope the referee will be convinced after the referee reads the revised manuscript and the response letter, and we sincerely hope our work can get the referee's recommendation for publication in Nature.

Referee #3 (Remarks to the Author):

The authors fabricated “twist-induced all-antiferromagnetic tunnel junction” consisting of two bilayers of 2-dimensional(2D) antiferromagnetic (AFM) CrSBr, in which in-plane crystallographic axes of the bilayers are twisted by about 10 – 90 degree. They observed magnetoresistance (MR) ratios > 700% at 2K and nonvolatile memory function(bistable magnetic alignments) at zero magnetic field. The study is well conducted, and the methods used are appropriate. The topic “antiferromagnetic tunnel junction” is timely and of interest to researchers of the field. The authors clearly answered to my questions and comments. In the revised manuscript, the authors put more emphasis on “2D twistrionics” by using additional experimental data and analyses. The revisions may strengthen the novelty of this work for the broad readers of Nature.

We thank the referee for the very positive evaluation of our previous revised manuscript and for the recommendation for publication in Nature.

Reviewer Reports on the Second Revision:

Referees' comments:

Referee #1 (Remarks to the Author):

In the resubmitted manuscript, the authors have addressed their misunderstanding of the role of the vdW gap in tunneling through reanalysis and theoretical recalculations. They now argue that the main factor is the inherent properties of the two-dimensional insulating material CrSBr itself, a perspective I concur with. The revised manuscript offers a richer physical explanation compared to the initial version. The authors have replaced the previous DFT calculations with the suggested transport calculation model, which better elucidates the core issues of their work. This substantial theoretical work effectively resolves the concerns I previously raised.

Overall, I believe the current version of the manuscript meets the publication standards of Nature, and I am pleased to recommend it for publication.

Referee #2 (Remarks to the Author):

We thank the authors for the detailed response regarding our previous comments. We appreciate the new additional theoretical calculations, but the concerns about the novelty remain.

(1) The authors argue the novelty in three key aspects, i.e., all-AFM MTJ, zero field nonvolatility, and twist. As discussed in the comment and response, these key advances have been achieved in previous studies: all-AFM MTJs have been achieved even at room temperature recently; zero field nonvolatility has been achieved in other MTJs; twisted CrSBr MTJ has been reported. Although the current study can be more comprehensive, in our opinion, the overall novelty and new advances made here are insufficient for the Nature journal.

(2) The authors highlight the practical applications, however, we note that the current study is still far from practical applications. The TMR is high but not the highest, and it is not valid to compare TMR at different temperatures. The MTJ here is based on 2D material systems, just like in other 2D MTJ studies, and is thus far from practical applications. The operating temperature is also far from room temperature.

(3) The authors add the novelty of claiming coherent tunneling and argue that “very few reported cases in literature”, but this new claim is only supported by new calculations instead of new direct experimental evidence.

(4) The authors add TMR temperature dependence, which can make it a more systematic study, but this does not contribute to the main novelty directly.

(5) Our comments remain negative about the authors' correction of the physical mechanism in the previous response, which was represented to support the main physics novelty but was later corrected. That said, our opinions remain the same as in our last report.